# FedChain: Chained Algorithms for Near-Optimal Communication Cost in Federated Learning

**Charlie Hou**
Department of Electrical and Computer Engineering
Carnegie Mellon University
Pittsburgh, Pennsylvania, USA
`charlieh@andrew.cmu.edu`

**Kiran K. Thekumparampil**
Department of Electrical and Computer Engineering
University of Illinois at Urbana-Champaign
Champaign, Illinois, USA
`thekump2@illinois.edu`

**Giulia Fanti**
Department of Electrical and Computer Engineering
Carnegie Mellon University
Pittsburgh, Pennsylvania, USA
`gfanti@andrew.cmu.edu`

**Sewoong Oh**
Allen School of Computer Science and Engineering
University of Washington
Seattle, Washington, USA
`sewoong@cs.washington.edu`

## Abstract

Federatedd learning (FL) aims to minimize the communication complexity of training a model over heterogeneous data distributed across many clients. A common approach is local update methods, where clients take multiple optimization steps over local data before communicating with the server (e.g., FedAvg). Local update methods can exploit similarity between clients' data. However, in existing analyses, this comes at the cost of slow convergence in terms of the dependence on the number of communication rounds $R$. On the other hand, global update methods, where clients simply return a gradient vector in each round (e.g., SGD), converge faster in terms of $R$ but fail to exploit the similarity between clients even when clients are homogeneous. We propose FedChain, an algorithmic framework that combines the strengths of local update methods and global update methods to achieve fast convergence in terms of $R$ while leveraging the similarity between clients. Using FedChain, we instantiate algorithms that improve upon previously known rates in the general convex and PL settings, and are near-optimal (via an algorithm-independent lower bound that we show) for problems that satisfy strong convexity. Empirical results support this theoretical gain over existing methods.

## 1 Introduction

In federated learning (FL) (McMahan et al., 2017; Kairouz et al., 2019; Li et al., 2020), distributed clients interact with a central server to jointly train a single model without directly sharing their data with the server. The training objective is to solve the following minimization problem:

$$\min_x \left[ F(x) = \frac{1}{N} \sum_{i=1}^N F_i(x) \right] \tag{1}$$

where variable $x$ is the model parameter, $i$ indexes the clients (or devices), $N$ is the number of clients, and $F_i(x)$ is a loss that only depends on that client's data. Typical FL deployments have two properties that make optimizing Eq. (1) challenging: ($i$) *Data heterogeneity:* We want to minimize the average of the expected losses $F_i(x) = \mathbb{E}_{z_i \sim \mathcal{D}_i}[f(x; z_i)]$ for some loss function $f$ evaluated on client data $z_i$ drawn from a client-specific distribution $\mathcal{D}_i$. The convergence rate of the optimization depends on the heterogeneity of the local data distributions, $\{\mathcal{D}_i\}_{i=1}^N$, and this dependence is captured by a popular notion of client heterogeneity, $\zeta^2$, defined as follows:

$$\zeta^2 := \max_{i \in [N]} \sup_x \|\nabla F(x) - \nabla F_i(x)\|^2 . \tag{2}$$

This captures the maximum difference between a local gradient and the global gradient, and is a standard measure of heterogeneity used in the literature (Woodworth et al., 2020a; Gorbunov et al., 2020; Deng et al., 2020; Woodworth et al., 2020a; Gorbunov et al., 2020; Yuan et al., 2020; Deng et al., 2021; Deng & Mahdavi, 2021). ($ii$) *Communication cost:* In many FL deployments, communication is costly because clients have limited bandwidths. Due to these two challenges, most federated optimization algorithms alternate between local rounds of computation, where each client locally processes only their own data to save communication, and global rounds, where clients synchronize with the central server to resolve disagreements due to heterogeneity in the locally updated models.

Several federated algorithms navigate this trade-off between reducing communication and resolving data heterogeneity by modifying the amount and nature of *local and global computation* (McMahan et al., 2017; Li et al., 2018; Wang et al., 2019b;a; Li et al., 2019; Karimireddy et al., 2020b;a; Al-Shedivat et al., 2020; Reddi et al., 2020; Charles & Konečný, 2020; Mitra et al., 2021; Woodworth et al., 2020a). These first-order federated optimization algorithms largely fall into one of two camps: ($i$) clients in *local update methods* send updated models after performing multiple steps of model updates, and ($ii$) clients in *global update methods* send gradients and do not perform any model updates locally. Examples of ($i$) include FedAvg (McMahan et al., 2017), SCAFFOLD (Karimireddy et al., 2020b), and FedProx (Li et al., 2018). Examples of ($ii$) include SGD and Accelerated SGD (ASG), where in each round $r$ the server collects from the clients gradients evaluated on local data at the current iterate $x^{(r)}$ and then performs a (possibly Nesterov-accelerated) model update.

As an illustrating example, consider the scenario when $F_i$'s are $\mu$-strongly convex and $\beta$-smooth such that the condition number is $\kappa = \beta/\mu$ as summarized in Table 1, and also assume for simplicity that all clients participate in each communication round (i.e., full participation). Existing convergence analyses show that local update methods have a favorable dependence on the heterogeneity $\zeta$. For example, Woodworth et al. (2020a) show that FedAvg achieves an error bound of $\tilde{\mathcal{O}}((\kappa\zeta^2/\mu)R^{-2})$ after $R$ rounds of communication. Hence FedAvg achieves theoretically faster convergence for smaller heterogeneity levels $\zeta$ by using local updates. However, this favorable dependency on $\zeta$ comes at the cost of slow convergence in $R$. On the other hand, global update methods converge faster in $R$, with error rate decaying as $\tilde{\mathcal{O}}(\Delta \exp(-R/\sqrt{\kappa}))$ (for the case of ASG), where $\Delta$ is the initial function value gap to the (unique) optimal solution. This is an exponentially faster rate in $R$, but it does not take advantage of small heterogeneity even when client data is fully homogeneous.

Our main contribution is to design a novel family of algorithms that combines the strengths of local and global methods to achieve a faster convergence while maintaining the favorable dependence on the heterogeneity, thus achieving an error rate of $\tilde{\mathcal{O}}((\zeta^2/\mu) \exp(-R/\sqrt{\kappa}))$ in the strongly convex scenario. By adaptively switching between this novel algorithm and the existing best global method, we can achieve an error rate of $\tilde{\mathcal{O}}(\min\{\Delta, \zeta^2/\mu\} \exp(-R/\sqrt{\kappa}))$. We further show that this is near-optimal by providing a matching lower bound in Thm. 5.4. This lower bound tightens an existing one from (Woodworth et al., 2020a) by considering a smaller class of algorithms that includes those presented in this paper.

We propose FedChain, a unifying *chaining framework* for federated optimization that enjoys the benefits of both local update methods and global update methods. For a given total number of rounds $R$, FedChain first uses a local-update method for a constant fraction of rounds and then switches to a global-update method for the remaining rounds. The first phase exploits client homogeneity when possible, providing fast convergence when heterogeneity is small. The second phase uses the unbiased stochastic gradients of global update methods to achieve a faster convergence rate in the heterogeneous setting. The second phase inherits the benefits of the first phase through the iterate output by the local-update method. An *instantiation* of FedChain consists of a combination of a specific local-update method and global-update method (e.g., FedAvg $\to$ SGD).

**Contributions.** We propose the FedChain framework and analyze various instantiations under strongly convex, general convex, and nonconvex objectives that satisfy the PL condition.[1] Achievable rates are summarized in Tables 1, 2 and 4. In the strongly convex setting, these rates nearly match the algorithm-independent lower bounds we introduce in Theorem 5.4, which shows the near-optimality of FedChain. For strongly convex functions, chaining is optimal up to a factor that decays exponentially in the condition number $\kappa$. For convex functions, it is optimal in the high-heterogeneity regime, when

---

[1] $F$ satisfies the $\mu$-PL condition if $2\mu(F(x) - F(x^*)) \leq \|\nabla F(x)\|^2$.

---

**Algorithm 1** Federated Chaining (FedChain)

---

**Input:** $\mathcal{A}_{\text{local}}$ local update algorithm, $\mathcal{A}_{\text{global}}$ centralized algorithm, initial point $\hat{x}_0$, $K$, rounds $R$
  ▷ **Run** $\mathcal{A}_{\text{local}}$ for $R/2$ rounds

---

$\hat{x}_{1/2} \leftarrow \mathcal{A}_{\text{local}}(\hat{x}_0)$
  ▷ **Choose the better point between** $\hat{x}_0$ and $\hat{x}_{1/2}$

---

Sample $S$ clients $\mathcal{S} \subseteq [N]$
Draw $\hat{z}_{i,k} \sim \mathcal{D}_i$, $i \in \mathcal{S}$, $k \in \{0, \ldots, K-1\}$
$\hat{x}_1 \leftarrow \arg\min_{x \in \{\hat{x}_0, \hat{x}_{1/2}\}} \frac{1}{SK} \sum_{i \in \mathcal{S}} \sum_{k=0}^{K-1} f(x; \hat{z}_{i,k})$
  ▷ **Finish convergence with** $\mathcal{A}_{\text{global}}$ for $R/2$ rounds

---

$\hat{x}_2 \leftarrow \mathcal{A}_{\text{global}}(\hat{x}_1)$
Return $\hat{x}_2$

---

$\zeta > \beta D R^{1/2}$. For nonconvex PL functions, it is optimal for constant condition number $\kappa$. In all three settings, FedChain instantiations *improve* upon previously known worst-case rates in certain regimes of $\zeta$. We further demonstrate the empirical gains of our chaining framework in the convex case (logistic regression on MNIST) and in the nonconvex case (ConvNet classification on EMNIST (Cohen et al., 2017) and ResNet-18 classification on CIFAR-100 (Krizhevsky, 2009)).

## 1.1 RELATED WORK

The convergence of FedAvg in convex optimization has been the subject of much interest in the machine learning community, particularly as federated learning (Kairouz et al., 2019) has increased in popularity. The convergence of FedAvg for convex minimization was first studied in the homogeneous client setting (Stich, 2018; Wang & Joshi, 2018; Woodworth et al., 2020b). These rates were later extended to the heterogeneous client setting (Khaled et al., 2020; Karimireddy et al., 2020b; Woodworth et al., 2020a; Koloskova et al., 2020), including a lower bound (Woodworth et al., 2020a). The only current known analysis for FedAvg that shows that FedAvg can improve on SGD/ASG in the heterogeneous data case is that of Woodworth et al. (2020a), which has been used to prove slightly tighter bounds in subsequent work Gorbunov et al. (2020).

Many new federated optimization algorithms have been proposed to improve on FedAvg and SGD/ASG, such as SCAFFOLD (Karimireddy et al., 2020b), S-Local-SVRG (Gorbunov et al., 2020), FedAdam (Reddi et al., 2020), FedLin (Mitra et al., 2021), FedProx (Li et al., 2018). However, under full participation (where all clients participate in a communication round; otherwise the setting is partial participation) existing analyses for these algorithms have not demonstrated any improvement over SGD (under partial participation, SAGA (Defazio et al., 2014)). In the smooth nonconvex full participation setting, MimeMVR (Karimireddy et al., 2020a) was recently proposed, which improves over SGD in the low-heterogeneity setting. Currently MimeMVR and SGD are the two best algorithms for worst-case smooth nonconvex optimization with respect to communication efficiency, depending on client heterogeneity. In partial participation, MimeMVR and SAGA are the two best algorithms for worst-case smooth nonconvex optimization, depending on client heterogeneity.

Lin et al. (2018) proposed a scheme, post-local SGD, opposite ours by switching from a global-update method to a local-update method (as opposed to our proposal of switching from local-update method to global-update method). They evaluate the setting where $\zeta = 0$. The authors found that while post-local SGD has a training accuracy worse than SGD, the test accuracy can be better (an improvement of 1% test accuracy on ResNet-20 CIFAR-10 classification), though theoretical analysis was not provided. Wang & Joshi (2019) decrease the number ($K$) of local updates to transition from FedAvg to SGD in the homogeneous setting. This is conceptually similar to FedChain, but they do not show order-wise convergence rate improvements. Indeed, in the homogeneous setting, a simple variant of ASG achieves the optimal worst-case convergence rate (Woodworth et al., 2021).

## 2 SETTING

Federated optimization proceeds in rounds. We consider the partial participation setting, where at the beginning of each round, $S \in \mathbb{Z}_+$ out of total $N$ clients are sampled uniformly at random without

replacement. Between each global communication round, the sampled clients each access either (i) their own stochastic gradient oracle $K$ times, update their local models, and return the updated model, or (ii) their own stochastic function value oracle $K$ times and return the average value to the server. The server aggregates the received information and performs a model update. For each baseline optimization algorithm, we analyze the sub-optimality error after $R$ rounds of communication between the clients and the server. Suboptimality is measured in terms of the function value $\mathbb{E}F(\hat{x}) - F(x^*)$, where $\hat{x}$ is the solution estimate after $R$ rounds and $x^* = \arg\min_x F(x)$ is a (possibly non-unique) optimum of $F$. We let the estimate for the initial suboptimality gap be $\Delta$ (Assumption B.9), and the initial distance to a (not necessarily unique) optimum be $D$ (Assumption B.10). If applicable, $\epsilon$ is the target expected function value suboptimality.

We study three settings: strongly convex $F_i$'s, convex $F_i$'s and $\mu$-PL $F$; for formal definitions, refer to App. B. Throughout this paper, we assume that the $F_i$'s are $\beta$-smooth (Assumption B.4). If $F_i$'s are $\mu$-strongly convex (Assumption B.1) or $F$ is $\mu$-PL (Assumption B.3), we denote $\kappa = \beta/\mu$ as the condition number. $\mathcal{D}_i$ is the data distribution of client $i$. We define the heterogeneity of the problem as $\zeta^2 := \max_{i\in[N]} \sup_x \|\nabla F(x) - \nabla F_i(x)\|^2$ in Assumption B.5. We assume unless otherwise specified that $\zeta^2 > 0$, i.e., that the problem is heterogeneous. We assume that the client gradient variance is upper bounded by $\sigma^2$ (Assumption B.6). We also define the analogous quantities for function value oracle queries: $\zeta_F^2$ Assumption B.8, $\sigma_F^2$ Assumption B.7. We use the notation $\tilde{\mathcal{O}}, \tilde{\Omega}$ to hide polylogarithmic factors, and $\mathcal{O}, \Omega$ if we are only hiding constant factors.

# 3 FEDERATED CHAINING (FEDCHAIN) FRAMEWORK

We start with a toy example (Fig. 1) to illustrate FedChain. Consider two strongly convex client objectives: $F_1(x) = (1/2)(x-1)^2$ and $F_2(x) = (x+1)^2$. The global objective $F(x) = (F_1(x) + F_2(x))/2$ is their average. Fig. 1 (top) plots the objectives, and Fig. 1 (bottom) displays their gradients. Far from the optimum, due to strong convexity, all client gradients point towards the optimal solution; specifically, this occurs when $x \in (-\infty, -1] \cup [1, \infty)$. In this regime, clients can use local steps (e.g., FedAvg) to reduce communication without sacrificing the consistency of per-client local updates. On the other hand, close to the optimum, i.e., when $x \in (-1, 1)$, some client gradients may point away from the optimum. This suggests that clients should *not* take local steps to avoid driving the global estimate away from the optimum. Therefore, when we are close to the optimum, we use an algorithm without local steps (e.g. SGD), which is less affected by client gradient disagreement.

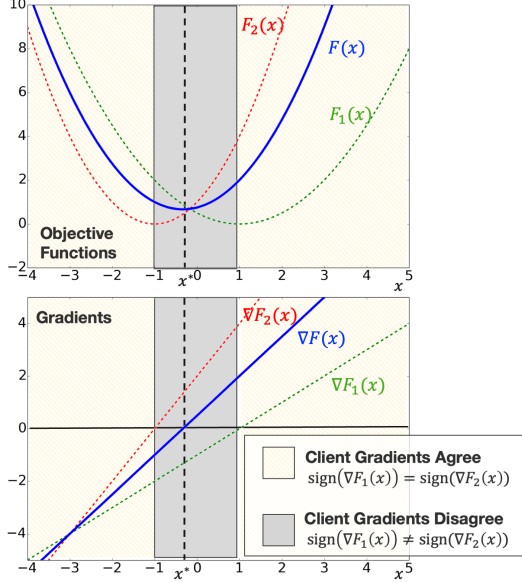

Figure 1: A toy example illustrating FedChain. We have two client objectives: $F_1(x)$ and $F_2(x)$. $F(x)$ is their average. The top plot displays the objectives and the bottom plot displays the gradients. In regions where client gradients agree in direction, i.e. $(-\infty, -1] \cup [1, \infty)$ it may be better to use an algorithm with local steps (like FedAvg), and in the region where the gradient disagree in direction, i.e. $(-1, 1)$ it may be better to use an algorithm without local steps (like SGD).

This intuition seems to carry over to the nonconvex setting: Charles et al. (2021) show that over the course of FedAvg execution on neural network StackOverflow next word prediction, client gradients become more orthogonal.

**FedChain:** To exploit the strengths of both local and global update methods, we propose the federated chaining (FedChain) framework in Alg. 1. There are three steps: (1) Run a local-update method $\mathcal{A}_{\text{local}}$, like FedAvg. (2) Choose the better point between the output of $\mathcal{A}_{\text{local}}$ (which we denote $\hat{x}_{1/2}$), and the initial point $\hat{x}_0$ to initialize (3) a global-update method $\mathcal{A}_{\text{global}}$ like SGD. Note that when heterogeneity is large, $\mathcal{A}_{\text{local}}$ can actually

Table 1: Rates for the strongly convex case. ♣ Rate requires $R \geq N/S$. ♠ Rate requires $S = N$.

| Method/Analysis | $\mathbb{E}F(\hat{x}) - F(x^*) \leq \tilde{\mathcal{O}}(\cdot)$ |
|---|---|
| *Centralized Algorithms* | |
| SGD | $\Delta \exp(-\kappa^{-1}R) + (1 - \frac{S}{N})\frac{\zeta^2}{\mu SR}$ |
| ASG | $\Delta \exp(-\kappa^{-\frac{1}{2}}R) + (1 - \frac{S}{N})\frac{\zeta^2}{\mu SR}$ |
| *Federated Algorithms* | |
| FedAvg (Karimireddy et al., 2020b) | $\Delta \exp(-\kappa^{-1}R) + \kappa(\frac{\zeta^2}{\mu})R^{-2}$ |
| FedAvg (Woodworth et al., 2020a) | $\kappa(\frac{\zeta^2}{\mu})R^{-2}$♠ |
| SCAFFOLD (Karimireddy et al., 2020b) | $\Delta \exp(-\min\{\kappa^{-1}, \frac{S}{N}\}R)$♣ |
| *This paper* | |
| FedAvg $\to$ SGD (Thm. 4.1) | $\min\{\Delta, \frac{\zeta^2}{\mu}\} \exp(-\kappa^{-1}R) + (1 - \frac{S}{N})\frac{\zeta^2}{\mu SR}$ |
| FedAvg $\to$ SAGA (Thm. 4.3) | $\min\{\Delta, \frac{\zeta^2}{\mu}\} \exp(-\min\{\kappa^{-1}, \frac{S}{N}\}R)$♣ |
| FedAvg $\to$ ASG (Thm. 4.2) | $\min\{\Delta, \frac{\zeta^2}{\mu}\} \exp(-\kappa^{-\frac{1}{2}}R) + (1 - \frac{S}{N})\frac{\zeta^2}{\mu SR}$ |
| FedAvg $\to$ SSNM (Thm. 4.4) | $\kappa \min\{\Delta, \frac{\zeta^2}{\mu}\} \exp(-\min\{\sqrt{\frac{S}{N\kappa}}, \frac{S}{N}\}R)$♣ |
| Algo.-independent LB (Thm. 5.4) | $\min\{\Delta, \kappa^{-\frac{3}{2}}(\frac{\zeta^2}{\beta})\} \exp(-\kappa^{-\frac{1}{2}}R)$ |

output an iterate with *higher* suboptimality gap than $\hat{x}_0$. Hence, selecting the point (between $\hat{x}_0$ and $\hat{x}_{1/2}$) with a smaller $F$ allows us to adapt to the problem's heterogeneity, and initialize $\mathcal{A}_{\text{global}}$ appropriately to achieve good convergence rates. To compare $\hat{x}_{1/2}$ and the initial point $\hat{x}_0$, we approximate $F(\hat{x}_0)$ and $F(\hat{x}_{1/2})$ by averaging; we compute $\frac{1}{SK} \sum_{i\in\mathcal{S}, k\in[K]} f(x; \hat{z}_{i,k})$ for $\hat{x}_{1/2}, \hat{x}_0$, where $K$ is also the number of local steps per client per round in $\mathcal{A}_{\text{local}}$, and $\mathcal{S}$ is a sample of $S$ clients. In practice, one might consider adaptively selecting how many rounds of $\mathcal{A}_{\text{local}}$ to run, which can potentially improve the convergence by a constant factor. Our experiments in App. J.1 show significant improvement when using only 1 round of $\mathcal{A}_{\text{local}}$ with a large enough $K$ for convex losses.

## 4 CONVERGENCE OF FEDCHAIN

We first analyze FedChain (Algo. 1) when $\mathcal{A}_{\text{local}}$ is FedAvg and $\mathcal{A}_{\text{global}}$ is (Nesterov accelerated) SGD. The first theorem is without Nesterov acceleration and the second with Nesterov acceleration.

**Theorem 4.1 (FedAvg $\to$ SGD).** *Suppose that client objectives $F_i$'s and their gradient queries satisfy Assumptions B.4, B.5, B.6, B.7 and B.8. Then running FedChain (Algo. 1) with $\mathcal{A}_{local}$ as FedAvg (Algo. 4) with the parameter choices of Thm. E.1, and $\mathcal{A}_{global}$ as SGD (Algo. 2) with the parameter choices of Thm. D.1, we get the following rates [2]:*

- **Strongly convex:** *If $F_i$'s satisfy Assumption B.1 for some $\mu > 0$ then there exists a finite $K$ above which we get, $\mathbb{E}F(\hat{x}_2) - F(x^*) \leq \tilde{\mathcal{O}}(\min\{\Delta, \zeta^2/\mu\} \exp(-R/\kappa) + (1 - S/N)\zeta^2/(\mu SR))$.*

- **General convex:** *If $F_i$'s satisfy Assumption B.2, then there exists a finite $K$ above which we get the rate $\mathbb{E}F(\hat{x}_2) - F(x^*) \leq \tilde{\mathcal{O}}(\min\{\beta D^2/R, \sqrt{\beta\zeta D^3}/\sqrt{R}\} + (\sqrt[4]{1 - S/N})(\sqrt{\beta\zeta D^3}/\sqrt[4]{SR}))$.*

- **PL condition:** *If $F_i$'s satisfy Assumption B.3 for $\mu > 0$, then there exists a finite $K$ above which we get the rate the rate $\mathbb{E}F(\hat{x}_2) - F(x^*) \leq \tilde{\mathcal{O}}(\min\{\Delta, \zeta^2/\mu\} \exp(-R/\kappa) + (1 - S/N)(\kappa\zeta^2/\mu SR))$.*

---

[2] We ignore variance terms and $\zeta_F^2 = \max_{i\in[N]} \sup_x (F_i(x) - F(x))^2$ as the former can be made negligible by increasing $K$ and the latter can be made zero by running the better of FedAvg $\to$ SGD and SGD instead of choosing the better of $\hat{x}_{1/2}$ and $\hat{x}_0$ as in Algo. 1. Furthermore, $\zeta_F^2$ terms are similar to the $\zeta^2$ terms. The rest of the theorems in the main paper will also be stated this way. To see the formal statements, see App. F.

Table 2: Rates for the general convex case. ♣ Rate requires $R \geq \frac{N}{S}$. ♠ Rate requires $S = N$. ♦ Analysis from Karimireddy et al. (2020b).

| Method/Analysis | $\mathbb{E}F(\hat{x}) - F(x^*) \leq \tilde{\mathcal{O}}(\cdot)$ |
|---|---|
| *Centralized Algorithms* | |
| SGD | $\frac{\beta D^2}{R} + \sqrt{1 - \frac{S}{N}} \frac{\zeta D}{\sqrt{SR}}$ |
| ASG | $\frac{\beta D^2}{R^2} + \sqrt{1 - \frac{S}{N}} \frac{\zeta D}{\sqrt{SR}}$ |
| *Federated Algorithms* | |
| FedAvg♦ | $\frac{\beta D^2}{R} + \sqrt[3]{\frac{\beta \zeta^2 D^4}{R^2}} + \sqrt{1 - \frac{S}{N}} \frac{\zeta D}{\sqrt{SR}}$ |
| FedAvg (Woodworth et al., 2020a) | $\sqrt[3]{\frac{\beta \zeta^2 D^4}{R^2}}$ |
| SCAFFOLD♦ | $\sqrt{\frac{N}{S}} \frac{\beta D^2}{R}$ ♣ |
| *This paper* | |
| FedAvg → SGD (Thm. 4.1) | $\min\{\frac{\beta D^2}{R}, \frac{\sqrt{\beta \zeta D^3}}{\sqrt{R}}\} + \sqrt[4]{1 - \frac{S}{N}} \frac{\sqrt{\beta \zeta D^3}}{\sqrt[4]{SR}}$ |
| FedAvg → ASG (Thm. 4.2) | $\min\{\frac{\beta D^2}{R^2}, \frac{\sqrt{\beta \zeta D^3}}{R}\} + \sqrt[4]{1 - \frac{S}{N}} \frac{\sqrt{\beta \zeta D^3}}{\sqrt[4]{SR}} + \sqrt{1 - \frac{S}{N}} \frac{\zeta D}{\sqrt{SR}}$ |
| Algo.-independent LB (Thm. 5.4) | $\min\{\frac{\beta D^2}{R^2}, \frac{\zeta D}{\sqrt{R^5}}\}$ |

For the formal statement, see Thm. F.1.

**Theorem 4.2 (FedAvg → ASG).** *Under the hypotheses of Thm. 4.1 with a choice of $\mathcal{A}_{global}$ as ASG (Algo. 3) and the parameter choices of Thm. D.3, we get the following guarantees:*

- *Strongly convex: If $F_i$'s satisfy Assumption B.1 for $\mu > 0$ then there exists a finite $K$ above which we get the rate, $\mathbb{E}F(\hat{x}_2) - F(x^*) \leq \tilde{\mathcal{O}}(\min\{\Delta, \zeta^2/\mu\} \exp(-R/\sqrt{\kappa}) + (1 - S/N)\zeta^2/\mu SR)$.*

- *General convex: If $F_i$'s satisfy Assumption B.2 then there exists a finite $K$ above which we get the rate, $\mathbb{E}F(\hat{x}_2) - F(x^*) \leq \tilde{\mathcal{O}}(\min\{\beta D^2/R^2, \sqrt{\beta \zeta D^3}/R\} + \sqrt{1 - S/N}(\zeta D/\sqrt{SR}) + \sqrt[4]{1 - S/N}(\sqrt{\beta \zeta D^3}/\sqrt[4]{SR}))$.*

For the formal statement, see Thm. F.2. We show and discuss the near-optimality of FedAvg → ASG under strongly convex and PL conditions (and under full participation) in Section 5, where we introduce matching lower bounds. Under strong-convexity shown in Table 1, FedAvg → ASG, when compared to ASG, converts the $\Delta \exp(-R/\sqrt{\kappa})$ term into a $\min\{\Delta, \zeta^2/\mu\} \exp(-R/\sqrt{\kappa})$ term, improving over ASG when heterogeneity moderately small: $\zeta^2/\mu < \Delta$. It also significantly improves over FedAvg, as $\min\{\Delta, \zeta^2/\mu\} \exp(-R/\sqrt{\kappa})$ is exponentially faster than $\kappa(\zeta^2/\mu)R^{-2}$. Under the PL condition (which is not necessarily convex) the story is similar (Table 4), except we use FedAvg → SGD as our representative algorithm, as Nesterov acceleration is not known to improve under non-convex settings.

In the general convex case (Table 2), let $\beta = D = 1$ for the purpose of comparisons. Then FedAvg → ASG's convergence rate is $\min\{1/R^2, \zeta^{1/2}/R\} + \sqrt{1 - S/N}(\zeta/\sqrt{SR}) + \sqrt[4]{1 - S/N}\sqrt{\zeta}/\sqrt[4]{SR}$. If $\zeta < \frac{1}{R^2}$, then $\zeta^{1/2}/R < 1/R^2$ and if $\zeta < \sqrt{S/R^7}$, then $\sqrt[4]{1 - S/N}\sqrt{\zeta}/\sqrt[4]{SR} < 1/R^2$, so the FedAvg → ASG convergence rate is better than the convergence rate of ASG under the regime $\zeta < \min\{1/R^2, \sqrt{S/R^7}\}$. The rate of Karimireddy et al. (2020b) for FedAvg (which does not require $S = N$) is strictly worse than ASG, and so has a worse convergence rate than FedAvg → ASG if $\zeta < \min\{1/R^2, \sqrt{S/R^7}\}$. Altogether, if $\zeta < \min\{1/R^2, \sqrt{S/R^7}\}$, FedAvg → ASG achieves the best known worst-case rate. Finally, in the $S = N$ case, FedAvg → ASG does not have a regime in $\zeta$ where it improves over both ASG and FedAvg (the analysis of Woodworth et al. (2020a), which requires $S = N$) at the same time. It is unclear if this is due to looseness in the analysis.

Next, we analyze variance reduced methods that improve convergence when a random subset of the clients participate in each round (i.e., partial participation).

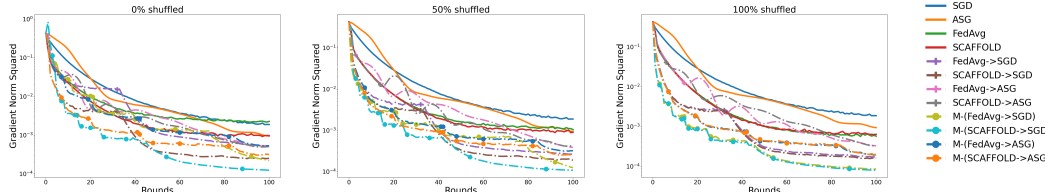

Figure 2: Plot titles denote data homogeneity (§ 6). "X→Y" denotes a FedChain instantiation with X as $\mathcal{A}_{\text{local}}$ and Y as $\mathcal{A}_{\text{global}}$, circle markers denote stepsize decay events and plusses denote switching from $\mathcal{A}_{\text{local}}$ to $\mathcal{A}_{\text{global}}$. Across all heterogeneity levels, the multistage algorithms perform the best. Stepsize decayed baselines are left out to simplify the plots; we display them in App. J.2.

**Theorem 4.3** (**FedAvg → SAGA**). *Suppose that client objectives $F_i$'s and their gradient queries satisfy Assumptions B.4, B.5, B.6, B.7 and B.8. Then running FedChain (Algo. 1) with $\mathcal{A}_{local}$ as FedAvg (Algo. 4 ) with the parameter choices of Thm. E.1, and $\mathcal{A}_{global}$ as SAGA (Algo. 5) with the parameter choices of Thm. D.4, we get the following guarantees as long as $R \geq \Omega(\frac{N}{S})$:*

- ***Strongly convex:*** *If $F_i$'s satisfy Assumption B.1 for $\mu > 0$, then there exists a finite $K$ above which we get the rate $\mathbb{E}F(\hat{x}_2) - F(x^*) \leq \tilde{\mathcal{O}}(\min\{\Delta, \zeta^2/\mu\} \exp(-\min\{S/N, 1/\kappa\}R))$.*

- ***PL condition:*** *If $F_i$'s satisfy Assumption B.3 for $\mu > 0$, then there exists a finite $K$ above which we get the rate $\mathbb{E}F(\hat{x}_2) - F(x^*) \leq \tilde{\mathcal{O}}(\min\{\Delta, \zeta^2/\mu\} \exp(-(S/N)^{\frac{2}{3}} R/\kappa))$.*

For the formal statement, see Thm. F.3.

**Theorem 4.4** (**FedAvg → SSNM**). *Suppose that client objectives $F_i$'s and their gradient queries satisfy Assumptions B.4, B.5, B.6 and B.8. Then running FedChain (Algo. 1) with $\mathcal{A}_{local}$ as FedAvg (Algo. 4) with the parameter choices of Thm. E.1, and $\mathcal{A}_{global}$ as SSNM (Algo. 6) with the parameter choices of Thm. D.5, we get the following guarantees as long as $R \geq \Omega(\frac{N}{S})$:*

- ***Strongly convex:*** *If $F_i$'s satisfy Assumption B.1 for $\mu > 0$, then there exists a finite $K$ (same as in FedAvg → SGD) above which we get the rate $\mathbb{E}F(\hat{x}_2) - F(x^*) \leq \tilde{\mathcal{O}}(\min\{\Delta, \zeta^2/\mu\} \exp(-\min\{S/N, \sqrt{S/(N\kappa)}\}R))$.*

For the formal statement, see Thm. F.4. SSNM (Zhou et al., 2019) is the Nesterov accelerated version of SAGA (Defazio et al., 2014).

The main contribution of using variance reduced methods in $\mathcal{A}_{\text{global}}$ is the removal of the sampling error (the terms depending on the sampling heterogeneity error $(1 - S/N)(\zeta^2/S)$) from the convergence rates, in exchange for requiring a round complexity of at least $N/S$. To illustrate this, observe that in the strongly convex case, FedAvg → SGD has convergence rate $\min\{\Delta, \zeta^2/\mu\} \exp(-R/\kappa) + (1-S/N)(\zeta^2/SR)$ and FedAvg → SAGA (FedAvg → SGD's variance-reduced counterpart) has convergence rate $\min\{\Delta, \zeta^2/\mu\} \exp(-\min\{1/\kappa, S/N\}R)$, dropping the $(1 - S/N)(\zeta^2/\mu SR)$ sampling heterogeneity error term in exchange for harming the rate of linear convergence from $1/\kappa$ to $\min\{1/\kappa, S/N\}$.

This same tradeoff occurs in finite sum optimization (the problem $\min_x (1/n) \sum_{i=1}^{n} \psi_i(x)$, where $\psi_i$'s (typically) represent losses on data points and the main concern is computation cost), which is what variance reduction is designed for. In finite sum optimization, variance reduction methods such as SVRG (Johnson & Zhang, 2013) and SAGA (Defazio et al., 2014; Reddi et al., 2016) achieve linear rates of convergence (given strong convexity of $\psi_i$'s) in exchange for requiring at least $N/S$ updates. Because we can treat FL as an instance of finite-sum optimization (by viewing Eq. (1) as a finite sum of objectives and $\zeta^2$ as the variance between $F_i$'s), these results from variance-reduced finite sum optimization can be extended to federated learning. This is the idea behind SCAFFOLD (Karimireddy et al., 2020b).

It is not always the case that variance reduction in $\mathcal{A}_{\text{global}}$ achieves better rates. In the strongly convex case if $(1 - S/N)(\zeta^2/\mu S\epsilon) > N/S$, then variance reduction gets gain, otherwise not.

## 5    LOWER BOUNDS

Our lower bound allows full participation. It assumes deterministic gradients and the following class from (Woodworth et al., 2020a; Woodworth, 2021; Carmon et al., 2020):

**Definition 5.1.** For a $v \in \mathbb{R}^d$, let $\text{supp}(v) = \{i \in [d] : v_i \neq 0\}$. An algorithm is *distributed zero-respecting* if for any $i, k, r$, the $k$-th iterate on the $i$-th client in the $r$-th round $x_{i,k}^{(r)}$ satisfy

$$\text{supp}(x_{i,k}^{(r)}) \subseteq \bigcup_{0 \leq k' < k} \text{supp}(\nabla F_i(x_{i,k'}^{(r)})) \bigcup_{i' \in [N], 0 \leq k' \leq K-1, 0 \leq r' < r} \text{supp}(\nabla F_{i'}(x_{i',k'}^{(r')})) \quad (3)$$

Distributed zero-respecting algorithms' iterates have components in coordinates that they have any information on. As discussed in (Woodworth et al., 2020a), this means that algorithms which are *not* distributed zero-respecting are just "wild guessing". Algorithms that are distributed zero-respecting include SGD, ASG, and FedAvg. We assume the following class of algorithms in order to bound the heterogeneity of our construction for the lower bound proof.

**Definition 5.2.** We say that an algorithm is *distributed distance-conserving* if for any $i, k, r$, we have for the $k$-th iterate on the $i$-th client in the $r$-th round $x_{i,k}^{(r)}$ satisfies $\|x_{i,k}^{(r)} - x^*\|^2 \leq (c/2)[\|x_{\text{init}} - x^*\|^2 + \sum_{i=1}^{N} \|x_{\text{init}} - x_i^*\|^2]$, where $x_j^* := \arg\min_x F_j(x)$ and $x^* := \arg\min_x F(x)$ and $x_{\text{init}}$ is the initial iterate, and $c$ is some scalar parameter.

Algorithms which do not satisfy Definition 5.2 with $c$ at most logarithmic in problem parameters (see § 2) are those that move substantially far away from $x^*$, even farther than the $x_i^*$'s are from $x^*$. With this definition in mind, we slightly overload the usual definition of heterogeneity for the lower bound:

**Definition 5.3.** A distributed optimization problem is $(\zeta, c)$-*heterogeneous* if $\max_{i \in [N]} \sup_{x \in A} \|\nabla F_i(x) - \nabla F(x)\|^2 \leq \zeta^2$, where we define $A := \{x : \|x - x^*\|^2 \leq (c/2)(\|x_{\text{init}} - x^*\|^2 + \sum_{i=1}^{N} \|x_{\text{init}} - x_i^*\|^2)\}$ for some scalar parameter $c$.

Those achievable convergence rates in FL that assume Eq. (2) can be readily extended to account for Definition 5.3 as long as the algorithm satisfies Definition 5.2. We show that the chaining algorithms we propose satisfy Definition 5.3 in Thm. D.1, Thm. D.3, and Thm. E.1 for $c$ at most polylogarithmic in the problem parameters defined in § 2. [3]. Other FL algorithms also satisfy Definition 5.2, notably FedAvg analyzed in Woodworth et al. (2020a) for $c$ an absolute constant, which is the current tightest known analysis of FedAvg in the full participation setting.

**Theorem 5.4.** *For any number of rounds $R$, number of local steps per-round $K$, and $(\zeta, c)$-heterogeneity (Definition 5.3), there exists a global objective $F$ which is the average of two $\beta$-smooth (Assumption B.4) and $\mu(\geq 0)$-strongly convex (Assumption B.1) quadratic client objectives $F_1$ and $F_2$ with an initial sub-optimality gap of $\Delta$, such that the output $\hat{x}$ of any distributed zero-respecting (Definition 5.1) and distance-conserving algorithm (Definition 5.2) satisfies*

- ***Strongly convex:*** *$F(\hat{x}) - F(x^*) \geq \Omega(\min\{\Delta, 1/(c\kappa^{3/2})(\zeta^2/\beta)\} \exp(-R/\sqrt{\kappa}).)$ when $\mu > 0$, and*

- ***General Convex*** *$F(\hat{x}) - F(x^*) \geq \Omega(\min\{\beta D^2/R^2, \zeta D/(c^{1/2}\sqrt{R^5})\})$ when $\mu = 0$.*

**Corollary 5.5.** *Under the hypotheses of Theorem 5.4, there exists a global objective $F$ which is $\mu$-PL and satisfies $F(\hat{x}) - F(x^*) \geq \Omega(\min\{\Delta, 1/(c\kappa^{3/2})(\zeta^2/\beta)\} \exp(-R/\sqrt{\kappa}))$.*

A proof of the lower bound is in App. G, and the corollary follows immediately from the fact that $\mu$-strong convexity implies $\mu$-PL. This result tightens the lower bound in Woodworth et al. (2020a), which proves a similar lower bound but in terms of a much larger class of functions with heterogeneity bounded in $\zeta_* = (1/N) \sum_{i=1}^{N} \|\nabla F_i(x^*)\|^2$. To the best of our knowledge, all achievable rates that can take advantage of heterogeneity require $(\zeta, c)$-heterogeneity (which is a smaller class than $\zeta_*$-heterogeneity), which are incomparable with the existing lower bound from (Woodworth et al., 2020a) that requires $\zeta_*$-heterogeneity. By introducing the class of distributed distance-conserving algorithms

---

[3]We do not formally show it for SAGA and SSNM, as the algorithms are functionally the same as SGD and ASG under full participation.

Table 3: Test accuracies (↑). "Constant" means the stepsize does not change during optimization, while "w/ Decay" means the stepsize decays during optimization. **Left:** Comparison among algorithms in the EMNIST task. **Right:** Comparison among algorithms in the CIFAR-100 task. "SCA." abbreviates SCAFFOLD.

| Algorithm | Constant | w/ Decay | Algorithm | Constant | w/ Decay |
|---|---|---|---|---|---|
| *Baselines* | | | *Baselines* | | |
| SGD | 0.7842 | 0.7998 | SGD | 0.1987 | 0.1968 |
| FedAvg | 0.8314 | 0.8224 | FedAvg | 0.4944 | 0.5059 |
| SCAFFOLD | 0.8157 | 0.8174 | SCAFFOLD | – | – |
| *FedChain* | | | *FedChain* | | |
| FedAvg → SGD | 0.8501 | 0.8355 | FedAvg → SGD | **0.5134** | **0.5167** |
| SCA. → SGD | **0.8508** | **0.8392** | SCA. → SGD | – | – |

(which includes most of the algorithms we are interested in), Thm. 5.4 allows us, for the first time, to identify the optimality of the achievable rates as shown in Tables 1, 2, and 4.

Note that this lower bound allows full participation and should be compared to the achievable rates with $S = N$; comparisons with variance reduced methods like FedAvg → SAGA and FedAvg → SSNM are unnecessary. Our lower bound proves that FedAvg → ASG is optimal up to condition number factors shrinking exponentially in $\sqrt{\kappa}$ among algorithms satisfying Definition 5.2 and Definition 5.3. Under the PL-condition, the situation is similar, except FedAvg → SGD loses $\sqrt{\kappa}$ in the exponential versus the lower bound. On the other hand, there remains a substantial gap between FedAvg → ASG and the lower bound in the general convex case. Meaningful progress in closing the gap in the general convex case is an important future direction.

## 6 EXPERIMENTS

We evaluate the utility of our framework on strongly convex and nonconvex settings. We compare the communication round complexities of four baselines: FedAvg, SGD, ASG, and SCAFFOLD, the stepsize decaying variants of these algorithms (which are prefixed by M- in plots) and the various instantiations of FedChain (Algo. 1).

**Convex Optimization (Logistic Regression)** We first study federated regularized logistic regression, which is strongly convex (objective function in App. I.1). In this experiment, we use the MNIST dataset of handwritten digits (LeCun et al., 2010). We (roughly) control client heterogeneity by creating "clients" through shuffling samples across different digit classes. The details of this shuffling process are described in App. I.1; if a dataset is more shuffled (more homogeneous), it roughly corresponds to a lower heterogeneity dataset. All clients participate in each round.

Fig. 2 compares the convergence of chained and non-chained algorithms in the stochastic gradient setting (minibatches are 1% of a client's data), over $R = 100$ rounds (tuning details in App. I.1). We observe that in all heterogeneity levels, FedChain instantiations (whether two or more stages) outperform other baselines. We also observe that SCAFFOLD→SGD outperforms all other curves, including SCAFFOLD→ASG. This can be explained by the fact that acceleration increases the effect of noise, which can contribute to error if one does not take $K$ large enough (we set $K = 20$ in the convex experiments). The effect of large $K$ is elaborated upon in App. J.1.

**Nonconvex Optimization (Neural Networks)** We also evaluated nonconvex image classification tasks with convolutional neural networks. In all experiments, we consider client sampling with $S = 10$. We started with digit classification over EMNIST (Cohen et al., 2017) (tuning details in App. I.2.1), where handwritten characters are partitioned by author. Table 3 (**Left**) displays test accuracies on the task. Overall, FedChain instantiations perform better than baselines.

We also considered image classification with ResNet-18 on CIFAR-100 (Krizhevsky, 2009) (tuning details in App. I.2.2). Table 3 (**Right**) displays the test accuracies on CIFAR-100; SCAFFOLD is not included due to memory constraints. FedChain instantiations again perform the best.

ACKNOWLEDGMENTS

This work is supported by Google faculty research award, JP Morgan Chase, Siemens, the Sloan Foundation, Intel, NSF grants CNS-2002664, CA-2040675, IIS-1929955, DMS-2134012, CCF-2019844 as a part of NSF Institute for Foundations of Machine Learning (IFML), and CNS-2112471 as a part of NSF AI Institute for Future Edge Networks and Distributed Intelligence (AI-EDGE). Most of this work was done prior to the second author joining Amazon, and it does not relate to his current position there.

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

# Appendices

Table 4: Rates under the PL condition. ♣ Rate requires $R \geq \frac{N}{S}$.

| Method/Analysis | $\mathbb{E}F(\hat{x}) - F(x^*) \leq \tilde{\mathcal{O}}(\cdot)$ |
|---|---|
| *Centralized Algorithms* | |
| SGD | $\Delta \exp(-\kappa^{-1}R) + (1 - \frac{S}{N})\frac{\kappa\zeta^2}{\mu SR}$ |
| *Federated Algorithms* | |
| FedAvg (Karimireddy et al., 2020a) | $\kappa\Delta \exp(-\kappa^{-1}R) + \frac{\kappa^2\zeta^2}{\mu R^2}$ |
| *This paper* | |
| FedAvg → SGD (Thm. 4.1) | $\min\{\Delta, \frac{\zeta^2}{\mu}\}\exp(-\kappa^{-1}R) + (1 - \frac{S}{N})\frac{\kappa\zeta^2}{\mu SR}$ |
| FedAvg → SAGA (Thm. 4.3) | $\min\{\Delta, \frac{\zeta^2}{\mu}\}\exp(-((\frac{N}{S})^{\frac{2}{3}}\kappa)^{-1}R)$♣ |
| Algo.-independent LB (Corollary 5.5) | $\min\{\Delta, \kappa^{-\frac{3}{2}}(\frac{\zeta^2}{\beta})\}\exp(-\kappa^{-\frac{1}{2}}R)$ |

# A ADDITIONAL RELATED WORK

**Multistage algorithms.** Our chaining framework has similarities in analysis to multistage algorithms which have been employed in the classical convex optimization setting (Aybat et al., 2019; Fallah et al., 2020; Ghadimi & Lan, 2012; Woodworth et al., 2020a). The idea behind multistage algorithms is to manage stepsize decay to balance fast progress in "bias" error terms while controlling variance error. However chaining differs from multistage algorithms in a few ways: (1) We do not necessarily decay stepsize (2) In the heterogeneous FL setting, we have to accomodate error due to client drift (Reddi et al., 2020; Karimireddy et al., 2020b), which is not analyzed in prior multistage literature. (3) Our goal is to minimize commuincation rounds rather than iteration complexity.

# B DEFINITIONS

**Assumption B.1.** $F_i$'s are $\mu$-strongly convex for $\mu > 0$, i.e.

$$\langle \nabla F_i(x), y - x \rangle \leq -(F_i(x) - F_i(y) + \frac{\mu}{2}\|x - y\|^2) \tag{4}$$

for any $i, x, y$.

**Assumption B.2.** $F_i$'s are general convex, i.e.

$$\langle \nabla F_i(x), y - x \rangle \leq -(F_i(x) - F_i(y)) \tag{5}$$

for any $i, x, y$.

**Assumption B.3.** $F$ is $\mu$-PL for $\mu > 0$, i.e.

$$2\mu(F(x) - F(x^*)) \leq \|\nabla F(x)\|^2 \tag{6}$$

for any $x$.

**Assumption B.4.** $F_i$'s are $\beta$-smooth where

$$\|\nabla F_i(x) - \nabla F_i(y)\| \leq \beta\|x - y\| \tag{7}$$

for any $i, x, y$.

**Assumption B.5.** $F_i$'s are $\zeta^2$-heterogeneous, defined as

$$\zeta^2 := \sup_x \max_{i \in [N]} \|\nabla F(x) - \nabla F_i(x)\|^2 \tag{8}$$

**Assumption B.6.** Gradient queries within a client have a uniform upper bound on variance and are also unbiased.

$$\mathbb{E}_{z_i \sim \mathcal{D}_i}\|\nabla f(x; z_i) - \nabla F_i(x)\|^2 \leq \sigma^2, \quad \mathbb{E}_{z_i \sim \mathcal{D}_i}[\nabla f(x; z_i)] = \nabla F_i(x) \tag{9}$$

**Assumption B.7.** Cost queries within a client have a uniform upper bound on variance and are also unbiased.

$$\mathbb{E}_{z_i \sim \mathcal{D}_i}(f(x; z_i) - F_i(x))^2 \leq \sigma_F^2, \quad \mathbb{E}_{z_i \sim \mathcal{D}_i}f(x; z_i) = F_i(x) \tag{10}$$

**Assumption B.8.** Cost queries within a client have an absolute deviation.

$$\zeta_F^2 = \sup_x \max_{i \in [N]} (F(x) - F_i(x))^2 \tag{11}$$

**Assumption B.9.** The initial suboptimality gap is upper bounded by $\Delta$, where $x_0$ is the initial feasible iterate and $x^*$ is the global minimizer.

$$\mathbb{E}F(x_0) - F(x^*) \leq \Delta \tag{12}$$

**Assumption B.10.** The expected initial distance from the optimum is upper bounded by $D$, where $x_0$ is the initial feasible iterate and $x^*$ is one of the global minimizers:

$$\mathbb{E}\|x_0 - x^*\|^2 \leq D^2 \tag{13}$$

## C  OMITTED ALGORITHM DEFINITIONS

---
**Algorithm 2** SGD

---
**Input:** initial point $x^{(0)}$, stepsize $\eta$, loss $f$
**for** $r = 0, \ldots, R - 1$ **do**
   ▷ **Run a step of SGD**
   Sample $S$ clients $\mathcal{S}_r \subseteq [1, \ldots, N]$
   **Send** $x^{(r)}$ to all clients in $\mathcal{S}_r$
   **Receive** $\{g_i^{(r)}\}_{i \in \mathcal{S}_r} = \mathrm{Grad}(x^{(r)}, \mathcal{S}_r, z^{(r)})$ (Algo. 7)
   $g^{(r)} = \frac{1}{S} \sum_{i \in \mathcal{S}_r} g_i^{(r)}$
   $x^{(r+1)} = x^{(r)} - \eta \cdot g^{(r)}$
**end for**

---

---
**Algorithm 3** ASG

---
**Input:** initial point $x^{(0)} = x_{\mathrm{ag}}^{(0)}$, $\{\alpha_r\}_{1 \leq r \leq R}$, $\{\gamma_r\}_{1 \leq r \leq R}$, loss $f$
**for** $r = 1, \ldots, R$ **do**
   ▷ **Run a step of ASG**
   Sample $S$ clients $\mathcal{S}_r \subseteq [1, \ldots, N]$
   **Send** $x^{(r)}$ to all clients in $\mathcal{S}_r$
   $x_{\mathrm{md}}^{(r)} = \frac{(1-\alpha_r)(\mu+\gamma_r)}{\gamma_r+(1-\alpha_r^2)\mu} x_{\mathrm{ag}}^{(r-1)} + \frac{\alpha_r[(1-\alpha_r)\mu+\gamma_r]}{\gamma_r+(1-\alpha_r^2)\mu} x^{(r-1)}$
   **Receive** $\{g_i^{(r)}\}_{i \in \mathcal{S}_r} = \mathrm{Grad}(x_{\mathrm{md}}^{(r)}, \mathcal{S}_r, z^{(r)})$ (Algo. 7)
   $g^{(r)} = \frac{1}{S} \sum_{i \in \mathcal{S}_r} g_i^{(r)}$
   $x^{(r)} = \arg\min_x \{\alpha_r[\langle g^{(r)}, x \rangle + \frac{\mu}{2}\|x_{\mathrm{md}}^{(r)} - x\|^2] + [(1-\alpha_r)\frac{\mu}{2} + \frac{\gamma_r}{2}\|x^{(r-1)} - x\|^2]\}$
   $x_{\mathrm{ag}}^{(r)} = \alpha_r x^{(r)} + (1 - \alpha_r)x_{\mathrm{ag}}^{(r-1)}$
**end for**

---

---

**Algorithm 4** FedAvg

---

**Input:** Stepsize $\eta$, initial point $x^{(0)}$
**for** $r = 0, \ldots, R - 1$ **do**
    Sample $S$ clients $\mathcal{S}_r \subseteq [1, \ldots, N]$
    **Send** $x^{(r)}$ to all clients in $\mathcal{S}_r$, all clients set $x_{i,0}^{(r)} = x^{(r)}$
    **for** client $i \in \mathcal{S}_r$ *in parallel* **do**
        **for** $k = 0, \ldots, \sqrt{K} - 1$ **do**
            Client samples $g_{i,k}^{(r)} = \frac{1}{\sqrt{K}} \sum_{k'=0}^{\sqrt{K}-1} \nabla f(x_{i,k}^{(r)}; z_{i,k\sqrt{K}+k'}^{(r)})$ where $z_{i,k\sqrt{K}+k'}^{(r)} \sim \mathcal{D}_i$
            $x_{i,k+1}^{(r)} = x_{i,k}^{(r)} - \eta \cdot g_{i,k}^{(r)}$
        **end for**
        **Receive** $g_i^{(r)} = \sum_{k=0}^{\sqrt{K}-1} g_{i,k}^{(r)}$ from client
    **end for**
    $g^{(r)} = \frac{1}{S} \sum_{i \in \mathcal{S}_r} g_i^{(r)}$
    $x^{(r+1)} \leftarrow x^{(r)} - \eta \cdot g^{(r)}$
**end for**

---

---

**Algorithm 5** SAGA

---

**Input:** stepsize $\eta$, loss $f$, initial point $x^{(0)}$
**Send** $\phi_i^{(0)} = x^{(0)}$ to all clients $i$
**Receive** $c_i^{(0)} = \frac{1}{K} \sum_{k=0}^{K-1} \nabla f(\phi_i^{(0)}; z_{i,k}^{(-1)})$, $z_{i,k}^{(-1)} \sim \mathcal{D}_i$ from all clients, $c^{(0)} = \frac{1}{N} \sum_{i=1}^{N} c_i^{(0)}$
**for** $r = 0, \ldots, R - 1$ **do**
    Sample $S$ clients $\mathcal{S}_r \subseteq [1, \ldots, N]$
    **Send** $x^{(r)}$ to all clients in $\mathcal{S}_r$, **Receive** $\{g_i^{(r)}\}_{i \in \mathcal{S}_r} = \mathrm{Grad}(x^{(r)}, \mathcal{S}_r, z^{(r)})$ (Algo. 7)
    $g^{(r)} = \frac{1}{S} \sum_{i \in \mathcal{S}_r} g_i^{(r)} - \frac{1}{S} \sum_{i \in \mathcal{S}_r} c_i^{(r)} + c^{(r)}$
    $x^{(r+1)} = x^{(r)} - \eta \cdot g^{(r)}$
    **Option I:** Set $c_i^{(r+1)} = g_i^{(r)}$, $\phi_i^{(r+1)} = x^{(r)}$ for $i \in \mathcal{S}_r$
    **Option II:** New *independent* sample of clients $\mathcal{S}_r'$, **send** $x^{(r)}$ to all clients in $\mathcal{S}_r'$, **receive**
    $\{\tilde{g}_i^{(r)}\}_{i \in \mathcal{S}_r'} = \mathrm{Grad}(x^{(r)}, \mathcal{S}_r', \tilde{z}^{(r)})$, set $c_i^{(r+1)} = \tilde{g}_i^{(r)}$, $\phi_i^{(r+1)} = x^{(r)}$ for $i \in \mathcal{S}_r'$

    $c_i^{(r+1)} = c_i^{(r)}$ and $\phi_i^{(r+1)} = \phi_i^{(r)}$ for any $i$ not yet updated
    $c^{(r+1)} = \frac{1}{N} \sum_{i=1}^{N} c_i^{(r+1)}$
**end for**

---

---

**Algorithm 6** SSNM

---

**Input:** stepsize $\eta$, momentum $\tau$, client losses are $F_i(x) = \mathbb{E}_{z_i \sim \mathcal{D}_i}[f(x; z_i)] + h(x) = \tilde{F}_i(x) + h(x)$ where $\tilde{F}_i$ can be general convex and $h(x)$ is $\mu$-strongly convex, initial point $x^{(0)}$

**Send** $\phi_i^{(0)} = x^{(0)}$ to all clients $i$

**Receive** $c_i^{(0)} = \frac{1}{K} \sum_{k=0}^{K-1} \nabla f(\phi_i^{(0)}; z_{i,k}^{(-1)})$, $z_{i,k}^{(-1)} \sim \mathcal{D}_i$ from all clients $c^{(0)} = \frac{1}{N} \sum_{i=1}^{N} c_i^{(0)}$

**for** $r = 0, \ldots, R-1$ **do**

    Sample $S$ clients $\mathcal{S}_r \subseteq [1, \ldots, N]$

    **Send** $x^{(r)}$ to all clients in $\mathcal{S}_r$, $y_{i_r}^{(r)} = \tau x^{(r)} + (1-\tau)\phi_{i_r}^{(r)}$ for $i_r \in \mathcal{S}_r$

    **Receive** $g_{i_r}^{(r)} = \frac{1}{K} \sum_{k=0}^{K-1} \nabla f(y_{i_r}^{(r)}; z_{i_r,k}^{(r)})$ for $i_r \in \mathcal{S}_r$, $z_{i_r,k}^{(r)} \sim \mathcal{D}_i$

    $g^{(r)} = \frac{1}{S} \sum_{i_r \in \mathcal{S}_r} g_{i_r}^{(r)} - \frac{1}{S} \sum_{i_r \in \mathcal{S}_r} c_{i_r}^{(r)} + c^{(r)}$

    $x^{(r+1)} = \arg\min_x \{ h(x) + \langle g^{(r)}, x \rangle + \frac{1}{2\eta} \|x^{(r)} - x\|^2 \}$

    Sample new *independent* sample of $S$ clients $\mathcal{S}_r' \subseteq [1, \ldots, N]$

    **Send** $x^{(r)}$ to all clients in $\mathcal{S}_r'$, $\phi_{I_r}^{(r+1)} = \tau x^{(r+1)} + (1-\tau)\phi_{I_r}^{(r)}$ for $I_r \in \mathcal{S}_r'$

    **Receive** $g_{I_r}^{(r)} = \frac{1}{K} \sum_{k=0}^{K-1} \nabla f(\phi_{I_r}^{(r+1)}; \tilde{z}_{I_r,k}^{(r)})$ for $I_r \in \mathcal{S}_r'$, $\tilde{z}_{I_r,k}^{(r)} \sim \mathcal{D}_i$

    $c_i^{(r+1)} = g_{I_r}^{(r)}$ for $I_r \in \mathcal{S}_r'$

    $c_i^{(r+1)} = c_i^{(r)}$ and $\phi_i^{(r+1)} = \phi_i^{(r)}$ for any $i$ not yet updated

    $c^{(r+1)} = \frac{1}{N} \sum_{i=1}^{N} c_i^{(r+1)}$

**end for**

---

**Algorithm 7** Grad$(x, \mathcal{S}, z)$

---

**for** client $i \in \mathcal{S}$ *in parallel* **do**

    Client samples $g_{i,k}^{(r)} = \nabla f(x^{(r)}; z_{i,k}^{(r)})$ where $z_{i,k}^{(r)} \sim \mathcal{D}_i$, $k \in \{0, \ldots, K-1\}$

    $g_i^{(r)} = \frac{1}{K} \sum_{k=0}^{K-1} g_{i,k}^{(r)}$

**end for**

**return** $\{g_i^{(r)}\}_{i \in \mathcal{S}_r}$

---

# D PROOFS FOR GLOBAL UPDATE METHODS

## D.1 SGD

**Theorem D.1.** *Suppose that client objectives $F_i$'s and their gradient queries satisfy Assumption B.4 and Assumption B.6. Then running Algo. 2 gives the following for returned iterate $\hat{x}$:*

- ***Strongly convex:*** *$F_i$'s satisfy Assumption B.1 for $\mu > 0$. If we return $\hat{x} = \frac{1}{W_R} \sum_{r=0}^{R} w_r x^{(r)}$ with $w_r = (1 - \eta\mu)^{1-(r+1)}$ and $W_R = \sum_{r=0}^{R} w_r$, $\eta = \tilde{\mathcal{O}}(\min\{\frac{1}{\beta}, \frac{1}{\mu R}\})$, and $R > \kappa$,*

$$\mathbb{E}F(\hat{x}) - F(x^*) \leq \tilde{\mathcal{O}}(\Delta \exp(-\frac{R}{\kappa}) + \frac{\sigma^2}{\mu SKR} + (1 - \frac{S}{N})\frac{\zeta^2}{\mu SR})$$

$$\mathbb{E}\|x^{(r)} - x^*\|^2 \leq \tilde{\mathcal{O}}(D^2) \text{ for any } 0 \leq r \leq R$$

- ***General convex (grad norm):*** *$F_i$'s satisfy Assumption B.2. If we return the average iterate $\hat{x} = \frac{1}{R} \sum_{r=1}^{R} x^{(r)}$, and $\eta = \min\{\frac{1}{\beta}, (\frac{\Delta}{\beta c R})^{1/2}\}$ where $c$ is the update variance*

$$\mathbb{E}\|\nabla F(\hat{x})\|^2 \leq \tilde{\mathcal{O}}(\frac{\beta\Delta}{R} + \frac{\beta\sigma D}{\sqrt{SKR}} + \sqrt{1 - \frac{S}{N}}\frac{\beta\zeta D}{\sqrt{SR}})$$

$$\mathbb{E}\|x^{(r)} - x^*\|^2 \leq \mathcal{O}(D^2) \text{ for any } 0 \leq r \leq R$$

- **PL condition:** $F_i$'s satisfy Assumption B.3. If we return the final iterate $\hat{x} = x^{(R)}$, set $\eta = \tilde{\mathcal{O}}(\min\{\frac{1}{\beta}, \frac{1}{\mu R}\})$, then

$$\mathbb{E}F(x^{(R)}) - F(x^*) \leq \tilde{\mathcal{O}}(\Delta \exp(-\frac{R}{\kappa}) + \frac{\kappa\sigma^2}{\mu SKR} + (1 - \frac{S}{N})\frac{\kappa\zeta^2}{\mu SR})$$

**Lemma D.2** (Update variance). *For SGD, we have that*

$$\mathbb{E}_r\|g^{(r)}\|^2 \leq \|\nabla F(x^{(r)})\|^2 + \frac{\sigma^2}{SK} + (1 - \frac{S-1}{N-1})\frac{\zeta^2}{S} \tag{14}$$

*Proof.* Observing that

$$\mathbb{E}_r\|g^{(r)}\|^2 = \mathbb{E}_r\|g^{(r)} - \nabla F(x^{(r)})\|^2 + \|\nabla F(x^{(r)})\|^2 \tag{15}$$

and using Lemma H.1, we have

$$\mathbb{E}_r\|g^{(r)} - \nabla F(x^{(r)})\|^2 \leq \frac{\sigma^2}{SK} + (1 - \frac{S-1}{N-1})\frac{\zeta^2}{S} \tag{16}$$

$\square$

### D.1.1   CONVERGENCE OF SGD FOR STRONGLY CONVEX FUNCTIONS

*Proof.* Using the update of SGD,

$$\|x^{(r+1)} - x^*\|^2 \leq \|x^{(r)} - x^*\|^2 - 2\eta\langle g^{(r)}, x^{(r)} - x^*\rangle + \eta^2\|g^{(r)}\|^2 \tag{17}$$

Taking expectation up to the r-th round

$$\mathbb{E}_r\|x^{(r+1)} - x^*\|^2 \leq \|x^{(r)} - x^*\|^2 - 2\eta\langle\nabla F(x^{(r)}), x^{(r)} - x^*\rangle + \eta^2\mathbb{E}_r\|g^{(r)}\|^2 \tag{18}$$

By Assumption B.1,

$$-2\eta\langle\nabla F(x^{(r)}), x^{(r)} - x^*\rangle \leq -2\eta(F(x^{(r)}) - F(x^*)) - \eta\mu\|x^{(r)} - x^*\|^2 \tag{19}$$

Using Lemma D.2, Assumption B.4, and setting $\eta \leq \frac{1}{2\beta}$,

$$\mathbb{E}_r\|x^{(r+1)} - x^*\|^2 \tag{20}$$

$$\leq (1 - \eta\mu)\|x^{(r)} - x^*\|^2 - \eta(F(x^{(r)}) - F(x^*)) + \eta^2(\frac{\sigma^2}{SK} + (1 - \frac{S-1}{N-1})\frac{\zeta^2}{S}) \tag{21}$$

Rearranging and taking full expectation,

$$\mathbb{E}F(x^{(r)}) - F(x^*) \leq \frac{(1 - \eta\mu)\mathbb{E}\|x^{(r)} - x^*\|^2 - \mathbb{E}\|x^{(r+1)} - x^*\|^2}{\eta} + \eta(\frac{\sigma^2}{SK} + (1 - \frac{S-1}{N-1})\frac{\zeta^2}{S}) \tag{22}$$

letting $w_r = (1 - \mu\eta)^{1-(r+1)}$, $W_R = \sum_{r=0}^{R} $, $\hat{x} = \frac{1}{W_R}\sum_{r=0}^{R} x^{(r)}$, then

$$\mathbb{E}F(\hat{x}) - F(x^*) \leq \frac{\mathbb{E}\|x^{(0)} - x^*\|^2}{\eta W_R} + \eta(\frac{\sigma^2}{SK} + (1 - \frac{S-1}{N-1})\frac{\zeta^2}{S}) \tag{23}$$

From (Karimireddy et al., 2020b) Lemma 1, if $R > \kappa$ and $\eta = \min\{\frac{1}{2\beta}, \frac{\log(\max(1, \mu^2 RD^2/c))}{\mu R}\}$ where $c = \frac{\sigma^2}{NK} + (1 - \frac{S-1}{N-1})\frac{\zeta^2}{S}$,

$$\mathbb{E}F(\hat{x}) - F(x^*) \leq \tilde{\mathcal{O}}(\Delta \exp(-\frac{R}{\kappa}) + \frac{\sigma^2}{\mu SKR} + (1 - \frac{S}{N})\frac{\zeta^2}{\mu SR}) \tag{24}$$

Which finishes our work on the convergence rate. Next we consider the distance bound. Returning to the recurrence

$$\mathbb{E}_r\|x^{(r+1)} - x^*\|^2 \tag{25}$$

$$\leq (1 - \eta\mu)\|x^{(r)} - x^*\|^2 - \eta(F(x^{(r)}) - F(x^*)) + \eta^2(\frac{\sigma^2}{SK} + (1 - \frac{S-1}{N-1})\frac{\zeta^2}{S}) \tag{26}$$

Taking full expectation and unrolling,

$$\mathbb{E}\|x^{(r)} - x^*\|^2 \tag{27}$$

$$\leq \mathbb{E}\|x^{(0)} - x^*\|^2 \exp(-\eta\mu r) + \frac{\eta}{\mu}\left(\frac{\sigma^2}{SK} + (1 - \frac{S-1}{N-1})\frac{\zeta^2}{S}\right) \tag{28}$$

Note that Karimireddy et al. (2020b) chose $\eta$ so that $\eta(\frac{\sigma^2}{SK} + (1 - \frac{S-1}{N-1})\frac{\zeta^2}{S}) \leq \tilde{\mathcal{O}}(\mu D^2 \exp(-\mu\eta R))$
And so

$$\mathbb{E}\|x^{(r)} - x^*\|^2 \leq \tilde{\mathcal{O}}(D^2) \tag{29}$$

$$\square$$

### D.1.2 CONVERGENCE OF SGD FOR GENERAL CONVEX FUNCTIONS

*Proof.* By smoothness (Assumption B.4), we have that

$$F(x^{(r+1)}) - F(x^{(r)}) \leq -\eta\langle\nabla F(x^{(r)}), g^{(r)}\rangle + \frac{\beta\eta^2}{2}\|g^{(r)}\|^2 \tag{30}$$

Taking expectation conditioned up to $r$ and using Lemma D.2,

$$\mathbb{E}_r F(x^{(r+1)}) - F(x^{(r)}) \leq -(\eta - \frac{\beta\eta^2}{2})\|\nabla F(x^{(r)})\|^2 + \frac{\beta\eta^2}{2}\left(\frac{\sigma^2}{SK} + (1 - \frac{S-1}{N-1})\frac{\zeta^2}{S}\right) \tag{31}$$

If $\eta \leq \frac{1}{\beta}$,

$$\mathbb{E}_r F(x^{(r+1)}) - F(x^{(r)}) \leq -\frac{\eta}{2}\|\nabla F(x^{(r)})\|^2 + \frac{\beta\eta^2}{2}\left(\frac{\sigma^2}{SK} + (1 - \frac{S-1}{N-1})\frac{\zeta^2}{S}\right) \tag{32}$$

Taking full expectation and rearranging,

$$\frac{1}{2}\mathbb{E}\|\nabla F(x^{(r)})\|^2 \leq \frac{\mathbb{E}F(x^{(r+1)}) - \mathbb{E}F(x^{(r)})}{\eta} + \frac{\beta\eta}{2}\left(\frac{\sigma^2}{SK} + (1 - \frac{S-1}{N-1})\frac{\zeta^2}{S}\right) \tag{33}$$

Summing both sides over $r$ and averaging,

$$\frac{1}{2}\frac{1}{R}\sum_{r=0}^{R-1}\mathbb{E}\|\nabla F(x^{(r)})\|^2 \leq \frac{\mathbb{E}F(x^{(0)}) - \mathbb{E}F(x^{(R)})}{\eta R} + \frac{\beta\eta}{2}\left(\frac{\sigma^2}{SK} + (1 - \frac{S-1}{N-1})\frac{\zeta^2}{S}\right) \tag{34}$$

Letting $\hat{x}$ be an average of all the $x^{(r)}$'s, noting that $\mathbb{E}F(x^{(0)}) - \mathbb{E}F(x^{(R)}) \leq \Delta$, $\Delta \leq \beta D^2$, and choosing $\eta = \min\{\frac{1}{\beta}, (\frac{\Delta}{\beta cR})^{1/2}\}$ where $c = \frac{\sigma^2}{SK} + (1 - \frac{S-1}{N-1})\frac{\zeta^2}{S}$,

$$\mathbb{E}\|\nabla F(\hat{x})\|^2 \leq \tilde{\mathcal{O}}\left(\frac{\beta\Delta}{R} + \frac{\beta\sigma D}{\sqrt{SKR}} + \sqrt{1 - \frac{S}{N}}\frac{\beta\zeta D}{\sqrt{SR}}\right) \tag{35}$$

So we have our convergence rate. Next, for the distance bound,

$$\|x^{(r+1)} - x^*\|^2 = \|x^{(r)} - x^*\|^2 - 2\eta\langle g^{(r)}, x^{(r)} - x^*\rangle + \eta^2\|g^{(r)}\|^2 \tag{36}$$

Taking expectation up to $r$, we know from Lemma D.2,

$$\mathbb{E}_r\|x^{(r+1)} - x^*\|^2 \tag{37}$$

$$\leq \|x^{(r)} - x^*\|^2 - 2\eta\langle g^{(r)}, x^{(r)} - x^*\rangle + \eta^2\|\nabla F(x^{(r)})\|^2 + \eta^2\left(\frac{\sigma^2}{SK} + (1 - \frac{S}{N})\frac{\zeta^2}{S}\right) \tag{38}$$

By Assumption B.2 and Assumption B.4,

$$\mathbb{E}_r\|x^{(r+1)} - x^*\|^2 \tag{39}$$

$$\leq \|x^{(r)} - x^*\|^2 - 2\eta(1 - \beta\eta)(F(x^{(r)}) - F(x^*)) + \eta^2\left(\frac{\sigma^2}{SK} + (1 - \frac{S}{N})\frac{\zeta^2}{S}\right) \tag{40}$$

And with $\eta \leq \frac{1}{\beta}$,

$$\mathbb{E}_r \|x^{(r+1)} - x^*\|^2 \leq \|x^{(r)} - x^*\|^2 + \eta^2 (\frac{\sigma^2}{SK} + (1 - \frac{S}{N})\frac{\zeta^2}{S}) \tag{41}$$

Unrolling and taking full expectation,

$$\mathbb{E}\|x^{(r)} - x^*\|^2 \leq \mathbb{E}\|x^{(0)} - x^*\|^2 + \eta^2 R(\frac{\sigma^2}{SK} + (1 - \frac{S}{N})\frac{\zeta^2}{S}) \tag{42}$$

We chose $\eta$ so that $\beta\eta(\frac{\sigma^2}{SK} + (1 - \frac{S}{N})\frac{\zeta^2}{S}) \leq \frac{\Delta}{\eta R}$. Therefore

$$\mathbb{E}\|x^{(r)} - x^*\|^2 \leq \mathbb{E}\|x^{(0)} - x^*\|^2 + \frac{\Delta}{\beta} \leq 2D^2 \tag{43}$$

$\square$

### D.1.3 CONVERGENCE OF SGD UNDER THE PL-CONDITION

*Proof.* Using Assumption B.4 we have that

$$F(x^{(r+1)}) - F(x^{(r)}) \leq -\eta\langle\nabla F(x^{(r)}), g^{(r)}\rangle + \frac{\beta\eta^2}{2}\|g^{(r)}\|^2 \tag{44}$$

Taking expectations of both sides conditioned on the $r$-th step,

$$\mathbb{E}_r F(x^{(r+1)}) - F(x^{(r)}) \leq -\eta\|F(x^{(r)})\|^2 + \frac{\beta\eta^2}{2}\mathbb{E}_r\|g^{(r)}\|^2 \tag{45}$$

Using Lemma D.2,

$$\mathbb{E}_r F(x^{(r+1)}) - F(x^{(r)}) \leq -\frac{\eta}{2}\|F(x^{(r)})\|^2 + \frac{\beta\eta^2}{2}(\frac{\sigma^2}{SK} + (1 - \frac{S-1}{N-1})\frac{\zeta^2}{S}) \tag{46}$$

Using Assumption B.3, we have that

$$\mathbb{E}_r F(x^{(r+1)}) - F(x^{(r)}) \leq -\eta\mu(F(x^{(r)}) - F(x^*)) + \frac{\beta\eta^2}{2}(\frac{\sigma^2}{SK} + (1 - \frac{S-1}{N-1})\frac{\zeta^2}{S}) \tag{47}$$

Rearranging,

$$\mathbb{E}_r F(x^{(r+1)}) - F(x^*) \leq (1 - \eta\mu)(F(x^{(r)}) - F(x^*)) + \frac{\beta\eta^2}{2}(\frac{\sigma^2}{SK} + (1 - \frac{S-1}{N-1})\frac{\zeta^2}{S}) \tag{48}$$

Letting $c = \frac{\sigma^2}{SK} + (1 - \frac{S-1}{N-1})\frac{\zeta^2}{S}$ and subtracting $\frac{\beta c\eta}{2\mu}$ from both sides,

$$\mathbb{E}_r F(x^{(r+1)}) - F(x^*) - \frac{\beta c\eta}{2\mu} \leq (1 - \eta\mu)(F(x^{(r)}) - F(x^*) - \frac{\beta c\eta}{2\mu}) \tag{49}$$

Which, upon unrolling the recursion, gives us

$$\mathbb{E}F(x^{(R)}) - F(x^*) \leq \Delta\exp(-\eta\mu R) + \frac{\beta c\eta}{2\mu} \tag{50}$$

Now, if we choose stepsize

$$\eta = \min\{\frac{1}{\beta}, \frac{\log(\max\{e, \frac{\mu^2\Delta R}{\beta c}\})}{\mu R}\} \tag{51}$$

Then the final convergence rate is

$$\mathbb{E}F(x^{(R)}) - F(x^*) \leq \tilde{\mathcal{O}}(\Delta\exp(-\frac{R}{\kappa}) + \frac{\kappa\sigma^2}{\mu SKR} + (1 - \frac{S}{N})\frac{\kappa\zeta^2}{\mu SR}) \tag{52}$$

$\square$

## D.2 ASG

The precise form of accelerated stochastic gradient we run is AC-SA(Ghadimi & Lan, 2012; 2013). The following specification is taken from Woodworth et al. (2020a):

We run Algo. 3 for $R_s$ iterations using $x^{(0)} = p_{s-1}$, $\{\alpha_t\}_{t \geq 1}$ and $\{\gamma_t\}_{t \geq 1}$, with definitions

- $R_s = \lceil \max\{4\sqrt{\frac{4\beta}{\mu}}, \frac{128c}{3\mu\Delta 2^{-(s+1)}}\} \rceil$
- $\alpha_r = \frac{2}{r+1}$, $\gamma_r = \frac{4\phi_s}{r(r+1)}$,
- $\phi_s = \max\{2\beta, [\frac{\mu c}{3\Delta 2^{-(s-1)} R_s(R_s+1)(R_s+2)}]^{1/2}\}$

Where $c = \frac{\sigma^2}{SK} + (1 - \frac{S-1}{N-1})\frac{\zeta^2}{S}$. Set $p_s = x_{\text{ag}}^{(R_s)}$ where $x_{\text{ag}}^{(R_s)}$ is the solution obtained in the previous step.

**Theorem D.3.** *Suppose that client objectives $F_i$'s and their gradient queries satisfy Assumption B.4 and Assumption B.6. Then running Algo. 3 gives the following for returned iterate $\hat{x}$:*

- ***Strongly convex:*** *$F_i$'s satisfy Assumption B.1 for $\mu > 0$. If we return $\hat{x} = x_{\text{ag}}^{(R_s)}$ after $R$ rounds of the multistage AC-SA specified above,*

$$\mathbb{E}F(\hat{x}) - F(x^*) \leq \mathcal{O}(\Delta \exp(-\frac{R}{\sqrt{\kappa}}) + \frac{\sigma^2}{\mu SKR} + (1 - \frac{S}{N})\frac{\zeta^2}{\mu SR})$$

$$\mathbb{E}\|\hat{x} - x^*\|^2 \leq \mathcal{O}(\kappa \mathbb{E}\|x^{(0)} - x^*\|^2 \exp(-\frac{R}{\sqrt{\kappa}}) + \frac{\sigma^2}{\mu^2 SKR} + (1 - \frac{S}{N})\frac{\zeta^2}{\mu^2 SR})$$

*And given no randomness, for any $0 \leq r \leq R$*

$$\|x^{(r)} - x^*\|^2 \leq \|x^{(0)} - x^*\|^2$$

$$\|x_{ag}^{(r)} - x^*\|^2 \leq \|x^{(0)} - x^*\|^2$$

*and for any $1 \leq r \leq R$*

$$\|x_{md}^{(r)} - x^*\|^2 \leq \|x^{(0)} - x^*\|^2$$

- ***General convex (grad norm):*** *$F_i$'s satisfy Assumption B.2. If we return $\hat{x} = x_{\text{ag}}^{(R_s)}$ after $R$ rounds of the multistage AC-SA specified above on the regularized objective $F_\mu(x) = F(x) + \frac{\mu}{2}\|x^{(0)} - x\|^2$ with $\mu = \max\{\frac{2\beta}{R^2}\log^2(e^2 + R^2), \sqrt{\frac{\beta\sigma^2}{\Delta SKR}}, \sqrt{\frac{(1-\frac{S}{N})\beta\zeta^2}{\Delta SR}}\}$, we have*

$$\mathbb{E}\|\nabla F(\hat{x})\|^2 \leq \tilde{\mathcal{O}}(\frac{\beta\Delta}{R^2} + \frac{\sigma^2}{SKR} + (1 - \frac{S}{N})\frac{\zeta^2}{SR} + \frac{\beta\sigma D}{\sqrt{SKR}} + \sqrt{1 - \frac{S}{N}}\frac{\zeta D}{\sqrt{SR}})$$

$$\mathbb{E}\|\hat{x} - x^*\|^2 \leq \tilde{\mathcal{O}}(D^2)$$

### D.2.1 CONVERGENCE OF ASG FOR STRONGLY CONVEX FUNCTIONS

*Proof.* The convergence rate proof comes from Woodworth et al. (2020a) Lemma 5. What remains is to show the distance bound.

From Proposition 5, (Eq. 3.25, 4.25) of Ghadimi & Lan (2012), we have that the generic (non-multistage) AC-SA satisfies (together with Lemma H.1)

$$\mathbb{E}F(x_{\text{ag}}^{(r)}) + \mu\mathbb{E}\|x^{(r)} - x^*\|^2 - F(x) \tag{53}$$

$$\leq \Gamma_r \sum_{\tau=1}^{r} \frac{\gamma_\tau}{\Gamma_\tau}[\mathbb{E}\|x^{(\tau-1)} - x^*\|^2 - \mathbb{E}\|x^{(\tau)} - x^*\|^2] + \Gamma_r \frac{4r}{\mu}(\frac{\sigma^2}{SK} + (1 - \frac{S}{N})\frac{\zeta^2}{S}) \tag{54}$$

Where $\Gamma_r = \begin{cases} 1, \ r = 1 \\ (1 - \alpha_r)\Gamma_{r-1} \ r \geq 2 \end{cases}$      Notice that

$$\Gamma_r \sum_{\tau=1}^{r} \frac{\gamma_\tau}{\Gamma_\tau} [\mathbb{E}\|x^{(\tau-1)} - x^*\|^2 - \mathbb{E}\|x^{(\tau)} - x^*\|^2] = \Gamma_t \gamma_1 [\|x^{(0)} - x^*\|^2 - \|x^{(r)} - x^*\|^2] \tag{55}$$

So

$$(\mu + \Gamma_t \gamma_1)\mathbb{E}\|x^{(r)} - x^*\|^2 \leq \Gamma_t \gamma_1 \mathbb{E}\|x^{(0)} - x^*\|^2 + \Gamma_r \frac{4r}{\mu}(\frac{\sigma^2}{SK} + (1 - \frac{S}{N})\frac{\zeta^2}{S}) \tag{56}$$

Which implies

$$\mathbb{E}\|x^{(r)} - x^*\|^2 \leq \mathbb{E}\|x^{(0)} - x^*\|^2 + \Gamma_r \frac{4r}{\mu^2}(\frac{\sigma^2}{SK} + (1 - \frac{S}{N})\frac{\zeta^2}{S}) \tag{57}$$

Furthermore, $\Gamma_r = \frac{2}{r(r+1)}$ so

$$\mathbb{E}\|x^{(r)} - x^*\|^2 \leq \mathbb{E}\|x^{(0)} - x^*\|^2 + \frac{8}{\mu^2(r+1)}(\frac{\sigma^2}{SK} + (1 - \frac{S}{N})\frac{\zeta^2}{S}) \tag{58}$$

If there is no gradient variance or sampling,

$$\|x^{(r)} - x^*\|^2 \leq \|x^{(0)} - x^*\|^2 \tag{59}$$

Next, we show the above two conclusions hold for $x_{md}^{(r)}$ and $x_{ag}^{(r)}$.

For $x_{ag}^{(r)}$, we show by induction:

**Base case:** $x_{ag}^{(r)} = x^{(0)}$, so it is true

**Inductive case:** $x_{ag}^{(r)}$ is a convex combination of $x^{(r)}$ and $x_{ag}^{(r-1)}$, so the above statements hold for $x_{ag}^{(r)}$ as well.

For $x_{md}^{(r)}$, note it is a convex combination of $x_{ag}^{(r-1)}$ and $x^{(r-1)}$, so the above statements on distance also hold for $x_{md}^{(r)}$.

For the distance bound on the returned solution, use strong convexity on the convergence rate:

$$\mathbb{E}\|\hat{x} - x^*\|^2 \leq \mathcal{O}(\kappa \mathbb{E}\|x^{(0)} - x^*\|^2 \exp(-\frac{R}{\sqrt{\kappa}}) + \frac{c}{\mu^2 R}) \tag{60}$$

$\square$

### D.2.2   CONVERGENCE OF ASG FOR GENERAL CONVEX FUNCTIONS

For the general convex case, we use Nesterov smoothing. Concretely, we will run Algo. 3 assuming strong convexity by optimizing instead a modified objective

$$F_\mu(x) = F(x) + \frac{\mu}{2}\|x^{(0)} - x\|^2 \tag{61}$$

Define $x_\mu^* = \arg\min_x F_\mu(x)$ and $\Delta_\mu = \mathbb{E}F_\mu(x^{(0)}) - F(x_\mu^*)$. We will choose $\mu$ carefully to balance the error introduced by the regularization term and the better convergence properties of having larger $\mu$.

*Proof.* Observe that

$$\mathbb{E}\|\nabla F(\hat{x})\|^2 = \mathbb{E}\|\nabla F_\mu(\hat{x}) - \mu(\hat{x} - x^{(0)})\|^2 \leq 2\mathbb{E}\|\nabla F_\mu(\hat{x})\|^2 + 2\mu^2 \mathbb{E}\|\hat{x} - x^{(0)}\|^2 \tag{62}$$

We evaluate the second term first. We know that from the theorem statement on strongly convex functions and letting $\kappa' = \frac{\beta+\mu}{\mu}$ and $\kappa = \frac{\beta}{\mu}$ because $F$ is now $\beta + \mu$-smooth,

$$\mathbb{E}F_\mu(\hat{x}) - F_\mu(x_\mu^*) \leq \mathcal{O}((\mathbb{E}F_\mu(x^{(0)}) - F_\mu(x_\mu^*))\exp(-\frac{R}{\sqrt{\kappa'}}) + \frac{\sigma^2}{\mu SKR} + (1 - \frac{S}{N})\frac{\zeta^2}{\mu SR}) \tag{63}$$

Which implies

$$\mathbb{E}F_\mu(\hat{x}) \leq \mathcal{O}(\mathbb{E}F_\mu(x^{(0)}) + \frac{\sigma^2}{\mu SKR} + (1 - \frac{S}{N})\frac{\zeta^2}{\mu SR}) \tag{64}$$

$$= \mathcal{O}(\mathbb{E}F(x^{(0)}) + \frac{\sigma^2}{\mu SKR} + (1 - \frac{S}{N})\frac{\zeta^2}{\mu SR}) \tag{65}$$

So it is true that

$$\mathbb{E}F_\mu(\hat{x}) = \mathbb{E}[F(\hat{x}) + \frac{\mu}{2}\|\hat{x} - x^{(0)}\|^2] \tag{66}$$

$$= \mathcal{O}(\mathbb{E}F(x^{(0)}) + \frac{\sigma^2}{\mu SKR} + (1 - \frac{S}{N})\frac{\zeta^2}{\mu SR}) \tag{67}$$

If we rearrange,

$$\frac{\mu}{2}\mathbb{E}\|\hat{x} - x^{(0)}\|^2 \leq \mathcal{O}(\mathbb{E}F(x^{(0)}) - F(\hat{x}) + \frac{\sigma^2}{\mu SKR} + (1 - \frac{S}{N})\frac{\zeta^2}{\mu SR}) \tag{68}$$

$$\leq \mathcal{O}(\mathbb{E}F(x^{(0)}) - F(x^*) + \frac{\sigma^2}{\mu SKR} + (1 - \frac{S}{N})\frac{\zeta^2}{\mu SR}) \tag{69}$$

$$\leq \mathcal{O}(\Delta + \frac{\sigma^2}{\mu SKR} + (1 - \frac{S}{N})\frac{\zeta^2}{\mu SR}) \tag{70}$$

Next we evaluate $2\mathbb{E}\|\nabla F_\mu(\hat{x})\|^2$. Observe that the smoothness of $F_\mu$ is $\beta + \mu$. Returning to the convergence rate,

$$\mathbb{E}F_\mu(\hat{x}) - F_\mu(x_\mu^*) \leq \mathcal{O}(\Delta_\mu \exp(-\frac{R}{\sqrt{\kappa'}}) + \frac{\sigma^2}{\mu SKR} + (1 - \frac{S}{N})\frac{\zeta^2}{\mu SR}) \tag{71}$$

We have that

$$F_\mu(x^{(0)}) - F_\mu(x_\mu^*) = F(x^{(0)}) - F(x_\mu^*) - \frac{\mu}{2}\|x^{(0)} - x_\mu^*\|^2 \leq F(x^{(0)}) - F(x^*) \tag{72}$$

So $\Delta_\mu \leq \Delta$. By Assumption B.4,

$$\mathbb{E}\|\nabla F_\mu(\hat{x})\|^2 \leq (\beta + \mu)\mathcal{O}(\Delta \exp(-\frac{R}{\sqrt{\kappa'}}) + \frac{\sigma^2}{\mu SKR} + (1 - \frac{S}{N})\frac{\zeta^2}{\mu SR}) \tag{73}$$

$$\leq \mathcal{O}(\beta\Delta \exp(-\frac{R}{\sqrt{\kappa'}}) + \mu\Delta + \frac{\beta\sigma^2}{\mu SKR} + \frac{\sigma^2}{SKR} + (1 - \frac{S}{N})\frac{\zeta^2}{SR} + (1 - \frac{S}{N})\frac{\beta\zeta^2}{\mu SR}) \tag{74}$$

So altogether,

$$\mathbb{E}\|\nabla F(\hat{x})\|^2 \leq \mathcal{O}(\beta\Delta \exp(-\frac{R}{\sqrt{\kappa'}}) + (1 + \kappa)\frac{\sigma^2}{SKR} + (1 + \kappa)(1 - \frac{S}{N})\frac{\zeta^2}{SR} + \mu\Delta) \tag{75}$$

Choose $\mu = \max\{\frac{2\beta}{R^2}\log^2(e^2 + R^2), \sqrt{\frac{\beta\sigma^2}{\Delta SKR}}, \sqrt{\frac{(1-\frac{S}{N})\beta\zeta^2}{\Delta SR}}\}$. By Theorem E.1's proof in Yuan & Ma (2020),

$$\mathbb{E}\|\nabla F(\hat{x})\|^2 \leq \tilde{\mathcal{O}}(\frac{\beta\Delta}{R^2} + \frac{\sigma^2}{SKR} + (1 - \frac{S}{N})\frac{\zeta^2}{SR} + \frac{\beta\sigma D}{\sqrt{SKR}} + \sqrt{1 - \frac{S}{N}}\frac{\zeta D}{\sqrt{SR}}) \tag{76}$$

Next we evaluate the distance bound for the returned iterate. Observe that from the distance bound in the strongly convex case,

$$\mathbb{E}\|\hat{x} - x_\mu^*\|^2 \leq \mathcal{O}(\frac{\beta + \mu}{\mu}\mathbb{E}\|x^{(0)} - x_\mu^*\|^2 \exp(-\frac{R}{\sqrt{\kappa}}) + \frac{\sigma^2}{\mu^2 SKR} + (1 - \frac{S}{N})\frac{\zeta^2}{\mu^2 SR}) \tag{77}$$

Given that $\mu = \max\{\frac{2\beta}{R^2}\log^2(e^2 + R^2), \sqrt{\frac{\beta\sigma^2}{\Delta SKR}}, \sqrt{\frac{(1-\frac{S}{N})\beta\zeta^2}{\Delta SR}}\}$,

$$\mathbb{E}\|\hat{x} - x_\mu^*\|^2 \leq \mathcal{O}(\mathbb{E}\|x^{(0)} - x_\mu^*\|^2 + \|\hat{x} - x_\mu^*\|^2) \tag{78}$$

$$\leq \tilde{\mathcal{O}}(\mathbb{E}\|x^{(0)} - x_\mu^*\|^2) \tag{79}$$

Now we look at the actual distance we want to bound:

$$\mathbb{E}\|\hat{x} - x^*\|^2 \le \tilde{\mathcal{O}}(\mathbb{E}\|\hat{x} - x_\mu^*\|^2 + \mathbb{E}\|x_\mu^* - x^{(0)}\|^2 + \mathbb{E}\|x^* - x^{(0)}\|^2) \tag{80}$$

$$\tag{81}$$

Observe that

$$F_\mu(x_\mu^*) = F(x_\mu^*) + \|x^{(0)} - x_\mu^*\|^2 \le F(x^*) + \|x^{(0)} - x^*\|^2 = F_\mu(x^*) \tag{82}$$

which implies that

$$\|x^{(0)} - x_\mu^*\|^2 \le \|x^{(0)} - x^*\|^2 \tag{83}$$

So altogether

$$\mathbb{E}\|\hat{x} - x^*\|^2 \le \tilde{\mathcal{O}}(D^2) \tag{84}$$

$\square$

## D.3 SAGA

**Theorem D.4.** *Suppose that client objectives $F_i$'s and their gradient queries satisfy Assumption B.4 and Assumption B.6. Then running Algo. 5 gives the following for returned iterate $\hat{x}$:*

- **Strongly convex:** *$F_i$'s satisfy Assumption B.1 for $\mu > 0$. If we return $\hat{x} = \frac{1}{W_R} \sum_{r=0}^R w_r x^{(r)}$ with $w_r = (1 - \eta\mu)^{1-(r+1)}$ and $W_R = \sum_{r=0}^R w_r$, $\eta = \tilde{\mathcal{O}}(\min\{\frac{1}{\beta}, \frac{1}{\mu R}\})$, $R > \kappa$, and we use Option I,*

$$\mathbb{E}F(\hat{x}) - F(x^*) \le \tilde{\mathcal{O}}(\Delta \exp(-\min\{\frac{\mu}{\beta}, \frac{S}{N}\}R) + \frac{\sigma^2}{\mu RKS})$$

- **PL condition:** *$F_i$'s satisfy Assumption B.3. If we set $\eta = \frac{1}{3\beta(N/S)^{2/3}}$, use Option II in a multistage manner (details specified in proof), and return a uniformly sampled iterate from the final stage,*

$$\mathbb{E}F(\hat{x}) - F(x^*) \le \mathcal{O}(\Delta \exp(-\frac{R}{\kappa(N/S)^{2/3}}) + \frac{\sigma^2}{\mu SK}) \tag{85}$$

### D.3.1 CONVERGENCE OF SAGA FOR STRONGLY CONVEX FUNCTIONS

The proof of this is similar to that of Karimireddy et al. (2020b, Theorem VII).

*Proof.* Following the standard analysis,

$$\mathbb{E}_r\|x^{(r+1)} - x^*\|^2 \tag{86}$$

$$= \|x^{(r)} - x^*\|^2 - 2\eta\mathbb{E}_r\langle g^{(r)}, x^{(r)} - x^*\rangle + \mathbb{E}_r\|\eta g^{(r)}\|^2 \tag{87}$$

$$\tag{88}$$

We treat each of these terms separately.

**Second term:** Observe first that

$$\mathbb{E}_r g^{(r)} = \mathbb{E}_r(\frac{1}{S}\sum_{i\in\mathcal{S}_r}[\frac{1}{K}\sum_{k=0}^{K-1}\nabla f(x^{(r)}; z_{i,k}^{(r)}) - c_i^{(r)}] + c^{(r)}) \tag{89}$$

$$= \nabla F(x^{(r)}) \tag{90}$$

And so the middle term has, by (strong) convexity,

$$-2\eta\mathbb{E}_r\langle g^{(r)}, x^{(r)} - x^*\rangle = -2\eta\langle\nabla F(x^{(r)}), x^{(r)} - x^*\rangle \tag{91}$$

$$\le -2\eta(F(x^{(r)}) - F(x^*) + \frac{\mu}{2}\|x^{(r)} - x^*\|^2) \tag{92}$$

**Third term:**

$$\mathbb{E}_r \|\eta g^{(r)}\|^2 \tag{93}$$

$$= \mathbb{E}_r \|\eta (\frac{1}{S} \sum_{i \in \mathcal{S}_r} [\frac{1}{K} \sum_{k=0}^{K-1} \nabla f(x^{(r)}; z_{i,k}^{(r)}) - c_i^{(r)}] + c^{(r)})\|^2 \tag{94}$$

$$= \eta^2 \mathbb{E}_r \|\frac{1}{KS} \sum_{k,i \in \mathcal{S}_r} \nabla f(x^{(r)}; z_{i,k}^{(r)}) - c_i^{(r)} + c^{(r)}\|^2 \tag{95}$$

$$\leq 4\eta^2 \mathbb{E}_r \|\frac{1}{KS} \sum_{k,i \in \mathcal{S}_r} \nabla f(x^{(r)}; z_{i,k}^{(r)}) - \nabla F_i(x^{(r)})\|^2 + 4\eta^2 \mathbb{E}_r \|c^{(r)}\|^2 \tag{96}$$

$$+ 4\eta^2 \mathbb{E}_r \|\frac{1}{KS} \sum_{k,i \in \mathcal{S}_r} \nabla F_i(x^*) - c_i^{(r)}\|^2 + 4\eta^2 \mathbb{E}_r \|\frac{1}{KS} \sum_{k,i \in \mathcal{S}_r} \nabla F_i(x^{(r)}) - \nabla F_i(x^*)\|^2 \tag{97}$$

$$\tag{98}$$

Where the last inequality comes from the relaxed triangle inequality. Then, by using Assumption B.6, Assumption B.4, and Assumption B.2,

$$\mathbb{E}_r \|\eta g^{(r)}\|^2 \tag{99}$$

$$\leq 4\eta^2 \mathbb{E}_r \|c^{(r)}\|^2 + 4\eta^2 \mathbb{E}_r \|\frac{1}{KS} \sum_{k,i \in \mathcal{S}_r} \nabla F_i(x^*) - c_i^{(r)}\|^2 + 8\beta\eta^2 [F(x^{(r)}) - F(x^*)] + \frac{4\eta^2 \sigma^2}{KS} \tag{100}$$

Taking full expectation and separating out the variance of the control variates,

$$\mathbb{E} \|\eta g^{(r)}\|^2 \tag{101}$$

$$\leq 4\eta^2 \|\mathbb{E} c^{(r)}\|^2 + 4\eta^2 \|\frac{1}{KS} \sum_{k,i \in \mathcal{S}_r} \nabla F_i(x^*) - \mathbb{E} c_i^{(r)}\|^2 + 8\beta\eta^2 \mathbb{E}[F(x^{(r)}) - F(x^*)] + \frac{12\eta^2 \sigma^2}{KS} \tag{102}$$

Now observe that because $c^{(r)} = \frac{1}{N} \sum_{i=1}^N c_i^{(r)}$,

$$\mathbb{E} \|\eta g^{(r)}\|^2 \leq \frac{8\eta^2}{N} \sum_{i=1}^N \|\nabla F_i(x^*) - \mathbb{E} c_i^{(r)}\|^2 + 8\beta\eta^2 \mathbb{E}[F(x^{(r)}) - F(x^*)] + \frac{12\eta^2 \sigma^2}{KS} \tag{103}$$

$$\leq 8\eta^2 \mathcal{C}_r + 8\beta\eta^2 \mathbb{E}[F(x^{(r)}) - F(x^*)] + \frac{12\eta^2 \sigma^2}{KS} \tag{104}$$

**Bounding the control lag:**

Recall that

$$c_i^{(r+1)} = \begin{cases} c_i^{(r)} \text{ w.p. } 1 - \frac{S}{N} \\ \frac{1}{K} \sum_{k=0}^{K-1} \nabla f(x^{(r)}; z_{i,k}^{(r)}) \text{ w.p. } \frac{S}{N} \end{cases} \tag{105}$$

Therefore

$$\mathbb{E} c_i^{(r+1)} = (1 - \frac{S}{N}) \mathbb{E}[c_i^{(r)}] + \frac{S}{N} \mathbb{E} \nabla F_i(x^{(r)}) \tag{106}$$

Returning to the definition of $\mathcal{C}_{r+1}$,

$$\mathcal{C}_{r+1} = \frac{1}{N} \sum_{i=1}^N \mathbb{E} \|\mathbb{E}[c_i^{(r+1)}] - \nabla F_i(x^*)\|^2 \tag{107}$$

$$= \frac{1}{N} \sum_{i=1}^N \mathbb{E} \|(1 - \frac{S}{N})(\mathbb{E}[c_i^{(r)}] - \nabla F_i(x^*)) + \frac{S}{N}(\mathbb{E} \nabla F_i(x^{(r)}) - \nabla F_i(x^*))\|^2 \tag{108}$$

$$\leq (1 - \frac{S}{N}) \mathcal{C}_r + \frac{S}{N^2} \sum_{i=1}^N \mathbb{E} \|\nabla F_i(x^{(r)}) - \nabla F_i(x^*)\|^2 \tag{109}$$

where in the last inequality we applied Jensen's inequality twice. By smoothness of $F_i$'s (Assumption B.4),

$$\mathcal{C}_{r+1} \leq (1 - \frac{S}{N})\mathcal{C}_r + \frac{2\beta S}{N}[\mathbb{E}F(x^{(r)}) - F(x^*)] \tag{110}$$

**Putting it together:**

So putting it all together, we have

$$\mathbb{E}\|x^{(r+1)} - x^*\|^2 \tag{111}$$

$$\leq (1 - \eta\mu)\mathbb{E}\|x^{(r)} - x^*\|^2 - \eta(2 - 8\beta\eta)(\mathbb{E}F(x^{(r)}) - F(x^*)) + 8\eta^2\mathcal{C}_r + \frac{12\eta^2\sigma^2}{KS} \tag{112}$$

From our bound on the control lag,

$$\frac{9\eta^2 N}{S}\mathcal{C}_{r+1} \leq \frac{9\eta^2 N}{S}(1 - \frac{S}{N})\mathcal{C}_r + 18\beta\eta^2[\mathbb{E}F(x^{(r)}) - F(x^*)] \tag{113}$$

$$= (1 - \eta\mu)\frac{9\eta^2 N}{S}\mathcal{C}_r + 9\eta^2(\frac{\eta\mu N}{S} - 1)\mathcal{C}_r + 18\beta\eta^2[\mathbb{E}F(x^{(r)}) - F(x^*)] \tag{114}$$

Adding both inequalities, we have

$$\mathbb{E}\|x^{(r+1)} - x^*\|^2 + \frac{9\eta^2 N}{S}\mathcal{C}_{r+1} \tag{115}$$

$$\leq (1 - \eta\mu)[\mathbb{E}\|x^{(r)} - x^*\|^2 + \frac{9\eta^2 N}{S}\mathcal{C}_r] - \eta(2 - 26\beta\eta)(\mathbb{E}F(x^{(r)}) - F(x^*)) \tag{116}$$

$$+ \eta^2(\frac{9\eta\mu N}{S} - 1)\mathcal{C}_r + \frac{12\eta^2\sigma^2}{KS} \tag{117}$$

Let $\eta \leq \frac{1}{26\beta}, \eta \leq \frac{S}{9\mu N}$, then

$$\mathbb{E}\|x^{(r+1)} - x^*\|^2 + \frac{9\eta^2 N}{S}\mathcal{C}_{r+1} \tag{118}$$

$$\leq (1 - \eta\mu)[\mathbb{E}\|x^{(r)} - x^*\|^2 + \frac{9\eta^2 N}{S}\mathcal{C}_r] - \eta(\mathbb{E}F(x^{(r)}) - F(x^*)) + \frac{12\eta^2\sigma^2}{KS} \tag{119}$$

Then by using Lemma 1 in (Karimireddy et al., 2020b), setting $\eta_{\max} = \min\{\frac{1}{26\beta}, \frac{S}{9\mu N}\}$, $R \geq \frac{1}{2\eta_{\max}\mu}$, and choosing $\eta = \min\{\frac{\log(\max(1, \mu^2 R d_0/c))}{\mu R}, \eta_{\max}\}$ where $d_0 = \mathbb{E}\|x^{(0)} - x^*\|^2 + \frac{9N\eta^2}{S}\mathcal{C}_0$ and $c = \frac{12\sigma^2}{KS}$, we have that by outputting $\hat{x} = \frac{1}{W_R}\sum_{r=0}^R w_r x^{(r)}$ with $W_R = \sum_{r=0}^R w_r$ and $w_r = (1 - \mu\eta)^{1-r}$, we have

$$\mathbb{E}F(\hat{x}) - F(x^*) \leq \tilde{\mathcal{O}}(\mu d_0 \exp(-\mu\eta_{\max}R) + \frac{c}{\mu R}) \tag{120}$$

$$= \tilde{\mathcal{O}}(\mu d_0 \exp(-\mu\eta_{\max}R) + \frac{c}{\mu R}) \tag{121}$$

$$= \tilde{\mathcal{O}}(\mu[\mathbb{E}\|x^{(0)} - x^*\|^2 + \frac{N\eta^2}{S}\mathcal{C}_0]\exp(-\mu\eta_{\max}R) + \frac{\sigma^2}{\mu RKS}) \tag{122}$$

We use a warm-start strategy to initialize all control variates in the first $N/S$ rounds such that

$$c_i^{(0)} = \frac{1}{K}\sum_{k=0}^{K-1}\nabla f(x^{(0)}; z_{i,k}^{(-1)})$$

By smoothness of $F_i$'s (Assumption B.4),

$$\mathcal{C}_0 = \frac{1}{N}\sum_{i=1}^N \mathbb{E}\|\mathbb{E}c_i^{(0)} - \nabla F_i(x^*)\|^2 \tag{123}$$

$$= \frac{1}{N}\sum_{i=1}^N \mathbb{E}\|\nabla F_i(x^{(0)}) - \nabla F_i(x^*)\|^2 \tag{124}$$

$$\leq \beta\mathbb{E}(F(x^{(0)}) - F(x^*)) \tag{125}$$

And recalling that $\eta \leq \min\{\frac{1}{26\beta}, \frac{S}{9\mu N}\}$, we know that

$$\frac{9N\eta^2}{S}\mathcal{C}_0 \leq \frac{9N\eta^2}{S}\beta\mathbb{E}(F(x^{(0)}) - F(x^*)) \leq \frac{\eta\beta}{\mu}(F(x^{(0)}) - F(x^*)) \leq \frac{1}{\mu}\Delta \qquad (126)$$

So altogether,

$$\mathbb{E}F(\hat{x}) - F(x^*) \leq \tilde{\mathcal{O}}(\Delta\exp(-\mu\eta_{\max}R) + \frac{\sigma^2}{\mu RKS}) \qquad (127)$$

$\square$

### D.3.2 CONVERGENCE OF SAGA UNDER THE PL CONDITION

This proof follows Reddi et al. (2016).

*Proof.* We start with Assumption B.4

$$\mathbb{E}F(x^{(r+1)}) \leq \mathbb{E}[F(x^{(r)}) + \langle\nabla F(x^{(r)}), x^{(r+1)} - x^{(t)}\rangle + \frac{\beta}{2}\|x^{(r+1)} - x^{(r)}\|^2] \qquad (128)$$

Using the fact that $g^{(r)}$ is unbiased,

$$\mathbb{E}F(x^{(r+1)}) \leq \mathbb{E}[F(x^{(r)}) - \eta\|\nabla F(x^{(r)})\|^2 + \frac{\beta\eta^2}{2}\|g^{(r)}\|^2] \qquad (129)$$

Now we consider the Lyapunov function

$$L_r = \mathbb{E}[F(x^{(r)}) + \frac{c_r}{N}\sum_{i=1}^{N}\|x^{(r)} - \phi_i^{(r)}\|^2] \qquad (130)$$

We bound $L_{r+1}$ using

$$\frac{1}{N}\sum_{i=1}^{N}\mathbb{E}\|x^{(r+1)} - \phi_i^{(r+1)}\|^2 \qquad (131)$$

$$= \frac{1}{N}\sum_{i=1}^{N}[\frac{S}{N}\mathbb{E}\|x^{(r+1)} - x^{(r)}\|^2 + \frac{N-S}{N}\mathbb{E}\|x^{(r+1)} - \phi_i^{(r)}\|^2] \qquad (132)$$

Where the equality comes from how $\phi_i^{(r+1)} = x^{(r)}$ with probability $S/N$ and is $\phi_i^{(r)}$ otherwise. Observe that we can bound

$$\mathbb{E}\|x^{(r+1)} - \phi_i^{(r)}\|^2 \qquad (133)$$

$$= \mathbb{E}[\|x^{(r+1)} - x^{(r)}\|^2 + \|x^{(r)} - \phi_i^{(r)}\|^2 + 2\langle x^{(r+1)} - x^{(r)}, x^{(r)} - \phi_i^{(r)}\rangle] \qquad (134)$$

$$\leq \mathbb{E}[\|x^{(r+1)} - x^{(r)}\|^2 + \|x^{(r)} - \phi_i^{(r)}\|^2] + 2\eta\mathbb{E}[\frac{1}{2b}\|\nabla F(x^{(r)})\|^2 + \frac{1}{2}b\|x^{(r)} - \phi_i^{(r)}\|^2] \qquad (135)$$

Where we used unbiasedness of $g^{(r)}$ and Fenchel-Young inequality. Plugging this into $L_{r+1}$,

$$L_{r+1} \leq \mathbb{E}[F(x^{(r)}) - \eta\|\nabla F(x^{(r)})\|^2 + \frac{\beta\eta^2}{2}\|g^{(r)}\|^2] \qquad (136)$$

$$+ \mathbb{E}[c_{r+1}\|x^{(r+1)} - x^{(r)}\|^2 + c_{r+1}\frac{N-S}{N^2}\sum_{i=1}^{N}\|x^{(r)} - \phi_i^{(r)}\|^2] \qquad (137)$$

$$+ \frac{2(N-S)c_{r+1}\eta}{N^2}\sum_{i=1}^{N}\mathbb{E}[\frac{1}{2b}\|\nabla F(x^{(r)})\|^2 + \frac{1}{2}b\|x^{(r)} - \phi_i^{(r)}\|^2] \qquad (138)$$

$$\leq \mathbb{E}[F(x^{(r)}) - (\eta - \frac{c_{r+1}\eta(N-S)}{bN})\|\nabla F(x^{(r)})\|^2 + (\frac{\beta\eta^2}{2} + c_{r+1}\eta^2)\mathbb{E}\|g^{(r)}\|^2] \qquad (139)$$

$$+ (\frac{N-S}{N}c_{r+1} + \frac{c_{r+1}\eta b(N-S)}{N})\frac{1}{N}\sum_{i=1}^{N}\mathbb{E}\|x^{(r)} - \phi_i^{(r)}\|^2 \qquad (140)$$

Now we must bound $\mathbb{E}\|g^{(r)}\|^2$.

$$\mathbb{E}\|(\frac{1}{S}\sum_{i\in\mathcal{S}_r}[\frac{1}{K}\sum_{k=0}^{K-1}\nabla f(x^{(r)};z_{i,k}^{(r)})-\nabla f(\phi_i^{(r)};\tilde{z}_{i,k}^{(r)})]+\frac{1}{NK}\sum_{i=1}^{N}\nabla f(\phi_i^{(r)};\tilde{z}_{i,k}^{(r)}))\|^2 \tag{141}$$

$$\leq 2\mathbb{E}\|(\frac{1}{S}\sum_{i\in\mathcal{S}_r}[\frac{1}{K}\sum_{k=0}^{K-1}\nabla f(x^{(r)};z_{i,k}^{(r)})-\nabla f(\phi_i^{(r)};\tilde{z}_{i,k}^{(r)})] \tag{142}$$

$$-\frac{1}{NK}\sum_{i=1}^{N}[\nabla f(x^{(r)};z_{i,k}^{(r)})-\nabla f(\phi_i^{(r)};\tilde{z}_{i,k}^{(r)}))]\|^2 + 2\mathbb{E}\|\frac{1}{NK}\sum_{i=1}^{N}\nabla f(x^{(r)};z_{i,k}^{(r)})\|^2 \tag{143}$$

$$\leq 2\mathbb{E}\|\frac{1}{S}\sum_{i\in\mathcal{S}_r}\nabla F_i(x^{(r)})-\nabla F_i(\phi_i^{(r)})\|^2 + 2\mathbb{E}\|\nabla F(x^{(r)})\|^2 + \frac{\nu\sigma^2}{SK} \tag{144}$$

Where the second to last inequality is an application of $\mathrm{Var}(X)\leq \mathbb{E}[X^2]$, and separating out the variance (taking advantage of the fact that $\tilde{z}$'s and $z$'s are independent), and $\nu$ is some constant.

We use Assumption B.4 and take expectation over sampled clients to get

$$\mathbb{E}\|g^{(r)}\|^2 \leq \frac{2\beta^2}{N}\sum_{i=1}^{N}\mathbb{E}\|x^{(t)}-\phi_i^{(r)}\|^2 + 2\mathbb{E}\|\nabla F(x^{(r)})\|^2 + \frac{\nu\sigma^2}{SK} \tag{145}$$

Returning to our bound on $L_{r+1}$,

$$L_{r+1} \leq \mathbb{E}F(x^{(r)}) - (\eta - \frac{c_{r+1}\eta(N-S)}{bN} - \eta^2\beta - 2c_{r+1}\eta^2)\mathbb{E}\|\nabla F(x^{(r)})\|^2 \tag{146}$$

$$+ (c_{r+1}(\frac{N-S}{N} + \frac{\eta b(N-S)}{N} + 2\eta^2\beta^2) + \eta^2\beta^3)\frac{1}{N}\sum_{i=1}^{N}\mathbb{E}\|x^{(r)}-\phi_i^{(r)}\|^2 \tag{147}$$

$$+ \frac{\nu\eta^2\sigma^2}{SK}(c_{r+1}+\frac{\beta}{2}) \tag{148}$$

We set $c_r = c_{r+1}(\frac{N-S}{N} + \frac{\eta b(N-S)}{N} + 2\eta^2\beta^2) + \eta^2\beta^3$, which results in

$$L_{r+1} \tag{149}$$

$$\leq L_r - (\eta - \frac{c_{r+1}\eta(N-S)}{bN} - \eta^2\beta - 2c_{r+1}\eta^2)\mathbb{E}\|\nabla F(x^{(r)})\|^2 + \frac{\nu\eta^2\sigma^2}{SK}(c_{r+1}+\frac{\beta}{2}) \tag{150}$$

Let $\Gamma_r = \eta - \frac{c_{r+1}\eta(N-S)}{bN} - \eta^2\beta - 2c_{r+1}\eta^2$. Then by rearranging

$$\Gamma_r\mathbb{E}\|\nabla F(x^{(r)})\|^2 \leq L_r - L_{r+1} + \frac{\nu\eta^2\sigma^2}{SK}(c_{r+1}+\frac{\beta}{2}) \tag{151}$$

Letting $\gamma_n = \min_{0\leq r\leq R-1}\Gamma_r$,

$$\gamma_n\sum_{r=0}^{R-1}\mathbb{E}\|\nabla F(x^{(r)})\|^2 \leq \sum_{r=0}^{R-1}\Gamma_r\mathbb{E}\|\nabla F(x^{(r)})\|^2 \tag{152}$$

$$\leq L_0 - L_R + \sum_{r=0}^{R-1}\frac{\nu\eta^2\sigma^2}{SK}(c_{r+1}+\frac{\beta}{2}) \tag{153}$$

Implying

$$\sum_{r=0}^{R-1}\mathbb{E}\|\nabla F(x^{(r)})\|^2 \leq \frac{\Delta}{\gamma_n} + \frac{1}{\gamma_n}\sum_{r=0}^{R-1}\frac{\nu\eta^2\sigma^2}{SK}(c_{r+1}+\frac{\beta}{2}) \tag{154}$$

Therefore, if we take a uniform sample from all the $x(r)$, denoted $\bar{x}^R$,

$$\mathbb{E}\|\nabla F(\bar{x}^{(R)})\|^2 \leq \frac{\Delta}{\gamma_n R} + \frac{1}{\gamma_n R}\sum_{r=0}^{R-1}\frac{\nu\eta^2\sigma^2}{SK}(c_{r+1}+\frac{\beta}{2}) \tag{155}$$

We start by bounding $c_r$. Take $\eta = \frac{1}{3\beta}(\frac{S}{N})^{2/3}$ and $b = \beta(\frac{S}{N})^{1/3}$. Let $\theta = \frac{S}{N} - \frac{\eta b(N-S)}{N} - 2\eta^2\beta^2$. Observe that $\theta < 1$ and $\theta > \frac{4}{9}\frac{S}{N}$. Then $c_r = c_{r+1}(1 - \theta) + \eta^2\beta^3$, which implies $c_r = \eta^2\beta^3\frac{1-(1-\theta)^{R-r}}{\theta} \le \frac{\eta^2\beta^3}{\theta} \le \frac{\beta}{4}(\frac{S}{N})^{1/3}$

So we can conclude that

$$\gamma_n = \min_r(\eta - \frac{c_{r+1}\eta}{\beta} - \eta^2\beta - 2c_{r+1}\eta^2) \ge \frac{1}{12\beta}(\frac{S}{N})^{2/3} \tag{156}$$

So altogether,

$$\mathbb{E}\|\nabla F(\bar{x}^{(R)})\|^2 \le \frac{12\beta\Delta}{R}(\frac{N}{S})^{2/3} + \frac{\nu\eta^2\beta\sigma^2}{SK}(\frac{1}{12\beta(N/S)^{2/3}}) \tag{157}$$

$$= \frac{12\beta\Delta}{R}(\frac{N}{S})^{2/3} + \frac{\nu\beta\sigma^2}{SK}(12\beta(N/S)^{2/3})(\frac{1}{9\beta^2(N/S)^{4/3}}) \tag{158}$$

$$\le \frac{12\beta\Delta}{R}(\frac{N}{S})^{2/3} + \frac{2\nu\sigma^2}{SK} \tag{159}$$

Now we run Algo. 5 in a repeated fashion, as follows:

1. Set $x^{(0)} = p_{s-1}$

2. Run Algo. 5 for $R_s$ iterations

3. Set $p_s$ to the result of Algo. 5

Repeat for $s$ stages. Let $p_0$ be the initial point. Letting $R_s = \lceil 24\kappa(\frac{N}{S})^{2/3}\rceil$ this implies that

$$2\mu(\mathbb{E}F(p_s) - F(x^*)) \le \mathbb{E}\|\nabla F(p_s)\|^2 \le \frac{\mu(\mathbb{E}F(p_{s-1}) - F(x^*))}{2} + \frac{2\nu\sigma^2}{SK} \tag{160}$$

Which gives

$$\mathbb{E}F(p_s) - F(x^*) \le \mathcal{O}(\Delta\exp(-s) + \frac{\sigma^2}{\mu SK}) \tag{161}$$

If the total number of rounds is $R$, then

$$\mathbb{E}F(\hat{x}) - F(x^*) \le \mathcal{O}(\Delta\exp(-\frac{R}{\kappa}) + \frac{\sigma^2}{\mu SK}) \tag{162}$$

$\square$

## D.4    SSNM

Note that our usual assumption $F_i(x)$ is $\mu$-strongly convex can be straightforwardly converted into the assumption that our losses are $F_i(x) = \tilde{F}_i(x) + h(x)$ where $\tilde{F}_i(x)$ is merely convex and $h(x)$ is $\mu$-strongly convex (see (Zhou et al., 2019) section 4.2).

**Theorem D.5.** *Suppose that client objectives $F_i$'s and their gradient queries satisfy Assumption B.4 and Assumption B.6. Then running Algo. 6 gives the following for returned iterate $\hat{x}$:*

- ***Strongly convex:*** *$F_i$'s satisfy Assumption B.1 for $\mu > 0$. If we return the final iterate and set $\eta = \frac{1}{2\mu(N/S)}, \tau = \frac{(N/S)\eta\mu}{1+\eta\mu}$ if $\frac{(N/S)}{\kappa} > \frac{3}{4}$ and $\eta = \sqrt{\frac{1}{3\mu(N/S)\beta}}, \tau = \frac{(N/S)\eta\mu}{1+\eta\mu}$ if $\frac{(N/S)}{\kappa} \le \frac{3}{4}$,*

$$\mathbb{E}F(x^{(R)}) - F(x^*) \le \mathcal{O}(\kappa\Delta\exp(-\min\{\frac{S}{N}, \sqrt{\frac{S}{N\kappa}}\}R) + \frac{\kappa\sigma^2}{\mu KS})$$

### D.4.1 CONVERGENCE OF SSNM ON STRONGLY CONVEX FUNCTIONS

Most of this proof follows that of (Zhou et al., 2019) Theorem 1.

First, we compute the variance of the update $g^{(r)}$.

$$\mathbb{E}[\|g^{(r)} - \frac{1}{N}\sum_{i=1}^{N}\nabla\tilde{F}_i(y_{i_r}^{(r)})\|^2] \tag{163}$$

$$\leq \mathbb{E}\|\frac{1}{S}\sum_{i_r \in \mathcal{S}_r}\nabla\tilde{F}_{i_r}(y_{i_r}^{(r)}) - \nabla\tilde{F}_{i_r}(\phi_{i_r}^{(r)}) - \frac{1}{N}\sum_{i=1}^{N}(\nabla\tilde{F}_i(y_i^{(r)}) - \nabla\tilde{F}_i(\phi_i^{(r)}))\|^2 + \frac{\nu\sigma^2}{KS} \tag{164}$$

$$\leq \mathbb{E}\|\frac{1}{S}\sum_{i_r \in \mathcal{S}_r}\nabla\tilde{F}_{i_r}(y_{i_r}^{(r)}) - \nabla\tilde{F}_{i_r}(\phi_{i_r}^{(r)})\|^2 + \frac{\nu\sigma^2}{KS} \tag{165}$$

$$\leq 2\beta\mathbb{E}[\frac{1}{S}\sum_{i_r \in \mathcal{S}_r}\tilde{F}_{i_r}(\phi_{i_r}^{(r)}) - \tilde{F}_{i_r}(y_{i_r}^{(r)}) - \langle\nabla\tilde{F}_{i_r}(y_{i_r}^{(r)}), \phi_{i_r}^{(r)} - y_{i_r}^{(r)}\rangle] + \frac{\nu\sigma^2}{KS} \tag{166}$$

$$= 2\beta[\frac{1}{N}\sum_{i=1}^{N}\tilde{F}_i(\phi_i^{(r)}) - \tilde{F}_i(y_i^{(r)}) - \frac{1}{N}\sum_{i=1}^{N}\langle\nabla\tilde{F}_i(y_i^{(r)}), \phi_i^{(r)} - y_i^{(r)}\rangle] + \frac{\nu\sigma^2}{KS} \tag{167}$$

For some constant $\nu$. In the first inequality we separated out the gradient variance, second inequality we use the fact that $\text{Var}(X) \leq \mathbb{E}[X^2]$, third inequality used Assumption B.4, and fourth we took expectation with respect to the sampled clients. From convexity we have that

$$\tilde{F}_{i_r}(y_{i_r}^{(r)}) - \tilde{F}_{i_r}(x^*) \tag{168}$$

$$\leq \langle\nabla\tilde{F}_{i_r}(y_{i_r}^{(r)}), y_{i_r}^{(r)} - x^*\rangle \tag{169}$$

$$= \frac{1-\tau}{\tau}\langle\nabla\tilde{F}_{i_r}(y_{i_r}^{(r)}), \phi_{i_r}^{(r)} - y_{i_r}^{(r)}\rangle + \langle\nabla\tilde{F}_{i_r}(y_{i_r}^{(r)}), x^{(r)} - x^*\rangle \tag{170}$$

$$= \frac{1-\tau}{\tau}\langle\nabla\tilde{F}_{i_r}(y_{i_r}^{(r)}), \phi_{i_r}^{(r)} - y_{i_r}^{(r)}\rangle + \langle\nabla\tilde{F}_{i_r}(y_{i_r}^{(r)}) - g^{(r)}, x^{(r)} - x^*\rangle \tag{171}$$

$$+ \langle g^{(r)}, x^{(r)} - x^{(r+1)}\rangle + \langle g^{(r)}, x^{(r+1)} - x^*\rangle \tag{172}$$

where the second to last inequality comes from the definition that $y_{i_r}^{(r)} = \tau x^{(r)} + (1-\tau)\phi_{i_r}^{(r)}$. Taking expectation with respect to the sampled clients, we have

$$\frac{1}{N}\sum_{i=1}^{N}\tilde{F}_i(y_i^{(r)}) - \tilde{F}(x^*) \leq \frac{1-\tau}{\tau N}\sum_{i=1}^{N}\langle\nabla\tilde{F}_i(y_i^{(r)}), \phi_i^{(r)} - y_i^{(r)}\rangle + \mathbb{E}_{\mathcal{S}_r}\langle g^{(r)}, x^{(r)} - x^{(r+1)}\rangle \tag{173}$$

$$+ \mathbb{E}_{\mathcal{S}_r}\langle g^{(r)}, x^{(r+1)} - x^*\rangle \tag{174}$$

For $\mathbb{E}_{\mathcal{S}_r}\langle g^{(r)}, x^{(r)} - x^{(r+1)}\rangle$, we can employ smoothness at $(\phi_{I_r}^{(r+1)}, y_{I_r}^{(r)})$, which holds for any $I_r \in \mathcal{S}_r'$:

$$\tilde{F}_{I_r}(\phi_{I_r}^{(r+1)}) - \tilde{F}_{I_r}(y_{I_r}^{(r)}) \leq \langle\nabla\tilde{F}_{I_r}(y_{I_r}^{(r)}), \phi_{I_r}^{(r+1)} - y_{I_r}^{(r)}\rangle + \frac{\beta}{2}\|\phi_{I_r}^{(r+1)} - y_{I_r}^{(r)}\|^2 \tag{175}$$

using $\phi_{I_r}^{(r+1)} = \tau x^{(r+1)} + (1-\tau)\phi_{I_r}^{(r)}$ and $y_{I_r}^{(r)} = \tau x^{(r)} + (1-\tau)\phi_{I_r}^{(r)}$ (though the second is never explicitly computed and only implicitly exists)

$$\tilde{F}_{I_r}(\phi_{I_r}^{(r+1)}) - \tilde{F}_{I_r}(y_{I_r}^{(r)}) \leq \tau\langle\nabla\tilde{F}_{I_r}(y_{I_r}^{(r)}), x^{(r+1)} - x^{(r)}\rangle + \frac{\beta\tau^2}{2}\|x^{(r+1)} - x^{(r)}\|^2 \tag{176}$$

Taking expectation over $\mathcal{S}_r'$ and using $\phi_{I_r}^{(r+1)} = \tau x^{(r+1)} + (1-\tau)\phi_{I_r}^{(r)}$ and $y_{I_r}^{(r)} = \tau x^{(r)} + (1-\tau)\phi_{I_r}^{(r)}$ we see that

$$\mathbb{E}_{\mathcal{S}_r'}[\frac{1}{S}\sum_{I_r \in \mathcal{S}_r'}\tilde{F}_{I_r}(\phi_{I_r}^{(r+1)})] - \frac{1}{N}\sum_{i=1}^{N}\tilde{F}_i(y_i^{(r)}) \leq \tau\langle\frac{1}{N}\sum_{i=1}^{N}\nabla\tilde{F}_i(y_i^{(r)}), x^{(r+1)} - x^{(r)}\rangle \tag{177}$$

$$+ \frac{\beta\tau^2}{2}\|x^{(r+1)} - x^{(r)}\|^2 \tag{178}$$

and rearranging,

$$\langle g^{(r)}, x^{(r)} - x^{(r+1)} \rangle \tag{179}$$

$$\leq \frac{1}{\tau N} \sum_{i=1}^{N} \tilde{F}_i(y_i^{(r)}) - \frac{1}{\tau S} \mathbb{E}_{\mathcal{S}_r'}[\sum_{I_r \in \mathcal{S}_r'} \tilde{F}_{I_r}(\phi_{I_r}^{(r+1)})] \tag{180}$$

$$+ \langle \frac{1}{N} \sum_{i=1}^{N} \nabla \tilde{F}_i(y_i^{(r)}) - g^{(r)}, x^{(r+1)} - x^{(r)} \rangle + \frac{\beta\tau}{2} \|x^{(r+1)} - x^{(r)}\|^2 \tag{181}$$

Substituting this into Eq. (173) after taking expectation over $\mathcal{S}_r$, and observing that from (Zhou et al., 2019, Lemma 2) we have identity

$$\langle g^{(r)}, x^{(r+1)} - u \rangle \leq -\frac{1}{2\eta} \|x^{(r+1)} - x^{(r)}\|^2 + \frac{1}{2\eta} \|x^{(r)} - u\|^2 \tag{182}$$

$$- \frac{1 + \eta\mu}{2\eta} \|x^{(r+1)} - u\|^2 + h(u) - h(x^{(r+1)}) \tag{183}$$

so with $u = x^*$, we get

$$\frac{1}{N} \sum_{i=1}^{N} \tilde{F}_i(y_i^{(r)}) - \tilde{F}(x^*) \tag{184}$$

$$\leq \frac{1 - \tau}{\tau N} \sum_{i=1}^{N} \langle \nabla \tilde{F}_i(y_i^{(r)}), \phi_i^{(r)} - y_i^{(r)} \rangle + \frac{1}{\tau N} \sum_{i=1}^{N} \tilde{F}_i(y_i^{(r)}) \tag{185}$$

$$- \frac{1}{\tau S} \mathbb{E}_{\mathcal{S}_r, \mathcal{S}_r'}[\sum_{I_r \in \mathcal{S}_r'} \tilde{F}_{I_r}(\phi_{I_r}^{(r+1)})] \tag{186}$$

$$+ \mathbb{E}_{\mathcal{S}_r} \langle \frac{1}{N} \sum_{i=1}^{N} \nabla \tilde{F}_i(y_i^{(r)}) - g^{(r)}, x^{(r+1)} - x^{(r)} \rangle + \frac{\beta\tau}{2} \mathbb{E}_{\mathcal{S}_r} \|x^{(r+1)} - x^{(r)}\|^2 \tag{187}$$

$$- \frac{1}{2\eta} \mathbb{E}_{\mathcal{S}_r} \|x^{(r+1)} - x^{(r)}\|^2 + \frac{1}{2\eta} \|x^{(r)} - x^*\|^2 - \frac{1 + \eta\mu}{2\eta} \mathbb{E}_{\mathcal{S}_r} \|x^{(r+1)} - x^*\|^2 \tag{188}$$

$$+ h(x^*) - \mathbb{E}_{\mathcal{S}_r} h(x^{(r+1)}) \tag{189}$$

We use the constraint that $\beta\tau \leq \frac{1}{\eta} - \frac{\beta\tau}{1-\tau}$ plus Young's inequality $\langle a, b \rangle \leq \frac{1}{2c}\|a\|^2 + \frac{c}{2}\|b\|^2$ with $c = \frac{\beta\tau}{1-\tau}$ on $\mathbb{E}_{\mathcal{S}_r} \langle \frac{1}{N} \sum_{i=1}^{N} \nabla \tilde{F}_i(y_i^{(r)}) - g^{(r)}, x^{(r+1)} - x^{(r)} \rangle$ to get

$$\frac{1}{N} \sum_{i=1}^{N} \tilde{F}_i(y_i^{(r)}) - \tilde{F}(x^*) \tag{190}$$

$$\leq \frac{1 - \tau}{\tau N} \sum_{i=1}^{N} \langle \nabla \tilde{F}_i(y_i^{(r)}), \phi_i^{(r)} - y_i^{(r)} \rangle \tag{191}$$

$$+ \frac{1}{\tau N} \sum_{i=1}^{N} \tilde{F}_i(y_i^{(r)}) - \frac{1}{\tau S} \mathbb{E}_{\mathcal{S}_r, \mathcal{S}_r'}[\sum_{I_r \in \mathcal{S}_r'} \tilde{F}_{I_r}(\phi_{I_r}^{(r+1)})] \tag{192}$$

$$+ \frac{1 - \tau}{2\beta\tau} \mathbb{E}_{\mathcal{S}_r} \|\frac{1}{N} \sum_{i=1}^{N} \nabla \tilde{F}_i(y_i^{(r)}) - g^{(r)}\|^2 + \frac{1}{2\eta} \|x^{(r)} - x^*\|^2 \tag{193}$$

$$- \frac{1 + \eta\mu}{2\eta} \mathbb{E}_{\mathcal{S}_r} \|x^{(r+1)} - x^*\|^2 + h(x^*) - \mathbb{E}_{\mathcal{S}_r} h(x^{(r+1)}) \tag{194}$$

using the bound on variance, we get

$$\frac{1}{N} \sum_{i=1}^{N} \tilde{F}_i(y_i^{(r)}) - \tilde{F}(x^*) \tag{195}$$

$$\leq \frac{1}{\tau N} \sum_{i=1}^{N} \tilde{F}_i(y_i^{(r)}) - \frac{1}{\tau S} \mathbb{E}_{\mathcal{S}_r, \mathcal{S}_r'} \Big[ \sum_{I_r \in \mathcal{S}_r'} \tilde{F}_{I_r}(\phi_{I_r}^{(r+1)}) \Big] + \frac{1-\tau}{\tau N} \sum_{i=1}^{N} \tilde{F}_i(\phi_i^{(r)}) - \tilde{F}_i(y_i^{(r)}) \tag{196}$$

$$+ \frac{1}{2\eta} \|x^{(r)} - x^*\|^2 - \frac{1+\eta\mu}{2\eta} \mathbb{E}_{\mathcal{S}_r} \|x^{(r+1)} - x^*\|^2 \tag{197}$$

$$+ h(x^*) - \mathbb{E}_{\mathcal{S}_r} h(x^{(r+1)}) + \frac{(1-\tau)\nu\sigma^2}{2\beta\tau KS} \tag{198}$$

combining terms,

$$\frac{1}{\tau S} \mathbb{E}_{\mathcal{S}_r, \mathcal{S}_r'} \Big[ \sum_{I_r \in \mathcal{S}_r'} \tilde{F}_{I_r}(\phi_{I_r}^{(r+1)}) \Big] - F(x^*) \tag{199}$$

$$\leq \frac{1-\tau}{\tau N} \sum_{i=1}^{N} \tilde{F}_i(\phi_i^{(r)}) + \frac{1}{2\eta} \|x^{(r)} - x^*\|^2 - \frac{1+\eta\mu}{2\eta} \mathbb{E}_{\mathcal{S}_r} \|x^{(r+1)} - x^*\|^2 \tag{200}$$

$$- \mathbb{E}_{\mathcal{S}_r} h(x^{(r+1)}) + \frac{(1-\tau)\nu\sigma^2}{2\beta\tau KS} \tag{201}$$

Using convexity of $h$ and $\phi_{I_k}^{(r+1)} = \tau x^{(r+1)} + (1-\tau)\phi_{I_k}^{(r)}$ for $I_k \in \mathcal{S}_r'$,

$$h(\phi_{I_k}^{(r+1)}) \leq \tau h(x^{(r+1)}) + (1-\tau) h(\phi_{I_k}^{(r)}) \tag{202}$$

After taking expectation with respect to $\mathcal{S}_r$ and $\mathcal{S}_r'$,

$$-\mathbb{E}_{\mathcal{S}_r}[h(x^{(r+1)})] \leq \frac{1-\tau}{\tau N} \sum_{i=1}^{N} h(\phi_i^{(r)}) - \frac{1}{\tau} \mathbb{E}_{\mathcal{S}_r, \mathcal{S}_r'} \Big[ \frac{1}{S} \sum_{I_r \in \mathcal{S}_r'} h(\phi_{I_k}^{(r+1)}) \Big] \tag{203}$$

and plugging this back in, multiplying by $S/N$ on both sides, and adding both sides by $\frac{1}{\tau N} \mathbb{E}_{\mathcal{S}_r'} [\sum_{i \notin \mathcal{S}_r'} (F_i(\phi_i^{(r)}) - F_i(x^*))]$

$$\frac{1}{\tau} \mathbb{E}_{\mathcal{S}_r, \mathcal{S}_r'} \Big[ \frac{1}{N} \sum_{i=1}^{N} F_i(\phi_i^{(r+1)}) - F_i(x^*) \Big] \tag{204}$$

$$\leq \frac{(1-\tau)S}{\tau N} \Big( \frac{1}{N} \sum_{i=1}^{N} F_i(\phi_i^{(r)}) - F_i(x^*) \Big) + \frac{1}{\tau N} \mathbb{E}_{\mathcal{S}_r'} \Big[ \sum_{i \notin \mathcal{S}_r'} (F_i(\phi_i^{(r)}) - F_i(x^*)) \Big] \tag{205}$$

$$+ \frac{S}{2\eta N} \|x^{(r)} - x^*\|^2 - \frac{(1+\eta\mu)S}{2\eta N} \mathbb{E}_{\mathcal{S}_r} \|x^{(r+1)} - x^*\|^2 + \frac{(1-\tau)\nu\sigma^2}{2\beta\tau KN} \tag{206}$$

Observe that the probability of choosing any client index happens with probability $S/N$, so

$$\frac{1}{\tau N} \mathbb{E}_{\mathcal{S}_r'} \Big[ \sum_{i \notin \mathcal{S}_r'} (F_i(\phi_i^{(r)}) - F_i(x^*)) \Big] = \frac{N-S}{\tau N} \Big( \frac{1}{N} \sum_{i=1}^{N} F_i(\phi_i^{(r)}) - F_i(x^*) \Big) \tag{207}$$

which implies

$$\frac{1}{\tau} \mathbb{E}_{\mathcal{S}_r, \mathcal{S}_r'} \Big[ \frac{1}{N} \sum_{i=1}^{N} F_i(\phi_i^{(r+1)}) - F_i(x^*) \Big] \tag{208}$$

$$\leq \frac{1 - \frac{\tau S}{N}}{\tau} \Big( \frac{1}{N} \sum_{i=1}^{N} F_i(\phi_i^{(r)}) - F_i(x^*) \Big) \tag{209}$$

$$+ \frac{S}{2\eta N} \|x^{(r)} - x^*\|^2 - \frac{(1+\eta\mu)S}{2\eta N} \mathbb{E}_{\mathcal{S}_r} \|x^{(r+1)} - x^*\|^2 + \frac{(1-\tau)\nu\sigma^2}{2\beta\tau KN} \tag{210}$$

To complete our Lyapunov function so the potential is always positive, we need another term:

$$-\frac{1}{N}\sum_{i=1}^{N}\langle\nabla F_i(x^*),\phi_i^{(r+1)}-x^*\rangle \tag{211}$$

$$=-\frac{1}{N}\sum_{I_r\in\mathcal{S}_r'}\langle\nabla F_{I_r}(x^*),\phi_{I_r}^{(r+1)}-x^*\rangle-\frac{1}{N}\sum_{j\notin\mathcal{S}_r'}\langle\nabla F_j(x^*),\phi_j^{(r)}-x^*\rangle \tag{212}$$

$$=\sum_{I_r\in\mathcal{S}_r'}-\frac{\tau}{N}\langle\nabla F_{I_r}(x^*),x^{(r+1)}-x^*\rangle+\frac{\tau}{N}\langle\nabla F_{I_r}(x^*),\phi_{I_r}^{(r)}-x^*\rangle \tag{213}$$

$$-\frac{1}{N}\sum_{i=1}^{N}\langle\nabla F_i(x^*),\phi_i^{(r)}-x^*\rangle \tag{214}$$

Taking expectation with respect to $\mathcal{S}_r,\mathcal{S}_r'$,

$$\mathbb{E}_{\mathcal{S}_r,\mathcal{S}_r'}[-\frac{1}{N}\sum_{i=1}^{N}\langle\nabla F_i(x^*),\phi_i^{(r+1)}-x^*\rangle]=-(1-\frac{\tau S}{N})(\frac{1}{N}\sum_{i=1}^{N}\langle\nabla F_i(x^*),\phi_i^{(r)}-x^*\rangle) \tag{215}$$

Let $B_r:=\frac{1}{N}\sum_{i=1}^{N}F_i(\phi_i^{(r)})-F(x^*)-\frac{1}{N}\sum_{i=1}^{N}\langle\nabla F_i(x^*),\phi_i^{(r)}-x^*\rangle$ and $P_r:=\|x^{(r)}-x^*\|^2$, then we can write

$$\frac{1}{\tau}\mathbb{E}_{\mathcal{S}_r,\mathcal{S}_r'}[B_{r+1}]+\frac{(1+\eta\mu)S}{2\eta N}\mathbb{E}_{\mathcal{S}_r}[P_{r+1}]\leq\frac{1-\frac{\tau S}{N}}{\tau}B_r+\frac{S}{2\eta N}P_r+\frac{(1-\tau)\nu\sigma^2}{2\beta\tau KN} \tag{216}$$

**Case 1:** Suppose that $\frac{(N/S)}{\kappa}\leq\frac{3}{4}$, then choosing $\eta=\sqrt{\frac{1}{3\mu(N/S)\beta}},\tau=\frac{(N/S)\eta\mu}{1+\eta\mu}=\frac{\sqrt{\frac{(N/S)}{3\kappa}}}{1+\sqrt{\frac{1}{3(N/S)\kappa}}}<\frac{1}{2}$, we evaluate the parameter constraint $\beta\tau\leq\frac{1}{\eta}-\frac{\beta\tau}{1-\tau}$:

$$\beta\tau\leq\frac{1}{\eta}-\frac{\beta\tau}{1-\tau}\implies(1+\frac{1}{1-\tau})\tau\leq\frac{1}{\beta\eta}\implies\frac{2-\tau}{1-\tau}\frac{\sqrt{\frac{(N/S)}{3\kappa}}}{1+\sqrt{\frac{1}{3(N/S)\kappa}}}\leq\sqrt{\frac{3(N/S)}{\kappa}} \tag{217}$$

which shows our constraint is satisfied. We also know that

$$\frac{1}{\tau(1+\eta\mu)}=\frac{1-\frac{\tau S}{N}}{\tau}=\frac{1}{(N/S)\eta\mu} \tag{218}$$

So we can write

$$\frac{1}{(N/S)\eta\mu}\mathbb{E}_{\mathcal{S}_r,\mathcal{S}_r'}[B_{r+1}]+\frac{1}{2\eta(N/S)}\mathbb{E}_{\mathcal{S}_r}[P_{r+1}] \tag{219}$$

$$\leq(1+\eta\mu)^{-1}(\frac{1}{(N/S)\eta\mu}B_r+\frac{1}{2\eta(N/S)}P_r)+\frac{(1-\tau)\nu\sigma^2}{2\beta\tau KN} \tag{220}$$

Telescoping the contraction and taking expectation with respect to all randomness, we have

$$\frac{1}{(N/S)\eta\mu}\mathbb{E}[B_R]+\frac{1}{2\eta(N/S)}\mathbb{E}[P_R] \tag{221}$$

$$\leq(1+\eta\mu)^{-R}(\frac{1}{(N/S)\eta\mu}\mathbb{E}B_0+\frac{1}{2\eta(N/S)}\mathbb{E}P_0)+\frac{(1-\tau)\nu\sigma^2}{2\beta\tau KN}(\frac{1+\eta\mu}{\eta\mu}) \tag{222}$$

$B_0=F(x^{(0)})-F(x^*)$ and $\mathbb{E}B_R\geq0$ based on convexity. Next, we calculate the coefficient of the variance term.

$$\frac{1-\tau}{\tau}\frac{1+\eta\mu}{\eta\mu}=(\frac{1}{\tau}-1)(1+\frac{1}{\eta\mu})=(\frac{S}{N}(1+\frac{1}{\eta\mu})-1)(1+\frac{1}{\eta\mu}) \tag{223}$$

$$=\mathcal{O}((\frac{S}{N}(1+\sqrt{\kappa}(\sqrt{\frac{N}{S}}))-1)(1+\sqrt{\kappa}(\sqrt{\frac{N}{S}}))) \tag{224}$$

$$=\mathcal{O}((\frac{S}{N}+\sqrt{\kappa}(\frac{S}{N})^{1/2}-1)(1+\sqrt{\kappa}(\frac{N}{S})^{1/2})) \tag{225}$$

$$=\mathcal{O}(\kappa) \tag{226}$$

Substituting the parameter choices and using strong convexity,

$$\mathbb{E}\|x^{(R)} - x^*\|^2 \le (1 + \sqrt{\frac{1}{3(N/S)\kappa}})^{-R}(\frac{2}{\mu}\Delta + D^2) + \mathcal{O}(\frac{\sigma^2}{\mu^2 KN}) \tag{227}$$

**Case 2:** On the other hand, we can have $\frac{(N/S)}{\kappa} > \frac{3}{4}$ and choose $\eta = \frac{1}{2\mu(N/S)}$, $\tau = \frac{(N/S)\eta\mu}{1+\eta\mu} = \frac{(1/2)}{1+\frac{1}{2(N/S)}} < \frac{1}{2}$. One can check that the constraint $\beta\tau \le \frac{1}{\eta} - \frac{\beta\tau}{1-\tau}$ is satisfied.

After telescoping and taking expectation,

$$2\mathbb{E}[B_R] + \frac{1}{2\eta(N/S)}\mathbb{E}[P_R] \tag{228}$$

$$\le (1 + \eta\mu)^{-R}(2\mathbb{E}B_0 + \frac{1}{2\eta(N/S)}\mathbb{E}P_0) + \frac{(1-\tau)\nu\sigma^2}{2\beta\tau KN}(\frac{1+\eta\mu}{\eta\mu}) \tag{229}$$

Next, we calculate the coefficient of the variance term.

$$\frac{1-\tau}{\tau}\frac{1+\eta\mu}{\eta\mu} = (\frac{1}{\tau} - 1)(1 + \frac{1}{\eta\mu}) = (\frac{S}{N}(1 + \frac{1}{\eta\mu}) - 1)(1 + \frac{1}{\eta\mu}) \tag{230}$$

$$= \mathcal{O}((\frac{S}{N}(1 + \frac{N}{S}) - 1)(1 + \frac{N}{S})) \tag{231}$$

$$= \mathcal{O}(\frac{N}{S}) \tag{232}$$

which gives us

$$\mathbb{E}\|x^{(R)} - x^*\|^2 \le (1 + \frac{1}{2(N/S)})^{-R}(\frac{2}{\mu}\Delta + D^2) + \mathcal{O}(\frac{\sigma^2}{\beta\mu KS}) \tag{233}$$

Altogether, supposing we choose our parameters as stated in the two cases, we have:

$$\mathbb{E}F(x^{(R)}) - F(x^*) \le \mathcal{O}(\kappa\Delta\exp(-\min\{\frac{S}{N}, \sqrt{\frac{S}{N\kappa}}\}R) + \frac{\kappa\sigma^2}{\mu KS}) \tag{234}$$

# E  PROOFS FOR LOCAL UPDATE METHODS

## E.1  FEDAVG

**Theorem E.1.** *Suppose that client objectives $F_i$'s and their gradient queries satisfy Assumption B.4 and Assumption B.6. Then running Algo. 4 gives the following for returned iterate $\hat{x}$:*

- ***Strongly convex:*** *$F_i$'s satisfy Assumption B.1 for $\mu > 0$. If we return the final iterate, set $\eta = \frac{1}{\beta}$, and sample $S$ clients per round arbitrarily,*

$$\mathbb{E}F(x^{(R)}) - F(x^*) \le \mathcal{O}(\Delta\exp(-\frac{R\sqrt{K}}{\kappa}) + \frac{\zeta^2}{\mu} + \frac{\sigma^2}{\mu\sqrt{K}})$$

*Further, if there is no gradient variance and only one client $i$ is sampled the entire algorithm,*

$$\|x^{(r)} - x^*\|^2 \le \mathcal{O}(\|x^{(0)} - x^*\|^2 + \|x^{(0)} - x_i^*\|^2) \text{ for any } 0 \le r \le R$$

- ***General convex:*** *$F_i$'s satisfy Assumption B.2. If we return the last iterate after running the algorithm on $F_\mu(x) = F(x) + \frac{\mu}{2}\|x^{(0)} - x\|^2$ with $\mu = \Theta(\max\{\frac{\beta}{R^2}\log^2(e^2 + R^2), \frac{\zeta}{D}, \frac{\sigma}{DK^{1/4}}\})$, and $\eta = \frac{1}{\beta+\mu}$,*

$$\mathbb{E}F(x^{(R)}) - F(x^*) \le \tilde{\mathcal{O}}(\frac{\beta D^2}{\sqrt{K}R} + \frac{\sigma D}{K^{1/4}} + \zeta D)$$

*and for any $0 \le r \le R$,*

$$\mathbb{E}\|x^{(r)} - x^*\|^2 \le \tilde{\mathcal{O}}(D^2)$$

- **PL condition:** $F_i$'s satisfy Assumption B.3 for $\mu > 0$. *If we return the final iterate, set* $\eta = \frac{1}{\beta}$, *and sample one client per round,*

$$\mathbb{E}F(x^{(R)}) - F(x^*) \leq \mathcal{O}(\Delta \exp(-\frac{R\sqrt{K}}{\kappa}) + \frac{\zeta^2}{2\mu} + \frac{\sigma^2}{2\mu\sqrt{K}})$$

### E.1.1 CONVERGENCE OF FEDAVG FOR STRONGLY CONVEX FUNCTIONS

*Proof.* By smoothness of F (Assumption B.4), we have for $k \in \{0, \ldots, \sqrt{K} - 1\}$

$$F(x_{i,k+1}^{(r)}) - F(x_{i,k}^{(r)}) \leq -\eta \langle \nabla F(x_{i,k}^{(r)}), g_{i,k}^{(r)} \rangle + \frac{\beta\eta^2}{2}\|g_{i,k}^{(r)}\|^2 \tag{235}$$

Using the fact that for any $a, b$ we have $-2ab = (a-b)^2 - a^2 - b^2$,

$$F(x_{i,k+1}^{(r)}) - F(x_{i,k}^{(r)}) \leq -\frac{\eta}{2}\|\nabla F(x_{i,k}^{(r)})\|^2 + \frac{\beta\eta^2 - \eta}{2}\|g_{i,k}^{(r)}\|^2 + \frac{\eta}{2}\|g_{i,k}^{(r)} - \nabla F(x_{i,k}^{(r)})\|^2 \tag{236}$$

Letting $\eta \leq \frac{1}{\beta}$,

$$F(x_{k+1}^{(r)}) - F(x_{i,k}^{(r)}) \leq -\frac{\eta}{2}\|\nabla F(x_{i,k}^{(r)})\|^2 + \frac{\eta}{2}\|g_{i,k}^{(r)} - \nabla F(x_{i,k}^{(r)})\|^2 \tag{237}$$

Conditioning on everything up to the $k$-th step of the $r$-th round,

$$\mathbb{E}_{r,k}F(x_{i,k+1}^{(r)}) - F(x_{i,k}^{(r)}) \leq -\frac{\eta}{2}\|\nabla F(x_{i,k}^{(r)})\|^2 + \frac{\eta}{2}\mathbb{E}_{r,k}\|g_{i,k}^{(r)} - \nabla F(x_{i,k}^{(r)})\|^2 \tag{238}$$

$$\leq -\frac{\eta}{2}\|\nabla F(x_{i,k}^{(r)})\|^2 + \frac{\eta\zeta^2}{2} + \frac{\eta\sigma^2}{2\sqrt{K}} \tag{239}$$

Where the last step used the fact that $\mathbb{E}[X^2] = \text{Var}(X) + \mathbb{E}[X]$, the assumption on heterogeneity (Assumption B.5), and the assumption on gradient variance (Assumption B.6) along with the fact that $g_{i,k}^{(r)}$ is an average over $\sqrt{K}$ client gradient queries. Next, using $\mu$-strong convexity of $F$ (Assumption B.1),

$$\mathbb{E}_{r,k}F(x_{i,k+1}^{(r)}) - F(x_{i,k}^{(r)}) \leq -\eta\mu(F(x_{i,k}^{(r)}) - F(x^*)) + \frac{\eta\zeta^2}{2} + \frac{\eta\sigma^2}{2\sqrt{K}} \tag{240}$$

which after taking full expectation gives

$$\mathbb{E}F(x_{i,k+1}^{(r)}) - F(x^*) \leq (1 - \eta\mu)(\mathbb{E}F(x_{i,k}^{(r)}) - F(x^*)) + \frac{\eta\zeta^2}{2} + \frac{\eta\sigma^2}{2\sqrt{K}} \tag{241}$$

Unrolling the recursion over $k$ we get

$$\mathbb{E}F(x_{i,K}^{(r)}) - F(x^*) \tag{242}$$

$$\leq (1 - \eta\mu)^{\sqrt{K}}(\mathbb{E}F(x_{i,0}^{(r)}) - F(x^*)) + (\frac{\eta\zeta^2}{2} + \frac{\eta\sigma^2}{2\sqrt{K}})\sum_{k=0}^{\sqrt{K}-1}(1 - \eta\mu)^k \tag{243}$$

Also note that $x^{(r+1)} = \frac{1}{S}\sum_{i \in \mathcal{S}_r} x_{i,\sqrt{K}}^{(r)}$. So by convexity of $F$,

$$\mathbb{E}F(x^{(r+1)}) - F(x^*) \tag{244}$$

$$\leq \frac{1}{S}\sum_{i \in \mathcal{S}_r}\mathbb{E}F(x_{i,\sqrt{K}}^{(r)}) - F(x^*) \tag{245}$$

$$\leq (1 - \eta\mu)^{\sqrt{K}}(\mathbb{E}F(x^{(r)}) - F(x^*)) + (\frac{\eta\zeta^2}{2} + \frac{\eta\sigma^2}{2\sqrt{K}})\sum_{k=0}^{\sqrt{K}-1}(1 - \eta\mu)^k \tag{246}$$

Unrolling the recursion over $R$, we get

$$\mathbb{E}F(x^{(r)}) - F(x^*) \tag{247}$$

$$\leq (1 - \eta\mu)^{r\sqrt{K}}(F(x^{(0)}) - F(x^*)) + (\frac{\eta\zeta^2}{2} + \frac{\eta\sigma^2}{2\sqrt{K}})\sum_{\tau=0}^{r-1}\sum_{k=0}^{\sqrt{K}-1}(1 - \eta\mu)^{\tau\sqrt{K}+k} \tag{248}$$

Which can be upper bounded as

$$\mathbb{E}F(x^{(r)}) - F(x^*) \leq (1 - \eta\mu)^{R\sqrt{K}}(F(x^{(0)}) - F(x^*)) + \frac{\zeta^2}{2\mu} + \frac{\sigma^2}{2\mu\sqrt{K}} \tag{249}$$

The final statement comes from the fact that $\|x - \eta\nabla F_i(x) - x_i^*\| \leq \|x - x_i^*\|$ because of $\eta \leq \frac{1}{\beta}$ and applying triangle inequality. $\square$

### E.1.2    CONVERGENCE OF FEDAVG FOR GENERAL CONVEX FUNCTIONS

For the general convex case, we use Nesterov smoothing. Concretely, we will run Algo. 3 assuming strong convexity by optimizing instead a modified objective

$$F_\mu(x) = F(x) + \frac{\mu}{2}\|x^{(0)} - x\|^2 \tag{250}$$

Define $x_\mu^* = \arg\min_x F_\mu(x)$ and $\Delta_\mu = \mathbb{E}F_\mu(x^{(0)}) - F(x_\mu^*)$. We will choose $\mu$ carefully to balance the error introduced by the regularization term and the better convergence properties of having larger $\mu$.

*Proof.* We know that running Algo. 4 with $\eta = \frac{1}{\beta+\mu}$ gives (from the previous proof)

$$\mathbb{E}F_\mu(x^{(R)}) - F_\mu(x_\mu^*) \leq \Delta_\mu \exp(-\frac{R\sqrt{K}}{\frac{\beta+\mu}{\mu}}) + \frac{\zeta^2}{2\mu} + \frac{\sigma^2}{2\mu\sqrt{K}} \tag{251}$$

We have that by (Yuan & Ma, 2020, Proposition E.7)

$$\mathbb{E}F(x^{(R)}) - F(x^*) \leq \mathbb{E}F_\mu(x^{(R)}) - F_\mu(x_\mu^*) + \frac{\mu}{2}D^2 \tag{252}$$

So we have (because $\Delta_\mu \leq \Delta$ as shown in Thm. D.3),

$$\mathbb{E}F(x^{(R)}) - F(x^*) \leq \Delta \exp(-\frac{R\sqrt{K}}{\frac{\beta+\mu}{\mu}}) + \frac{\zeta^2}{2\mu} + \frac{\sigma^2}{2\mu\sqrt{K}} + \frac{\mu}{2}D^2 \tag{253}$$

Then if we choose $\mu \geq \Theta(\frac{\beta}{\sqrt{K}R}\log^2(e^2 + \sqrt{K}R))$, $\mu \geq \Theta(\frac{\zeta}{D})$, and $\mu \geq \Theta(\frac{\sigma}{DK^{1/4}})$,

$$\mathbb{E}F(x^{(R)}) - F(x^*) \leq \tilde{\mathcal{O}}(\frac{\beta D^2}{\sqrt{K}R} + \frac{\sigma D}{K^{1/4}} + \zeta D) \tag{254}$$

Now we show the distance bound. Recall that

$$\mathbb{E}F_\mu(x^{(R)}) - F_\mu(x_\mu^*) \leq \Delta_\mu \exp(-\frac{R\sqrt{K}}{\frac{\beta+\mu}{\mu}}) + \frac{\zeta^2}{2\mu} + \frac{\sigma^2}{2\mu\sqrt{K}} \tag{255}$$

By smoothness, strong convexity of $F_\mu$, and the choice of $\mu$ (as we chose each term above divided by $\mu$ to match $D^2$ up to log factors), we have that

$$\mathbb{E}\|x^{(R)} - x_\mu^*\|^2 \leq \tilde{\mathcal{O}}(D^2) \tag{256}$$

So,

$$\mathbb{E}\|x^{(R)} - x^*\|^2 \leq 3\mathbb{E}\|x^{(R)} - x_\mu^*\|^2 + 3\mathbb{E}\|x^{(0)} - x_\mu^*\|^2 + 3\mathbb{E}\|x^* - x^{(0)}\|^2 \leq \tilde{\mathcal{O}}(D^2) \tag{257}$$

Where the last inequality follows because

$$F(x_\mu^*) + \frac{\mu}{2}\|x^{(0)} - x_\mu^*\|^2 \leq F(x^*) + \frac{\mu}{2}\|x^{(0)} - x^*\|^2 \tag{258}$$

$\square$

### E.1.3    CONVERGENCE OF FEDAVG UNDER THE PL CONDITION

*Proof.* The same proof follows as in the strongly convex case, except Eq. (244), where we avoid having to use convexity by only sampling one client at a time. This follows previous work such as Karimireddy et al. (2020a). Averaging when the functions are not convex can cause the error to blow up, though in practice this is not seen (as mentioned in Karimireddy et al. (2020a)). $\square$

# F PROOFS FOR FEDCHAIN

## F.1 FEDAVG → SGD

### F.1.1 CONVERGENCE OF FEDAVG → SGD ON STRONGLY CONVEX FUNCTIONS

**Theorem F.1.** *Suppose that client objectives $F_i$'s and their gradient queries satisfy Assumption B.4, Assumption B.6, Assumption B.7, Assumption B.5, Assumption B.8. Then running Algo. 1 where $\mathcal{A}_{local}$ is Algo. 4 in the setting of Thm. E.1 and $\mathcal{A}_{global}$ is Algo. 2 in the setting Thm. D.1:*

- **Strongly convex:** $F_i$'s satisfy Assumption B.1 for $\mu > 0$. Then there exists a lower bound of $K$ such that we have the rate

$$\tilde{\mathcal{O}}(\min\{\frac{\zeta^2}{\mu}, \Delta\} \exp(-\frac{R}{\kappa}) + \frac{\sigma^2}{\mu SKR} + (1 - \frac{S}{N})\frac{\zeta^2}{\mu SR} + \sqrt{1 - \frac{S}{N}}\frac{\zeta_F}{\sqrt{S}})$$

- **General convex:** $F_i$'s satisfy Assumption B.2. Then there exists a lower bound of $K$ such that we have the rate

$$\tilde{\mathcal{O}}(\min\{\frac{\beta^{1/2}\zeta^{1/2}D^{3/2}}{R^{1/2}}, \frac{\beta D^2}{R}\} + \frac{\beta^{1/2}\sigma^{1/2}D^{3/2}}{(SKR)^{1/4}}$$
$$+ (1 - \frac{S}{N})^{1/4}\frac{\beta^{1/2}\zeta_F^{1/2}D}{S^{1/4}R^{1/2}} + (1 - \frac{S}{N})^{1/4}\frac{\beta^{1/2}\zeta^{1/2}D^{3/2}}{(SR)^{1/4}})$$

- **PL condition:** $F_i$'s satisfy Assumption B.3 for $\mu > 0$. Then there exists a lower bound of $K$ such that we have the rate

$$\tilde{\mathcal{O}}(\min\{\frac{\zeta^2}{\mu}, \Delta\} \exp(-\frac{R}{\kappa}) + \frac{\kappa\sigma^2}{\mu NKR} + (1 - \frac{S}{N})\frac{\kappa\zeta^2}{\mu SR} + \sqrt{1 - \frac{S}{N}}\frac{\zeta_F}{\sqrt{S}})$$

*Proof.* From Thm. E.1, we know that with $K > \max\{\frac{\sigma^4}{\zeta^4}, \frac{\kappa^2}{R^2}\log^2(\frac{\Delta\mu}{\zeta^2})\}$

$$\mathbb{E}F(\hat{x}_{1/2}) - F(x^*) \leq \mathcal{O}(\frac{\zeta^2}{\mu}) \tag{259}$$

From Lemma H.2, if $K > \frac{\sigma_F^2}{S\min\{\Delta, \frac{\zeta^2}{\mu}\}}$

$$\mathbb{E}F(\hat{x}_1) - F(x^*) \leq \mathcal{O}(\min\{\frac{\zeta^2}{\mu}, \Delta\} + \sqrt{1 - \frac{S-1}{N-1}}\frac{\zeta_F}{\sqrt{S}}) \tag{260}$$

And so from Thm. D.1, we know that

$$\mathbb{E}F(\hat{x}_2) - F(x^*) \tag{261}$$

$$\leq \tilde{\mathcal{O}}(\min\{\frac{\zeta^2}{\mu}, \Delta\} \exp(-\frac{R}{\kappa}) + \frac{\sigma^2}{\mu NKR} + (1 - \frac{S}{N})\frac{\zeta^2}{\mu SR} + \sqrt{1 - \frac{S}{N}}\frac{\zeta_F}{\sqrt{S}})) \tag{262}$$

$\square$

### F.1.2 CONVERGENCE OF FEDAVG → SGD ON GENERAL CONVEX FUNCTIONS

*Proof.* From Thm. E.1, we know that with $K > \max\{\frac{\sigma^4}{\zeta^4}, \frac{\beta^2 D^2}{\zeta R}\}$

$$\mathbb{E}F(x^{(R)}) - F(x^*) \leq \tilde{\mathcal{O}}(\zeta D)$$

From Lemma H.2, if $K > \frac{\sigma_F^2}{S\min\{\Delta, \frac{\zeta^2}{\mu}\}}$

$$\mathbb{E}F(\hat{x}_1) - F(x^*) \leq \tilde{\mathcal{O}}(\min\{\zeta D, \Delta\} + \sqrt{1 - \frac{S-1}{N-1}}\frac{\zeta_F}{\sqrt{S}}) \tag{263}$$

And so from Thm. D.1, we know that

$$\mathbb{E}\|\nabla F(\hat{x}_2)\|^2 \leq \tilde{\mathcal{O}}(\frac{\beta \min\{\zeta D, \Delta\}}{R} + \sqrt{1 - \frac{S}{N}}\frac{\beta \zeta_F}{\sqrt{S}R} + \frac{\beta \sigma D}{\sqrt{SKR}} + \sqrt{1 - \frac{S}{N}}\frac{\beta \zeta D}{\sqrt{SR}}) \quad (264)$$

Next, using that $\mathbb{E}\|\hat{x}_2 - x^*\| \leq \tilde{\mathcal{O}}(D^2)$ from Thm. E.1 and Thm. D.1 as well as convexity,

$$\mathbb{E}F(\hat{x}_2) - F(x^*) \quad (265)$$

$$\leq \sqrt{\mathbb{E}\|\nabla F(\hat{x}_2)\|^2}\sqrt{\mathbb{E}\|\hat{x}_2 - x^*\|^2} \quad (266)$$

$$\leq \tilde{\mathcal{O}}(\frac{\beta^{1/2}\min\{\zeta^{1/2}D^{3/2}, \Delta^{1/2}D\}}{R^{1/2}} \quad (267)$$

$$+ \frac{\beta^{1/2}\sigma^{1/2}D^{3/2}}{(SKR)^{1/4}} + (1 - \frac{S}{N})^{1/4}\frac{\beta^{1/2}\zeta_F^{1/2}D}{S^{1/4}R^{1/2}} + (1 - \frac{S}{N})^{1/4}\frac{\beta^{1/2}\zeta^{1/2}D^{3/2}}{(SR)^{1/4}}) \quad (268)$$

One can pre-run SGD before all of this for a constant fraction of rounds so that (Woodworth et al., 2020a, Section 7):

$$\Delta \leq \tilde{\mathcal{O}}(\frac{\beta D^2}{R} + \frac{\sigma D}{\sqrt{SKR}} + \sqrt{1 - \frac{S}{N}}\frac{\zeta D}{\sqrt{SR}}) \quad (269)$$

This likely not practically necessary § 6. Altogether,

$$\mathbb{E}F(\hat{x}_2) - F(x^*) \quad (270)$$

$$\leq \sqrt{\mathbb{E}\|\nabla F(\hat{x}_2)\|^2}\sqrt{\mathbb{E}\|\hat{x}_2 - x^*\|^2} \quad (271)$$

$$\leq \tilde{\mathcal{O}}(\min\{\frac{\beta^{1/2}\zeta^{1/2}D^{3/2}}{R^{1/2}}, \frac{\beta D^2}{R}\} \quad (272)$$

$$+ \frac{\beta^{1/2}\sigma^{1/2}D^{3/2}}{(SKR)^{1/4}} + (1 - \frac{S}{N})^{1/4}\frac{\beta^{1/2}\zeta_F^{1/2}D}{S^{1/4}R^{1/2}} + (1 - \frac{S}{N})^{1/4}\frac{\beta^{1/2}\zeta^{1/2}D^{3/2}}{(SR)^{1/4}}) \quad (273)$$

$$\square$$

### F.1.3 CONVERGENCE OF FEDAVG → SGD UNDER THE PL-CONDITION

*Proof.* The proof is the same as in the strongly convex case. $\square$

### F.2 FEDAVG → ASG

**Theorem F.2.** *Suppose that client objectives $F_i$'s and their gradient queries satisfy Assumption B.4, Assumption B.6, Assumption B.7, Assumption B.5, Assumption B.8. Then running Algo. 1 where $\mathcal{A}_{local}$ is Algo. 4 in the setting of Thm. E.1 and $\mathcal{A}_{global}$ is Algo. 3 in the setting Thm. D.1:*

- **Strongly convex:** *$F_i$'s satisfy Assumption B.1 for $\mu > 0$. Then there exists a lower bound of $K$ (same as in FedAvg → SGD) such that we have the rate*

$$\tilde{\mathcal{O}}(\min\{\frac{\zeta^2}{\mu}, \Delta\}\exp(-\frac{R}{\sqrt{\kappa}}) + \frac{\sigma^2}{\mu SKR} + (1 - \frac{S}{N})\frac{\zeta^2}{\mu SR} + \sqrt{1 - \frac{S}{N}}\frac{\zeta_F}{\sqrt{S}})$$

- **General convex:** *$F_i$'s satisfy Assumption B.2. Then there exists a lower bound of $K$ (same as in FedAvg → SGD) such that we have the rate*

$$\tilde{\mathcal{O}}(\min\{\frac{\beta^{1/2}\zeta^{1/2}D^{3/2}}{R}, \frac{\beta D^2}{R^2}\} + \frac{\beta^{1/2}\sigma^{1/2}D^{3/2}}{(SKR)^{1/4}} + \frac{\sigma D}{(SKR)^{1/2}} + (1 - \frac{S}{N})^{1/2}\frac{\zeta D}{(SR)^{1/2}}$$

$$+ (1 - \frac{S}{N})^{1/4}\frac{\beta^{1/2}\zeta_F^{1/2}D}{S^{1/4}R^{1/2}} + (1 - \frac{S}{N})^{1/4}\frac{\beta^{1/2}\zeta^{1/2}D^{3/2}}{(SR)^{1/4}})$$

Proof follows those of FedAvg → SGD (Thm. F.1).

### F.3 FEDAVG → SAGA

**Theorem F.3.** *Suppose that client objectives $F_i$'s and their gradient queries satisfy Assumption B.4, Assumption B.6, Assumption B.7, Assumption B.5, Assumption B.8. Then running Algo. 1 where $\mathcal{A}_{local}$ is Algo. 4 in the setting of Thm. E.1 and $\mathcal{A}_{global}$ is Algo. 5 in the setting Thm. D.4:*

- **Strongly convex:** *$F_i$'s satisfy Assumption B.1 for $\mu > 0$. Then there exists a lower bound of $K$ (same as in FedAvg → SGD) such that we have the rate*

$$\tilde{\mathcal{O}}(\min\{\frac{\zeta^2}{\mu}, \Delta\} \exp(-\max\{\frac{N}{S}, \kappa\}^{-1}R) + \frac{\sigma^2}{\mu SKR})$$

- **PL condition:** *$F_i$'s satisfy Assumption B.3 for $\mu > 0$. Then there exists a lower bound of $K$ such that we have the rate*

$$\tilde{\mathcal{O}}(\min\{\frac{\zeta^2}{\mu}, \Delta\} \exp(-(\kappa(\frac{N}{S})^{2/3})^{-1}R) + \frac{\sigma^2}{\mu SK})$$

Proof follows those of FedAvg → SGD (Thm. F.1) and using Lemma H.2 with $S = N$ as the algorithm already requires $R > \frac{N}{S}$.

### F.4 FEDAVG → SSNM

**Theorem F.4.** *Suppose that client objectives $F_i$'s and their gradient queries satisfy Assumption B.4, Assumption B.6, Assumption B.5, Assumption B.8. Then running Algo. 1 where $\mathcal{A}_{local}$ is Algo. 4 in the setting of Thm. E.1 and $\mathcal{A}_{global}$ is Algo. 6 in the setting Thm. D.5:*

- **Strongly convex:** *$F_i$'s satisfy Assumption B.1 for $\mu > 0$. Then there exists a lower bound of $K$ (same as in FedAvg → SGD) such that we have the rate*

$$\tilde{\mathcal{O}}(\min\{\frac{\zeta^2}{\mu}, \Delta\} \exp(-\max\{\frac{N}{S}, \sqrt{\kappa(\frac{N}{S})}\}^{-1}R) + \frac{\kappa\sigma^2}{\mu KS})$$

Proof follows those of FedAvg → SGD (Thm. F.1) and using Lemma H.2 with $S = N$ as the algorithm already requires $R > \frac{N}{S}$.

## G LOWER BOUND

In this section we prove the lower bound. All gradients will be noiseless, and we will allow full communication to all clients every round. There will be only two functions in this lower bound: $F_1$ and $F_2$. If $N > 2$, then $F_1$ is assigned to the first $\lfloor N/2 \rfloor$ clients and $F_2$ to the next $\lfloor N/2 \rfloor$ clients. If there is an odd number of machines we let the last machine be $F_3(x) = \frac{\mu}{2}\|x\|^2$. This only reduces the lower bound by a factor of at most $\frac{N-1}{N}$. So we look at the case $N = 2$.

Let $\ell_2, C, \hat{\zeta}$ be values to be chosen later. Let $d$ be even. At a high level, $\ell_2$ essentially controls the smoothness of $F_1$ and $F_2$, $C$ is basically a constant, and $\hat{\zeta}$ is a quantity that affects the heterogeneity, initial suboptimality gap, and initial distance. We give the instance:

$$F_1(x) = -\ell_2\hat{\zeta}x_1 + \frac{C\ell_2}{2}x_d^2 + \frac{\ell_2}{2}\sum_{i=1}^{\frac{d}{2}-1}(x_{2i+1} - x_{2i})^2 + \frac{\mu}{2}\|x\|^2 \tag{274}$$

$$F_2(x) = \frac{\ell_2}{2}\sum_{i=1}^{d/2}(x_{2i} - x_{2i-1})^2 + \frac{\mu}{2}\|x\|^2 \tag{275}$$

Where $F = \frac{F_1 + F_2}{2}$.

These functions are the same as those used in (Woodworth et al., 2020a; Woodworth, 2021) for lower bounds in distributed optimization. They are also similar to the instances used to prove convex

optimization lower bounds (Nesterov, 2003) and distributed optimization lower bounds (Arjevani & Shamir, 2015).

These two functions have the following property

$$x_{\text{even}} \in \text{span}\{e_1, e_2, \ldots, e_{2i}\} \implies \begin{cases} \nabla F_1(x_{\text{even}}) \in \text{span}\{e_1, e_2, \ldots, e_{2i+1}\} \\ \nabla F_2(x_{\text{even}}) \in \text{span}\{e_1, e_2, \ldots, e_{2i}\} \end{cases} \tag{276}$$

$$x_{\text{odd}} \in \text{span}\{e_1, e_2, \ldots, e_{2i-1}\} \implies \begin{cases} \nabla F_1(x_{\text{odd}}) \in \text{span}\{e_1, e_2, \ldots, e_{2i-1}\} \\ \nabla F_2(x_{\text{odd}}) \in \text{span}\{e_1, e_2, \ldots, e_{2i}\} \end{cases} \tag{277}$$

What the property roughly says is the following. Suppose we so far have managed to turn an even number of coordinates nonzero. Then we can only use gradient queries of $F_1$ to access the next coordinate. After client 1 queries the gradient of $F_1$, it now has an odd number of unlocked coordinates, and cannot unlock any more coordinates until a communication occurs. Similar is true if the number of unlocked coordinates is odd. And so each round of communication can only unlock a single new coordinate.

We start by writing out the gradients of $F_1$ and $F_2$. Assume that $d$ is even. Then:

$$[\nabla F_1(x)]_1 = -\ell_2 \hat{\zeta} + \mu x_1 \tag{278}$$
$$[\nabla F_1(x)]_d = C\ell_2 x_d + \mu x_d \tag{279}$$

$$\begin{cases} [\nabla F_1(x)]_i = \ell_2(x_i - x_{i-1}) + \mu x_i & i \text{ odd}, 2 \leq i \leq d-1 \\ [\nabla F_1(x)]_i = -\ell_2(x_{i+1} - x_i) + \mu x_i & i \text{ even}, 2 \leq i \leq d-1 \end{cases} \tag{280}$$

and

$$[\nabla F_2(x)]_1 = -\ell_2(x_2 - x_1) + \mu x_1 \tag{281}$$
$$[\nabla F_2(x)]_d = \ell_2(x_d - x_{d-1}) + \mu x_d \tag{282}$$

$$\begin{cases} [\nabla F_2(x)]_i = -\ell_2(x_{i+1} - x_i) + \mu x_i & i \text{ odd}, 2 \leq i \leq d-1 \\ [\nabla F_2(x)]_i = \ell_2(x_i - x_{i-1}) + \mu x_i & i \text{ even}, 2 \leq i \leq d-1 \end{cases} \tag{283}$$

We define the class of functions that our lower bound will apply to. This definition follows (Woodworth et al., 2020a; Woodworth, 2021; Carmon et al., 2020):

**Definition G.1** (Distributed zero-respecting algorithm)**.** For a vector $v$, let $\text{supp}(v) = \{i \in \{1, \ldots, d\} : v_i \neq 0\}$. We say that an optimization algorithm is distributed zero-respecting if for any $i, k, r$, the $k$-th iterate on the $i$-th client in the $r$-th round $x_{i,k}^{(r)}$ satisfy

$$\text{supp}(x_{i,k}^{(r)}) \subseteq \bigcup_{0 \leq k' < k} \text{supp}(\nabla F_i(x_{i,k'}^{(r)})) \bigcup_{i' \in [N], 0 \leq k' \leq K-1, 0 \leq r' < r} \text{supp}(\nabla F_{i'}(x_{i',k'}^{(r')})) \tag{284}$$

Broadly speaking, distributed zero-respecting algorithms are those whose iterates have components in only dimensions that they can possibly have information on. As discussed in (Woodworth et al., 2020a), this means that algorithms which are *not* distributed zero-respecting are just "wild guessing". Algorithms that are distributed zero-respecting include SGD, ASG, FedAvg, SCAFFOLD, SAGA, and SSNM.

Next, we define the next condition we require on algorithms for our lower bound.

**Definition G.2.** We say that an algorithm is *distributed distance-conserving* if for any $i, k, r$, we have for the $k$-th iterate on the $i$-th client in the $r$-th round $x_{i,k}^{(r)}$ satisfies $\|x_{i,k}^{(r)} - x^*\|^2 \leq (c/2)[\|x_{\text{init}} - x^*\|^2 + \sum_{i=1}^N \|x_{\text{init}} - x_i^*\|^2]$, where $x_j^* := \arg\min_x F_j(x)$ and $x^* := \arg\min_x F(x)$ and $x_{\text{init}}$ is the initial iterate, for some scalar parameter $c$.

Algorithms which do not satisfy Definition 5.2 for polylogarithmic $c$ in problem parameters (see § 2) are those that move substantially far away from $x^*$, even farther than the $x_i^*$'s are from $x^*$. With this definition in mind, we slightly overload the usual definition of heterogeneity for the lower bound:

**Definition G.3.** A distributed optimization problem is $(\zeta, c)$-*heterogeneous* if $\max_{i \in [N]} \sup_{x \in A} \|\nabla F_i(x) - \nabla F(x)\|^2 \leq \zeta^2$, where we define $A := \{x : \|x - x^*\|^2 \leq (c/2)(\|x_{\text{init}} - x^*\|^2 + \sum_{i=1}^N \|x_{\text{init}} - x_i^*\|^2)\}$ for some scalar parameter $c$.

While convergence rates in FL are usually proven under Assumption B.5, the proofs can be converted to work under Definition G.3 as well, so long as one can prove all iterates stay within $A$ as defined in Definition G.3. We show that our algorithms satisfy Definition G.3 as well as a result (Thm. E.1, Thm. D.3, Thm. D.1)[4]. Other proofs of FL algorithms also satisfy this requirement, most notably the proof of the convergence of FedAvg in Woodworth et al. (2020a).

Following (Woodworth et al., 2020a), we start by making the argument that, given the algorithm is distributed zero-respecting, we can only unlock one coordinate at a time. Let $E_i = \text{span}\{e_1, \ldots, e_i\}$, and $E_0$ be the null span. Then

**Lemma G.4.** *Let $\hat{x}$ be the output of a distributed zero-respecting algorithm optimizing $F = \frac{1}{2}(F_1 + F_2)$ after $R$ rounds of communication. Then we have*

$$\text{supp}(\hat{x}) \in E_R \tag{285}$$

*Proof.* A proof is in Woodworth et al. (2020a, Lemma 9). □

We now compute various properties of this distributed optimization problem.

## G.1 STRONG CONVEXITY AND SMOOTHNESS OF $F, F_1, F_2$

From Woodworth (2021, Lemma 25), as long as $\ell_2 \leq \frac{\beta - \mu}{4}$, we have that $F, F_1, F_2$ are $\beta$-smooth and $\mu$-strongly convex.

## G.2 THE SOLUTIONS OF $F, F_1, F_2$

First, observe that from the gradient of $F_2$ computed in Eq. (281), $x_2^* = \arg\min_x F_2(x) = \vec{0}$. Next observe that from the gradient of $F_1$ computed in Eq. (278), $x_1^* = \arg\min_x F_1(x) = \frac{\ell_2 \hat{\zeta}}{\mu} e_1$. Thus $\|x_2^*\|^2 = 0$ and $\|x_1^*\|^2 = \frac{\ell_2^2 \hat{\zeta}^2}{\mu^2}$.

From Woodworth (2021, Lemma 25), if we let $\alpha = \sqrt{1 + \frac{2\ell_2}{\mu}}$, $q = \frac{\alpha - 1}{\alpha + 1}$, $C = 1 - q$, then $\|x^*\|^2 = \frac{\hat{\zeta}^2}{(1-q)^2} \sum_{i=1}^{d} q^{2i} = \frac{\hat{\zeta}^2 q^2 (1 - q^{2d})}{(1-q)^2 (1 - q^2)}$.

## G.3 INITIAL SUBOPTIMALITY GAP

From Woodworth (2021, Lemma 25), $F(0) - F(x^*) \leq \frac{q \ell_2 \hat{\zeta}^2}{4(1-q)}$.

## G.4 SUBOPTIMALITY GAP AFTER $R$ ROUNDS OF COMMUNICATION

From (Woodworth, 2021, Lemma 25), $F(\hat{x}) - F(x^*) \geq \frac{\hat{\zeta}^2 \mu q^2}{16(1-q)^2(1-q^2)} q^{2R}$, as long as $d \geq R + \frac{\log 2}{2 \log(1/q)}$.

## G.5 COMPUTATION OF $\zeta^2$

We start by writing out the difference between the gradient of $F_1$ and $F_2$, using Eq. (278) and Eq. (281).

$$\|\nabla F_1(x) - \nabla F_2(x)\|^2 = \ell_2^2 (-\hat{\zeta} + x_2 - x_1)^2 + \ell_2^2 (Cx_d - x_d + x_{d-1})^2 + \sum_{i=2}^{d-1} \ell_2^2 (x_{i+1} - x_{i-1})^2 \tag{286}$$

We upper bound each of the terms:

$$(x_{i+1} - x_{i-1})^2 = x_{i+1}^2 - 2x_{i+1}x_{i-1} + x_{i-1}^2 \leq 2x_{i+1}^2 + 2x_{i-1}^2 \tag{287}$$

---

[4]We do not formally show it for SAGA and SSNM, as the algorithms are functionally the same as SGD and ASG under full participation.

$$(-\hat{\zeta} + x_2 - x_1)^2 = \hat{\zeta}^2 + x_2^2 + x_1^2 - 2\hat{\zeta}x_2 + 2\hat{\zeta}x_1 - 2x_2x_1 \tag{288}$$
$$\leq 3\hat{\zeta}^2 + 3x_2^2 + 3x_1^2 \tag{289}$$

$$(Cx_d - x_d + x_{d-1})^2 = C^2x_d^2 + x_d^2 + x_{d-1}^2 - 2Cx_d^2 + 2Cx_dx_{d-1} - 2x_dx_{d-1} \tag{290}$$
$$\leq C^2x_d^2 + 2x_d^2 + 2x_{d-1}^2 + 2Cx_{d-1}^2 \tag{291}$$

So altogether, using the fact that $C \leq 1$,

$$\frac{1}{\ell_2^2}\|\nabla F_1(x) - \nabla F_2(x)\|^2 \tag{292}$$

$$\leq 3\hat{\zeta}^2 + 3x_2^2 + 3x_1^2 + 3x_d^2 + 3x_{d-1}^2 + \sum_{i=2}^{d-1} 2x_{i+1}^2 + 2x_{i-1}^2 \tag{293}$$

$$\leq 3\hat{\zeta}^2 + 7\|x\|^2 \tag{294}$$

$$\leq 3\hat{\zeta}^2 + 14\|x - x^*\|^2 + 14\|x^*\|^2 \tag{295}$$

$$\leq 5\hat{\zeta}^2 + 28c\frac{\hat{\zeta}^2q^2}{(1-q)^2(1-q^2)} + 28c\frac{\ell_2^2\hat{\zeta}^2}{\mu^2} \tag{296}$$

Next we use Definition G.2 which says that $\|x - x^*\| \leq \frac{c}{2}[\|x_{\text{init}} - x^*\|^2 + \sum_{i=1}^{N} \|x_{\text{init}} - x_i^*\|^2]$. For ease of calculation, we assume $c \geq 1$. Recall that we calculated $\|x^*\|^2 \leq \frac{\hat{\zeta}^2q^2}{(1-q)^2(1-q^2)}$ (because $0 < q < 1$), $\|x_2^*\|^2 = 0$, $\|x_1^*\|^2 = \frac{\ell_2^2\hat{\zeta}^2}{\mu^2}$.

$$\frac{1}{\ell_2^2}\|\nabla F_1(x) - \nabla F_2(x)\|^2 \leq 3\hat{\zeta}^2 + 24c\|x^*\|^2 + 7c\|x_1^*\|^2 + 7c\|x_2^*\|^2 \tag{297}$$

$$\leq 3\hat{\zeta}^2 + \frac{24c\hat{\zeta}^2q^2}{(1-q)^2(1-q^2)} + \frac{7c\ell_2^2\hat{\zeta}^2}{\mu^2} \tag{298}$$

Altogether,

$$\|\nabla F_1(x) - \nabla F_2(x)\|^2 \leq 3\ell_2^2\hat{\zeta}^2 + \frac{24c\ell_2^2\hat{\zeta}^2q^2}{(1-q)^2(1-q^2)} + \frac{7c\ell_2^4\hat{\zeta}^2}{\mu^2} \tag{299}$$

### G.6 Theorem

With all of the computations earlier, we are now prepared to prove the theorem.

**Theorem G.5.** *For any number of rounds $R$, number of local steps per-round $K$, and $(\zeta, c)$-heterogeneity (Definition 5.3), there exists a global objective $F$ which is the average of two $\beta$-smooth (Assumption B.4) and $\mu(\geq 0)$-strongly convex (Assumption B.1) quadratic client objectives $F_1$ and $F_2$ with an initial sub-optimality gap of $\Delta$, such that the output $\hat{x}$ of any distributed zero-respecting (Definition 5.1) and distance-conserving algorithm (Definition 5.2) satisfies*

- ***Strongly convex:*** $F(\hat{x}) - F(x^*) \geq \Omega(\min\{\Delta, 1/(c\kappa^{3/2})(\zeta^2/\beta)\}\exp(-R/\sqrt{\kappa}).)$ *when $\mu > 0$, and*

- ***General Convex*** $F(\hat{x}) - F(x^*) \geq \Omega(\min\{\beta D^2/R^2, \zeta D/(c^{1/2}\sqrt{R^5})\})$ *when $\mu = 0$.*

*Proof.* **The convex case:** By the previous computations, we know that after $R$ rounds,

$$F(\hat{x}) - F(x^*) \geq \frac{\mu\hat{\zeta}^2q^2}{16(1-q)^2(1-q^2)}q^{2R} \tag{300}$$

$$\|x^*\|^2 \leq \frac{\hat{\zeta}^2q^2}{(1-q)^2(1-q^2)} \tag{301}$$

$$\|\nabla F_1(x) - \nabla F_2(x)\|^2 \leq 3\ell_2^2\hat{\zeta}^2 + \frac{24c\ell_2^2\hat{\zeta}^2q^2}{(1-q)^2(1-q^2)} + \frac{7c\ell_2^4\hat{\zeta}^2}{\mu^2} \tag{302}$$

To maintain the property that $\|x^{(0)} - x^*\| = \|x^*\| \leq D$ and Definition G.3, Choose $\hat{\zeta}$ s.t.

$$\hat{\zeta}^2 = \nu \min\{\frac{\zeta^2}{\ell_2^2}, \frac{(1-q)^2(1-q^2)\zeta^2}{c\ell_2^2 q^2}, \frac{\mu^2\zeta^2}{c\ell_2^4}, \frac{(1-q)^2(1-q^2)D^2}{q^2}\} \tag{303}$$

For an absolute constant $\nu$.

This leads to (throughout we use $\frac{\mu}{\ell_2} = \frac{(1-q)^2}{2q}$, (Woodworth, 2021, Eq 769))

$$F(\hat{x}) - F(x^*) \tag{304}$$

$$\geq \frac{\mu\hat{\zeta}^2 q^2}{16(1-q)^2(1-q^2)} q^{2R} \tag{305}$$

$$\geq \nu \min\{\frac{\mu\zeta^2}{\ell_2^2}, \frac{\mu(1-q)^2(1-q^2)\zeta^2}{c\ell_2^2 q^2}, \frac{\mu^3\zeta^2}{c\ell_2^4}, \frac{\mu(1-q)^2(1-q^2)D^2}{q^2}\} \frac{q^2}{16(1-q)^2(1-q^2)} q^{2R} \tag{306}$$

$$\geq \nu \min\{\frac{(1-q)^2\zeta^2}{2\ell_2 q}, \frac{\mu(1-q)^2(1-q^2)\zeta^2}{c\ell_2^2 q^2}, \tag{307}$$

$$\frac{\mu\zeta^2(1-q)^4}{4c\ell_2^2 q^2}, \frac{\mu(1-q)^2(1-q^2)D^2}{q^2}\} \frac{q^2}{16(1-q)^2(1-q^2)} q^{2R} \tag{308}$$

$$\geq \nu \min\{\frac{\zeta^2 q}{32\ell_2(1-q^2)}, \frac{\mu\zeta^2}{16c\ell_2^2}, \frac{\mu\zeta^2(1-q)}{64c\ell_2^2(1+q)}, \frac{\mu D^2}{16}\} q^{2R} \tag{309}$$

$$\tag{310}$$

We choose $\mu = \frac{\ell_2}{64R^2}$, which ensures $\alpha \geq 2$. So noting that $(1-q^2) = (1-q)(1+q) \leq (1+q)$, $1+q = \frac{2\alpha}{\alpha+1}$, $1-q = \frac{2}{\alpha+1}$:

$$F(\hat{x}) - F(x^*) \tag{311}$$

$$\geq \nu \min\{\frac{\zeta^2}{32\ell_2}(\frac{\alpha-1}{\alpha+1}\frac{\alpha+1}{2\alpha}), \frac{\mu\zeta^2}{16c\ell_2^2}, \frac{\mu\zeta^2}{64c\ell_2^2}(\frac{2}{\alpha+1}\frac{\alpha+1}{2\alpha}), \frac{\mu D^2}{16}\} q^{2R} \tag{312}$$

$$\geq \nu \min\{\frac{\zeta^2}{32\ell_2}(\frac{\alpha-1}{\alpha+1}\frac{\alpha+1}{2\alpha}), \frac{\mu\zeta^2}{16c\ell_2^2}, \frac{\mu\zeta^2}{64c\ell_2^2}(\frac{2}{\alpha+1}\frac{\alpha+1}{2\alpha}), \frac{\mu D^2}{16}\} \exp(-2R\log(\frac{\alpha+1}{\alpha-1})) \tag{313}$$

$$\geq \nu \min\{\frac{\zeta^2}{64\ell_2}, \frac{\mu\zeta^2}{16c\ell_2^2}, \frac{\mu\zeta^2}{64c\ell_2^2}(\frac{1}{\alpha}), \frac{\mu D^2}{16}\} \exp(-\frac{4R}{\alpha-1}) \tag{314}$$

So,

$$F(\hat{x}) - F(x^*) \tag{315}$$

$$\geq \nu \min\{\frac{\zeta^2}{64\ell_2}, \frac{\mu\zeta^2}{16c\ell_2^2}, \frac{\mu\zeta^2}{64c\ell_2^2}(\frac{\mu}{3\ell_2})^{1/2}, \frac{\mu D^2}{16}\} \exp(-\frac{8R\sqrt{\mu}}{\sqrt{\ell_2}}) \tag{316}$$

$$\geq \Omega(\min\{\frac{\zeta^2}{\ell_2}, \frac{\zeta^2}{c\ell_2 R^2}, \frac{\zeta^2}{c\ell_2 R^3}, \frac{\ell_2 D^2}{R^2}\}) \tag{317}$$

$$\geq \Omega(\frac{\zeta^2}{c\ell_2 R^3}, \frac{\ell_2 D^2}{R^2}\}) \tag{318}$$

$$\tag{319}$$

Where we used $c > 1$. Setting $\ell_2 = \Theta(\min\{\beta, \frac{\zeta}{c^{1/2}DR^{1/2}}\})$ (which will ensure that $\ell_2 \leq \frac{\beta-\mu}{4}$ with appropriate constants chosen), Which altogether gives us

$$F(\hat{x}) - F(x^*) \geq \Omega(\min\{\frac{\zeta D}{c^{1/2}R^{5/2}}, \frac{\beta D^2}{R^2}\}) \tag{320}$$

**Strongly Convex Case:**

By the previous computations, we know that after $R$ rounds,

$$F(\hat{x}) - F(x^*) \geq \frac{\mu \hat{\zeta}^2 q^2}{16(1-q)^2(1-q^2)} q^{2R} \tag{321}$$

$$F(\hat{x}) - F(x^*) \leq \frac{q \ell_2 \hat{\zeta}^2}{4(1-q)} \tag{322}$$

$$\|\nabla F_1(x) - \nabla F_2(x)\|^2 \leq 3\ell_2^2 \hat{\zeta}^2 + \frac{24c\ell_2^2 \hat{\zeta}^2 q^2}{(1-q)^2(1-q^2)} + \frac{7c\ell_2^4 \hat{\zeta}^2}{\mu^2} \tag{323}$$

To maintain the property that $F(\hat{x}) - F(x^*) \leq \Delta$ and that $\|\nabla F_1(x) - \nabla F_2(x)\|^2 \leq \zeta^2$ for all $x$ encountered during the execution of the algorithm,

Choose $\hat{\zeta}$ s.t.

$$\hat{\zeta}^2 = \nu \min\{\frac{\zeta^2}{\ell_2^2}, \frac{(1-q)^2(1-q^2)\zeta^2}{c\ell_2^2 q^2}, \frac{\mu^2 \zeta^2}{c\ell_2^4}, \frac{(1-q)\Delta}{q\ell_2}\} \tag{324}$$

For some constant $\nu$.

Again using $\frac{\mu}{\ell_2} = \frac{(1-q)^2}{2q}$) and following the same calculations as in the convex case, we use the fact that $9\mu \leq \beta$, and that it is possible to choose $\ell_2$ s.t. $2\mu \leq \ell_2 \leq \frac{\beta-\mu}{4}$, which ensures $\beta \geq 2$.

$$F(\hat{x}) - F(x^*) \tag{325}$$

$$\geq \frac{\mu \hat{\zeta}^2 q^2}{16(1-q)^2(1-q^2)} q^{2R} \tag{326}$$

$$\geq \nu \min\{\frac{\zeta^2}{64\ell_2}, \frac{\mu \zeta^2}{16c\ell_2^2}, \frac{\mu \zeta^2}{64c\ell_2^2}(\frac{1}{\alpha}), \frac{\Delta}{4}\} \exp(-\frac{4R}{\alpha-1}) \tag{327}$$

$$\tag{328}$$

We use $9\mu \leq \beta$ and because it is possible to choose $\ell_2$ such that $2\mu \leq \ell_2 \leq \frac{\beta-\mu}{4}$ (Woodworth, 2021, Eq. 800), which ensures $\alpha \geq 2$ and $\frac{1}{\alpha} \geq \sqrt{\frac{\mu}{3\ell_2}}$

$$F(\hat{x}) - F(x^*) \tag{329}$$

$$\geq \Omega(\min\{\frac{\zeta^2}{\ell_2}, \frac{\mu \zeta^2}{c\ell_2^2}, \frac{\mu \zeta^2}{c\ell_2^2}(\frac{\mu}{\ell_2})^{1/2}, \Delta\} \exp(-\frac{8R\sqrt{\mu}}{\sqrt{\ell_2}})) \tag{330}$$

$$\geq \Omega(\min\{\frac{\mu \zeta^2}{c\ell_2^2}(\frac{\mu}{\ell_2})^{1/2}, \Delta\} \exp(-\frac{8R\sqrt{\mu}}{\sqrt{\ell_2}})) \tag{331}$$

Next we note that $\ell_2 \leq \beta$ and $\ell_2 \geq \frac{\beta}{5}$ (Woodworth, 2021, Eq 801), which gives us

$$F(\hat{x}) - F(x^*) \geq \Omega(\min\{\frac{\mu^{3/2}\zeta^2}{c\beta^{5/2}}, \Delta\} \exp(-\frac{18R}{\sqrt{\kappa}})) \tag{332}$$

$\square$

## H   TECHNICAL LEMMAS

**Lemma H.1.** *Let*

$$g^{(r)} := \frac{1}{SK} \sum_{i \in \mathcal{S}_r} \sum_{k=1}^K g_{i,k}^{(r)} \tag{333}$$

*Given Assumption B.6, Assumption B.5, and uniformly sampled clients per round,*

$$\mathbb{E}\|\frac{1}{SK} \sum_{i \in \mathcal{S}_r} \sum_{k=1}^K g_{i,k}^{(r)} - \nabla F(x^{(r)})\|^2 \leq \frac{\sigma^2}{SK} + (1 - \frac{S-1}{N-1})\frac{\zeta^2}{S} \tag{334}$$

*If instead the clients are arbitrarily (possibly randomly) sampled and there is no gradient variance,*

$$\|\frac{1}{SK} \sum_{i \in \mathcal{S}_r} \sum_{k=1}^{K} g_{i,k}^{(r)} - \nabla F(x^{(r)})\|^2 \leq \zeta^2 \tag{335}$$

*Proof.*

$$\mathbb{E}\|\frac{1}{SK} \sum_{i \in \mathcal{S}_r} \sum_{k=1}^{K} \nabla f(x; z_i) - \nabla F(x)\|^2 \tag{336}$$

$$= \mathbb{E}\|\frac{1}{S} \sum_{i \in \mathcal{S}_r} \nabla F_i(x) - \nabla F(x)\|^2 + \mathbb{E}_{\mathcal{S}_r} \|\frac{1}{SK} \sum_{i \in \mathcal{S}_r} \sum_{k=0}^{K-1} \nabla f(x; z_i) - \nabla F_i(x)\|^2 \tag{337}$$

$$\leq \mathbb{E}\|\frac{1}{S} \sum_{i \in \mathcal{S}_r} \nabla F_i(x) - \nabla F(x)\|^2 + \frac{1}{SK} \mathbb{E}\|\nabla f(x; z_i) - \nabla F_i(x)\|^2 \tag{338}$$

$$\tag{339}$$

Where the first inequality uses the fact that $\mathbb{E}[X^2] = \text{Var}(X) + \mathbb{E}[X]^2$, and the second equality uses the fact that the variance is i.i.d across $i$ and $k$. Note that $\mathbb{E}\|\nabla f(x; z_i) - \nabla F_i(x)\|^2 \leq \sigma^2$ by Assumption B.6. On the other hand by Assumption B.5

$$\mathbb{E}\|\frac{1}{S} \sum_{i \in \mathcal{S}_r} \nabla F_i(x) - \nabla F(x)\|^2 \leq (1 - \frac{S-1}{N-1}) \frac{\mathbb{E}\|\nabla F_i(x) - \nabla F(x)\|^2}{S} \leq (1 - \frac{S-1}{N-1}) \frac{\zeta^2}{S} \tag{340}$$

So altogether

$$\mathbb{E}\|\frac{1}{SK} \sum_{i \in \mathcal{S}_r} \sum_{k=1}^{K} \nabla f(x; z_i) - \nabla F(x)\|^2 \leq \frac{\sigma^2}{SK} + (1 - \frac{S-1}{N-1}) \frac{\zeta^2}{S} \tag{341}$$

The second conclusion follows by noting

$$\|\frac{1}{S} \sum_{i \in \mathcal{S}_r} \nabla F_i(x) - \nabla F(x)\|^2 \leq \frac{1}{S} \sum_{i \in \mathcal{S}_r} \|\nabla F_i(x) - \nabla F(x)\|^2 \tag{342}$$

$\square$

**Lemma H.2.** *Let $u, v$ be arbitrary (possibly random) points. Define $\hat{F}(x) = \frac{1}{SK} \sum_{i \in \mathcal{S}} \sum_{k=0}^{K-1} f(x; \hat{z}_{i,k})$ where $\hat{z}_{i,k} \sim \mathcal{D}_i$. Let*

$$w = \underset{x \in \{u,v\}}{\arg\min} \hat{F}(x)$$

*Then*

$$\mathbb{E}[F(w) - F(x^*)] \tag{343}$$

$$\leq \min\{F(u) - F(x^*), F(v) - F(x^*)\} + 4\frac{\sigma_F}{\sqrt{SK}} + 4\sqrt{1 - \frac{S-1}{N-1}} \frac{\zeta_F}{\sqrt{S}} \tag{344}$$

*and*

$$\mathbb{E}[F(w) - F(x^*)] \tag{345}$$

$$\leq \min\{\mathbb{E}F(u) - F(x^*), \mathbb{E}F(v) - F(x^*)\} + 4\frac{\sigma_F}{\sqrt{SK}} + 4\sqrt{1 - \frac{S-1}{N-1}} \frac{\zeta_F}{\sqrt{S}} \tag{346}$$

*Proof.* Suppose that $F(u) + 2a = F(v)$, where $a \geq 0$. Then,

$$\mathbb{E}[F(w) - F(x^*)] = P(w = u)(F(u) - F(x^*)) + P(w = v)(F(v) - F(x^*)) \tag{347}$$

Substituting $F(v) = F(u) + 2a$,

$$\mathbb{E}[F(w) - F(x^*)] = P(w = u)(F(u) - F(x^*)) + P(w = v)(F(u) + 2a - F(x^*)) \quad (348)$$

$$\leq F(u) - F(x^*) + 2a \quad (349)$$

Observe that $\mathbb{E}(\hat{F}(x) - F(x))^2 \leq \frac{\sigma_F^2}{SK} + (1 - \frac{S-1}{N-1})\frac{\zeta_F^2}{S}$ by Assumption B.7 and Assumption B.8. Therefore by Chebyshev's inequality,

$$P(|\hat{F}(x) - F(x)| \geq a) \leq \frac{1}{a^2}(\frac{\sigma_F^2}{SK} + (1 - \frac{S-1}{N-1})\frac{\zeta_F^2}{S}) \quad (350)$$

Observe that

$$P(w = v) \leq P(\hat{F}(u) > F(u) + a) + P(\hat{F}(v) < F(v) - a) \quad (351)$$

$$\leq \frac{2}{a^2}(\frac{\sigma_F^2}{SK} + (1 - \frac{S-1}{N-1})\frac{\zeta_F^2}{S}) \quad (352)$$

And so it is also true that

$$\mathbb{E}[F(w) - F(x^*)] = F(u) - F(x^*) + 2aP(w = v) \quad (353)$$

$$\leq F(u) - F(x^*) + 2a(\frac{2}{a^2}(\frac{\sigma_F^2}{SK} + (1 - \frac{S-1}{N-1})\frac{\zeta_F^2}{S})) \quad (354)$$

$$= F(u) - F(x^*) + \frac{4}{a}(\frac{\sigma_F^2}{SK} + (1 - \frac{S-1}{N-1})\frac{\zeta_F^2}{S}) \quad (355)$$

Therefore altogether we have that

$$\mathbb{E}[F(w) - F(x^*)] \leq F(u) - F(x^*) + \max_a \min\{\frac{4}{a}(\frac{\sigma_F^2}{SK} + (1 - \frac{S-1}{N-1})\frac{\zeta_F^2}{S}), 2a\} \quad (356)$$

So by maximizing for $a$

$$\frac{4}{a}(\frac{\sigma_F^2}{SK} + (1 - \frac{S-1}{N-1})\frac{\zeta_F^2}{S}) = 2a \implies a = \sqrt{2(\frac{\sigma_F^2}{SK} + (1 - \frac{S-1}{N-1})\frac{\zeta_F^2}{S})} \quad (357)$$

which gives us

$$\mathbb{E}[F(w) - F(x^*)] \leq F(u) - F(x^*) + 4\frac{\sigma_F}{\sqrt{SK}} + 4\sqrt{1 - \frac{S-1}{N-1}}\frac{\zeta_F}{\sqrt{S}} \quad (358)$$

Then proof also holds if we switch the roles of $u$ and $v$, and so altogether we have that: if $F(u) < F(v)$,

$$\mathbb{E}[F(w) - F(x^*)] \leq F(u) - F(x^*) + 4\frac{\sigma_F}{\sqrt{SK}} + 4\sqrt{1 - \frac{S-1}{N-1}}\frac{\zeta_F}{\sqrt{S}} \quad (359)$$

if $F(u) > F(v)$,

$$\mathbb{E}[F(w) - F(x^*)] \leq F(v) - F(x^*) + 4\frac{\sigma_F}{\sqrt{SK}} + 4\sqrt{1 - \frac{S-1}{N-1}}\frac{\zeta_F}{\sqrt{S}} \quad (360)$$

implying the first part of the theorem.

$$\mathbb{E}[F(w) - F(x^*)] \leq \min\{F(u) - F(x^*), F(v) - F(x^*)\} + 4\frac{\sigma_F}{\sqrt{SK}} + 4\sqrt{1 - \frac{S-1}{N-1}}\frac{\zeta_F}{\sqrt{S}} \quad (361)$$

For the second part,

$$\mathbb{E}[\min\{F(u) - F(x^*), F(v) - F(x^*)\}|v] = \int_{F(u) \leq F(v)} F(u) - F(x^*) \quad (362)$$

$$\leq \min\{\mathbb{E}F(u) - F(x^*), F(v) - F(x^*)\} \quad (363)$$

and then

$$\mathbb{E}[\min\{\mathbb{E}F(u) - F(x^*), F(v) - F(x^*)\}] = \int_{F(v) \leq \mathbb{E}F(u)} F(u) - F(x^*) \quad (364)$$

$$\leq \min\{\mathbb{E}F(u) - F(x^*), \mathbb{E}F(v) - F(x^*)\} \quad (365)$$

$\square$

# I  EXPERIMENTAL SETUP DETAILS

## I.1  CONVEX OPTIMIZATION

We empirically evaluate FedChain on federated regularized logistic regression. Let $(x_{i,j}, y_{i,j})$ be the $j$th datapoint of the $i$th client and $n_i$ is the number of datapoints for the $i$-th client. We minimize Eq. (1) where

$$F_i(w) = \frac{1}{n_i}\left(\sum_{j=1}^{n_i} -y_{i,j}\log(w^\top x_{i,j}) - (1-y_{i,j})\log(1-w^\top x_{i,j})\right) + \frac{\mu}{2}\|w\|^2.$$

**Dataset.** We use the MNIST dataset of handwritten digits (LeCun et al., 2010). We model a federated setting with five clients by partitioning the data into groups. First, we take 500 images from each digit class (total 5,000). Each client's local data is a mixture of data drawn from two digit classes (leading to heterogeneity), and data sampled uniformly from all classes.

We call a federated dataset $X\%$ homogeneous if the first $X\%$ of each class's 500 images is shuffled and evenly partitioned to each client. The remaining $(100-X)\%$ is partitioned as follows: client $i \in \{1,\ldots,5\}$ receives the remaining non-shuffled data from classes $2i-2$ and $2i-1$. For example, in a 50% homogeneous setup, client 3 has 250 samples from digit 4, 250 samples from digit 5, and 500 samples drawn uniformly from all classes. Note that 100% homogeneity is *not* the same thing as setting heterogeneity $\zeta = 0$ due to sampling randomness; we use this technique for lack of a better control over $\zeta$. To model binary classification, we let even classes represent 0's and odd classes represent 1's, and set $K = 20$. **All clients participate per round.**

**Hyperparameters.** All experiments initialize iterates at 0 with regularization $\mu = 0.1$. We fix the total number of rounds $R$ (differs across experiments). For algorithms without stepsize decay, the tuning process is as follows.

We tune all stepsizes $\eta$ in the range below:

$$\{10^{-3}, 10^{-2.5}, 10^{-2}, 10^{-1.5}, 10^{-1}\} \tag{366}$$

We tune the percentage of rounds before switching from $\mathcal{A}_{\text{local}}$ to $\mathcal{A}_{\text{global}}$ (if applicable) as

$$\{10^{-2}, 10^{-1.625}, 10^{-1.25}, 10^{-0.875}, 10^{-0.5}\} \tag{367}$$

For algorithms with acceleration, we run experiments using the more easily implementable (but less mathematically tractable for our analysis) version in Aybat et al. (2019) as opposed to Algo. 3.

For algorithms with stepsize decay, we instead use the range Eq. (367) to decide the percentage of rounds to wait before decreasing the stepsize by half–denote this number of rounds as $R_{\text{decay}}$. Subsequently, every factor of 2 times $R_{\text{decay}}$ we decrease the stepsize by half.

We use the heuristic that when the stepsize decreases to $\eta/K$, we switch to $\mathcal{A}_{\text{global}}$ and begin the stepsize decay process again. All algorithms are tuned to the best final gradient norm averaged over 1000 runs.

## I.2  NONCONVEX OPTIMIZATION

### I.2.1  EMNIST

In this section we detail how we use the EMNIST (Cohen et al., 2017) dataset for our nonconvex experiments, where the handwritten characters are partitioned by their author. In this dataset, there are 3400 clients, with 671,585 images in the train set and 77,483 in the test set. The task is to train a convolutional network with two convolutional layers, max-pooling, dropout, and two dense layers as in the Federated Learning task suite from (Reddi et al., 2020).

**Hyperparameters.** We set $R = 500$, and rather than fixing the number of local steps, we fix the number of local client epochs (the number of times a client performs gradient descent over its own dataset) to 20. The number of clients sampled per round is 10. These changes were made in light of the imbalanced dataset sizes across clients, and is standard for this dataset-task (Reddi et al., 2020; Charles & Konečný, 2020; Charles et al., 2021). For all algorithms aside from (M-SGD), we tune

$\eta$ in the range $\{0.05, 0.1, 0.15, 0.2, 0.25\}$. We tune the percentage of rounds before switching from $\mathcal{A}_{\text{local}}$ to $\mathcal{A}_{\text{global}}$ in $\{0.1, 0.3, 0.5, 0.7, 0.9\}$, where applicable.

We next detail the way algorithms with stepsize decay are run. We tune the stepsize $\eta$ in $\{0.1, 0.2, 0.3, 0.4, 0.5\}$. Let $R_{\text{decay}}$ be the number of rounds before the first decay event. We tune $R_{\text{decay}} \in \lceil \{0.1R, 0.275R, 0.45R\} \rceil$. Whenever the number of passed rounds is a power of 2 of $R_{\text{decay}}$, we halve the stepsize/ After $R'$ rounds have passed, we enter $\mathcal{A}_{\text{global}}$, where we tune $R' \in \lceil \{0.5R, 0.7R, 0.9R\} \rceil$. We restart the decay process upon entering $\mathcal{A}_{\text{global}}$.

For M-SGD we tune

$$\eta \in \{0.1, 0.2, 0.3, 0.4, 0.5\}$$

and

$$R_{\text{decay}} \in \lceil \{10^{-2}R, 10^{-1.625}R, 10^{-1.25}R, 10^{-0.875}R, 10^{-0.5}R\} \rceil$$

For stepsize decaying SCAFFOLD and FedAvg, we tune $\eta \in \{0.1, 0.2, 0.3, 0.4, 0.5\}$ and $R_{\text{decay}} \in \lceil \{0.1R, 0.275R, 0.45R\} \rceil$. When evaluating a particular metric the algorithms are tuned according to the mean of the last 5 evaluated iterates of that particular metric. Reported accuracies are also the mean of the last 5 evaluated iterates, as the values mostly stabilized during training.

### I.2.2 CIFAR-100

For the CIFAR-100 experiments, we use the CIFAR-100 (Krizhevsky, 2009) dataset, where the handwritten characters are partitioned via the method proposed in (Reddi et al., 2020). In this dataset, there are 500 train clients, with a total of 50,000 images in the train set. There are 100 test clients and 10,000 images in the test set. The task is to train a ResNet-18, replacing batch normalization layers with group normalization as in the Federated Learning task suite from (Reddi et al., 2020). We leave out SCAFFOLD in all experiments because of out of memory issues.

**Hyperparameters.** We set $R = 5000$, and rather than fixing the number of local steps, we fix the number of local client epochs (the number of times a client performs gradient descent over its own dataset) to 20. The number of clients sampled per round is 10. These changes were made in light of the imbalanced dataset sizes across clients, and is standard for this dataset-task (Reddi et al., 2020; Charles & Konečný, 2020; Charles et al., 2021). For all algorithms we tune $\eta$ in $\{0.1, 0.2, 0.3, 0.4, 0.5\}$. Algorithms without stepsize decay tune the percentage of rounds before switching to $\mathcal{A}_{\text{global}}$ in $\{0.1, 0.3, 0.5, 0.7, 0.9\}$. Algorithms with stepsize decay (aside from M-SGD) tune the number of rounds before the first decay event in $\{0.1, 0.275, 0.45\}$ (and for every power of 2 of that percentage, the stepsize decays in half), and the number of rounds before swapping to $\mathcal{A}_{\text{global}}$ in $\{0.5, 0.7, 0.9\}$ if applicable (stepsize decay process restarts after swapping). M-SGD tunes the number of rounds before the first decay event in $\{0.01, 0.0237, 0.0562, 0.1334, 0.3162\}$. We tune for and return the test accuracy of the last iterate, as accuracy was still improving as we trained.

## J   ADDITIONAL CONVEX EXPERIMENTS

In this section, we give supplemental experimental results to verify the following claims:

1. Higher $K$ allows (1) accelerated algorithms to perform better than non-accelerated algorithms, and furthermore (2) allows us to run $\mathcal{A}_{\text{local}}$ for one round to get satisfactory results.

2. Our FedChain instantiations outperform stepsize decaying baselines

### J.1   VERIFYING THE EFFECT OF INCREASED $K$ AND $R = 1$

Our proofs of FedChain all also work if we run $\mathcal{A}_{\text{local}}$ for only one communication round, so long as $K$ is sufficiently large (exact number is in the formal theorem proofs). In this section we verify the effect of large $K$ in conjunction with running $\mathcal{A}_{\text{local}}$ for only one round. The setup is the same as in App. I.1, except we tune the stepsize $\eta$ in a larger grid:

$$\eta \in \{10^{-3}, 10^{-2.5}, 10^{-2}, 10^{-1.5}, 10^{-1}, 10^{-0.5}, 10^0\} \tag{368}$$

The plots are in Fig. 3. For algorithms that are "1-X→Y", we run algorithm X for one round and algorithm Y for the rest of the rounds. These algorithms are tuned in the following way. Algorithm X

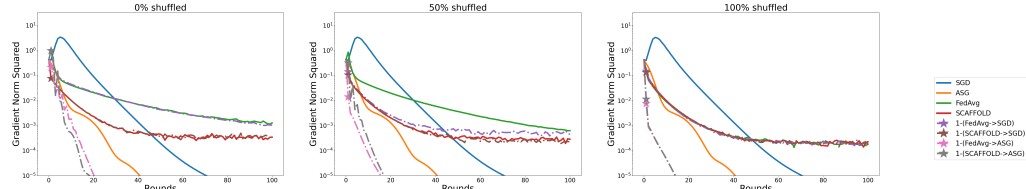

Figure 3: We investigate the effect of high $K$ ($K = 100$). "1-X→Y" denotes a chained algorithm with X run for one round and Y run for the rest of the rounds. Chained algorithms, even with one round allocated to $\mathcal{A}_{\text{local}}$, perform the best. Furthermore, accelerated algorithms outperform their non-accelerated counterparts.

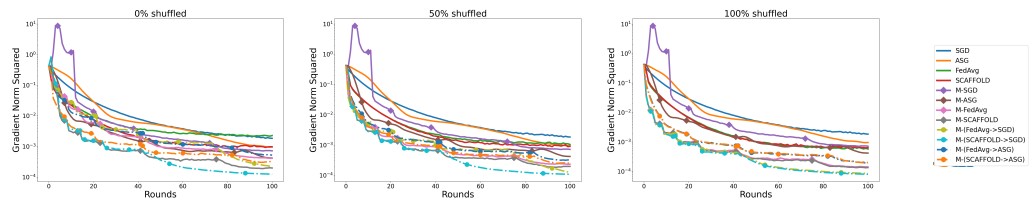

Figure 4: The same as Fig. 2, except we include stepsize decay for baseline algorithms. Chained algorithms still perform the best.

has its stepsize tuned in Eq. (368), and Algorithm Y independently has its stepsize tuned in Eq. (368). We require these to be tuned separately because local update algorithms and centralized algorithms have stepsizes that depend on $K$ differently if $K$ is large (see, for example, Thm. D.1 and Thm. E.1). The baselines have a single stepsize tuned in Eq. (368). These are tuned for the lowest final gradient norm averaged over 1000 runs.

Overall, we see that in the large $K$ regime, algorithms that start with a local update algorithm and then finish with an accelerated centralized algorithm have the best communication complexity. This is because larger $K$ means the error due to variance decreases, for example as seen in Thm. F.2. Acceleration increases instability (if one does not perform a carefully calibrated stepsize decay, as we do not in the experiments), so decreasing variance helps accelerated algorithms the most. Furthermore, a single round for $\mathcal{A}_{\text{local}}$ suffices to see large improvement as seen in Fig. 3. This is expected, because both the "bias" term (the $\Delta \exp(-\frac{R\sqrt{K}}{\kappa})$ term in Thm. E.1) and the variance term become negligible, leaving just the heterogeneity term as desired.

## J.2 INCLUDING STEPSIZE DECAY BASELINES

In this section, we compare the performance of the algorithms against learning rate decayed SGD, FedAvg, ASG, and SCAFFOLD. We use the same setting as App. I.1 for the decay process and the stepsize. In this case, we still see that the chained methods outperform their non-chained counterparts.

