# OpenReview forum: "FedChain: Chained Algorithms for Near-optimal Communication Cost in Federated Learning"
_ICLR.cc/2022/Conference — ICLR 2022 Poster_

### Official Review · Reviewer_AN2w · 2021-10-27

**Correctness:** 3
**Technical Novelty And Significance:** 3
**Empirical Novelty And Significance:** 3
**Recommendation:** 6
**Confidence:** 4

**Main Review:**

The paper tries to answer a theoretical problem in federated optimization.
The paper proposed a multi-stage algorithm to match the lower bound established by Woodworth for intermediate heterogeneity levels.
In particular, they first run a local method up to reach certain error floor and then switch to a centralized method for the rest steps.
The method is intuitive and supported with theoretical analysis assuming strongly-convex objective functions.
The result is new for FL.
The proof seems correct and the writing is clear and easy to follow.
They also demonstrate the effectiveness of their method in mage classification tasks.

I have the following concerns:
1. In the proof of Theorem 1, the $n$ in the expression of $\Phi$ and $\eta$ of the first stage should be $N$.
2. I find that both Theorem 1 and Theorem 2 require sufficiently large $K$, which is important to make the $\widetilde{O}(\frac{\beta\zeta^2}{\mu^2R^2})$ dominate the error (e.g., see (25) and (29)). Besides, Section 3.1 provides a two-stage instantiation of proposed algorithm. It first lets each device run Local SGD and then run SGD to approach the local optimal point. The two results make me to wonder, why not we first run pure local training, letting each device obtain a local minimizer, and the run (accelerated) SGD. This is equivalent to let the first stage be a pure local training rather than local SGD. In this way, the communication cost is saved as much as possible, since it only requires one communication round. By contrast, current result (Theorem 1 and Theorem 2) requires $R/2$ rounds.
3. The paper only analyzes the strongly-convex case and omit the generally-convex case. Woodworth et al. (2020a) provides analysis for Local SGD under both cases. Considering the analysis method used in the paper is just to combine convergence rates of outputs from two (super)stages, I think it is not hard to extend the result to the generally-convex case. However, it is quite interesting and unclear to see the theoretical performance of proposed multistage algorithms for non-convex functions.
4. I don’t quite see why the work explains empirically-successful stepsize decay methods in FL, which is declaimed in the abstract. I feel it is exaggerated and should be supported with more evidence. In my opinion, what makes the proposed algorithm have a better dependence on $\zeta^2$ is not the stepsize decay but the switch to centralize methods. It supports my point that most of theorems is derived under constant small step sizes. The discussion in Appendix A is weak and farfetched. I think the author should support the claim with other suppport analysis, for example, analysis for simple cases or additional experiments to show the adaptive optimization methods indeed swtich to a centralized counterpart.

============================================

I have read the authors’ responses as well as other reviewers’ feedback.
At the moment, I think the proposed two-stage method is more like a trick rather than a universal method that can be implemented to other scenarios. There are two reasons supporting my thought.
First, the author can’t extend the theory to general convex and nonconvex settings, which implies the method replies on the strong-convexity, which brings great convenience for analysis.
Second, the author didn’t justify the necessity of the first-stage. From the author’s feedback (the added footnote 3), using one-round even has better performance. In this case, the two-stage algorithm is reduced to the last stage which is the accelerated SGD. Hence, it is not surprising it has so good performance. At this moment, I feel the proposed method is more like a trick rather than a universal component.
Therefore, I think the paper could provide limited insights for future work.

I think the author could improve their work in the following directions: (i) provide analysis for general convex and nonconvex settings, (ii) find a more distinguishable method to close the gap. For example, a softer decay of local update length.

**Summary Of The Paper:**

The paper proposed a provable multi-stage algorithm to match the lower bound established by Woodworth for intermediate heterogeneity levels in federated optimization.


**Summary Of The Review:**

The theoretical result is new in FL.
The proposed method has strong empirical performance in typical datasets.
Their match the lower bound and  fill the blank.
Though it has some limitations, I think it is worth to be published.

---

> ### Author Response · Authors · 2021-11-23
> **Response to Reviewer AN2w [1/1]**
>
> We thank the reviewer for their comments.  We correct the typos mentioned in the revised draft.
>
> **On concern 2, running only one round of FedAvg before switching**:
>
> This observation is correct.  We can save some communication in the way described.  We chose to present our results using R/2 to accommodate some of our experiments where client subsampling prevents us from spending only one round on local update methods.  We clarify this in the next revision and state this fact explicitly (Footnote 3).
>
> **On concern 3, extending the theory to general convex and nonconvex settings**:
>
> The general convex setting is more difficult than it looks.  Convergence rates in the general convex setting depend on the initial distance to optimum: for example SGD:
>
> $\mathbb{E}[F(\hat{x}) - F(x^*)] \leq \mathcal{O}(\frac{\beta D^2}{R} + \frac{\sigma D}{\sqrt{NKR}})$
>
> where $D = \|x^{(0)} - x^*\|$ and the rest of the notation is from our paper.  This is in contrast with results for the strongly convex setting, where rates can be obtained in terms of the initial suboptimality gap.
>
> From [1] we know that with large enough K,
>
> $\mathbb{E}[F(\hat{x}) - F(x^*)] \leq \mathcal{O}(\frac{(\beta \zeta^2 D^4)^{1/3}}{R^{2/3}})$
>
> But there is nowhere to substitute that progress into the rate for SGD (or for any other general convex rate that we know of). So our proof technique would not apply in this case.
>
> The nonconvex case is even more challenging because we cannot even get function value progress from the first superstage.
>
> One way to extend our results past the strongly convex case is to look at the PL condition, under which we may be able to achieve similar rates as in our paper.
>
> **On concern 4, the claim on our method possibly explaining stepsize scheduling in FL**:
>
> We will remove the claim from the abstract, as it is not a part of the main message and may be too strong (it is currently discussed in Appendix A).  However, we do think that stepsize decays like in [2] can be seen as gradual versions of our switching strategy, because in both cases the server update learning rate $\eta_g$ decreases much slower than the local update learning rate $\eta_l$.  As a result we can see that the decay schedules are slowly switching from a local step algorithm to a centralized algorithm.  In that sense, our work analyzes a simplified version of existing stepsize decay schemes, and demonstrates theoretically why such schemes are effective.  We include a plot showing learning rates under a decay strategy in Appendix A of the revised paper, which we hope illustrates our point.  We have also removed mention of [3] on this point of the revised paper, as its discussion was vague and did not provide much value.
>
> 1. Woodworth et al., "Minibatch vs local sgd for heterogeneous distributed learning." NeurIPS 2020
> 2. Charles, Zachary, and Jakub Konečný. "On the outsized importance of learning rates in local update methods." arXiv preprint arXiv:2007.00878 (2020)
> 3. Reddi et al., "Adaptive federated optimization." ICLR 2021

---

### Official Review · Reviewer_y87n · 2021-10-27

**Correctness:** 2
**Technical Novelty And Significance:** 2
**Empirical Novelty And Significance:** 2
**Recommendation:** 5
**Confidence:** 4

**Main Review:**

- **Writing**

In general, the writing of the main paper is good and well organized, which is not the case for the appendix, where there are many typos and missing details in the proofs.

I would point the following statement in the main paper. At the end of page 4, it is mentioned that $\zeta_*^2$ is scaled by exponentially decaying term in the lower bound (6), while $\zeta^2$ is scaled by $\frac{1}{R^2}$ in the rate of FedAvg (6). Then, Theorem 1 claims to improve exponentially the dependence on heterogeneity from $\frac{\zeta^2}{R^2}$ to $\frac{\zeta^2}{R^2} \textrm{exp}(-\frac{R}{\kappa})$. What about the rate of SCAFFOLD, where instead of $\zeta^2$ it is (smaller term) $\zeta_*^2$ and dependence is $\frac{\zeta_*^2}{\beta^2} \textrm{exp}(-\frac{R}{\kappa})$ ?


- **Correctness and optimality of the rates.**

First, the sub-optimality rates of FedAvg$\to$SGD and FedAvg$\to$ASG given in Table 1 are slightly different from those shown in Theorems 4&5. For example, for FedAvg$\to$ASG, it $\min(\Delta, \frac{\beta\zeta^2}{\mu^2 R^2})$ in Table 1, and $ \frac{\beta\zeta^2}{\mu^2 R^2}$ in Theorem 5 (see (27)). More importantly, I failed to get the claimed sub-optimality rates from the proof.

Let me provide some details for the proof of FedAvg$\to$ASG. From the proof of Theorem 5, we get (28) (ignoring constants and poly-log factors) by running FedAvg.
$$
\mathbb{E}[F(x^{(1)})] - F(x^*) \le  \Delta_1 := \beta \\,\textrm{exp}(-\frac{KR}{\kappa})\||x^{(0)} - x^*\||^2 + \frac{\sigma^2}{\mu N K R} + \frac{\beta \sigma^2}{\mu^2 K R^2} + \frac{\beta \zeta^2}{\mu^2 R^2} \qquad (28)
$$


Then switching to ASG, we have (31)
$$
\mathbb{E}[F(x^{(2)})] - F(x^*) \le  2\Delta_1 \\,\textrm{exp}(-\frac{R}{\sqrt{\kappa}}) + \frac{\sigma^2}{\mu N K R} + \frac{\beta \sigma^2}{\mu^2 K R^3} \qquad (31)
$$

The proof is terminated by plugging $\Delta_1$ in (31). However, it is easy to see that then we have (among others) the term $\beta \textrm{exp}(-\frac{K R}{\kappa}) \textrm{exp}(-\frac{R}{\sqrt{\kappa}})$, which remains even in the case when $\zeta^2=\sigma^2=0$. On the other hand, the rates shown in Table 1 and in Theorem 5 vanish in this case. Furthermore, if we let $\zeta^2=\zeta_*^2=0$, then the lower bound becomes $\frac{\sigma^2}{\mu N K R}$, while the actual rate of FedAvg$\to$ASG includes other terms like $\frac{\beta \sigma^2}{\mu^2 K R^2}\textrm{exp}(-\frac{R}{\sqrt{\kappa}})$.

A similar argument can be made for FedAvg$\to$SGD in the proof of Theorem 4 when plugging (24) into (26).

Could you please provide complete proofs of the theorems, addressing the above points, either in the rebuttal or updating the paper (if possible)?


- **Technical novelty**

Most of the analyses heavily use the results of previous works. Besides, the analyses of the central claims of Theorems 4 (1 in the main paper) and 5 (2 in the main paper) are direct concatenations of the previous rates. I did not find anything new in the analyses.

- **Experiments**

Experiments do not seem to be aligned with the given theory.  In the convex experiment in Figure 2, the best performing method is SCAFFOLD$\to$SGD, which (i) has an inferior rate than, say SCAFFOLD$\to$ASG, and (ii) as far as I get, the presented theory for SCAFFOLD$\to$SGD is the same as FedAvg$\to$SGD, which is not the case for SCAFFOLD versus FedAvg. It is mentioned that the good performance of SCAFFOLD$\to$SGD can be attributed to small $K$ in the experiment. Do you have an experiment with large $K$ with the predicted behavior?

The non-convex experiment in Figure 3 is strange too. Multi-staging does improve over the two-stage method, and in the second plot, FedAvg almost reaches to FedAvg$\to$SGD.









**Summary Of The Paper:**

It is known that if the level of heterogeneity is sufficiently high, then accelerated minibatch SGD is optimal for federated optimization matching the known lower bound of (Woodworth et al., 2020a). On the other hand, when the level of heterogeneity is very low, then FedAvg/LocalSGD outperforms the former in terms of communication complexity and needs only a few communication rounds given enough local computation. This paper *proposes* a multi-stage optimization procedure and *claims* that it nearly matches the lower bound for all heterogeneity levels.

**Summary Of The Review:**

The paper provides a clear intuition on why multistage algorithms are useful in practice. However, I made several critical comments in my main review regarding the correctness of the results, the technical novelty of the proofs, and the connection between experiments and theory. As of now, the paper seems weak (at least the way it is presented now). However, I am looking forward to reading the authors' rebuttal.

---

> ### Author Response · Authors · 2021-11-23
> **Response to Reviewer y87n [1/2]**
>
> We thank the reviewer for their comments.
>
> **On correctness**:
>
> We would like to emphasize that K (local computations) can be scaled up to a sufficiently large number in our model, because we are minimizing only communication complexity.  Accordingly, every theorem statement (Theorems 1,2,3,4,5) mentions that we must take local steps K to be large enough.
>
> We have updated the paper (Section 2, all formal theorem statements in the appendix) to explain this assumption more clearly and explicitly.  Furthermore, we write out all constraints in K in proofs and in the formal theorem statements.
>
> **On technical significance and novelty**:
>
> Given the framework, we agree that the proof is simple and straightforward. We see this as a strength of our framework, and this showcases its generalizability in combining centralized and federated algorithms to achieve better theoretical guarantees. Further, we believe the proposed framework is novel for the following reasons:
>
> 1. The framework’s exponential improvement in communication complexity over previous work in heterogeneous-data FL [1,2,3,4,5,6] was not previously known (to the best of our knowledge).  It also resolves an open problem from [1] (end of Section 5) by producing improvements over AC-SA when $\zeta_*$ is bounded but not insignificant.  Thus, we believe our framework’s contributions are significant and novel.
>
> 2. The framework does combine old algorithms in a simple and intuitive way.  But the simplicity should not count against the framework’s novelty.  We believe this simplicity is a positive attribute, as it encourages adoption in practice and provides clarity in understanding where the gain is coming from. Our proposed changes are simple but  effective in achieving performance that has remained unachievable in the literature with existing algorithms.
>
> 3. The idea of starting with a local update algorithm and finishing with a centralized algorithm for heterogeneous FL appears obvious in hindsight after our work.  However, we would like to emphasize that even in the influential recent work of [7], it was not known that one should combine these approaches.  Indeed, they posed an open question (end of section 6 in [7]) of whether one could constructively combine the strengths of both FedAvg and SGD together to achieve faster convergence rates.  We give a new (and simple) approach that answers this open question affirmatively in the heterogeneous setting, which is technically significant and novel.
>
> **On comparing SCAFFOLD to our rates**:
>
> Here we will compare SCAFFOLD's rate to our best rate, which is FedAvg->ASG.  Let K be taken large enough to match the terms that do not decrease with K.  Then the rate of SCAFFOLD [4] is $(\beta \Delta + \frac{\mu \zeta_*^2}{\beta^2})\exp(-R/\kappa)$ while the rate for FedAvg->ASG is $\min(\Delta, \frac{\beta \zeta^2}{\mu^2 R^2} )\exp(-R/\sqrt{\kappa})$.
>
> Notice the difference between $+$ and $\min$.  We can see then that FedAvg->ASG’s rate is better than that of SCAFFOLD, because (1) $\min(\Delta, \frac{\beta \zeta^2}{\mu^2 R^2} ) < \beta \Delta + \frac{\mu \zeta_*^2}{\beta^2}$ as long as $\beta > 1$, And (2) $\exp(-R/\sqrt{\kappa}) < \exp(-R/\kappa)$.
>
> In fact, the rate for SCAFFOLD [4] is worse than even AC-SA, where (again, assuming K is large enough), we have that $\Delta \exp(-R\sqrt{\kappa}) < (\beta \Delta + \frac{\mu \zeta_*^2}{\beta^2})\exp(-R/\kappa)$ as long as $\beta >1$.
>
> 1. Woodworth et al., "Minibatch vs local sgd for heterogeneous distributed learning." NeurIPS 2020
> 2. Gorbunov et al., "Local sgd: Unified theory and new efficient methods." AISTATS 2021
> 3. Mitra et al., "Linear Convergence in Federated Learning: Tackling Client Heterogeneity and Sparse Gradients." NeurIPS 2021
> 4. Karimireddy et al., "Scaffold: Stochastic controlled averaging for federated learning." ICML 2020
> 5. Reddi et al., "Adaptive federated optimization." ICLR 2021
> 6. Yuan et al., "Federated composite optimization." ICML 2021
> 7. Woodworth et al., "Is local SGD better than minibatch SGD?" ICML 2020

---

> > ### Author Response · Authors · 2021-11-23
> > **Response to Reviewer y87n [2/2]**
> >
> > **On experiments**:
> >
> > You are correct in seeing that the presented theory for SCAFFOLD->SGD is the same as FedAvg->SGD.  We presented two rates for SCAFFOLD: the rate from [4] and the rate from Theorem 10.  In the rate from [4], they take the local stepsizes very small, while in our rate in Theorem 10, we make the stepsizes equal to the FedAvg stepsizes in [2].  The latter is more in line with what we do in the experiments.  While this rate is not better than FedAvg’s rate, it has been observed empirically [4] that SCAFFOLD performs well on the logistic regression problem.
> >
> > We include a new section in the appendix of the revised paper (Appendix M) which empirically validates the theory that larger K benefits our algorithms that utilize acceleration in the second superstage.
> >
> > We assume that "Multi-staging does improve over the two-stage method” was a typo and "Multi-staging does not improve over the two-stage method” was meant, because in the submitted paper the two-stage methods performed best in the EMNIST experiment of the original paper.  The reason was due to poor tuning choices.  We update the paper to fix this (Fig 3, left).
> >
> > In the CIFAR-100 plot, it is clear that we have yet to reach the noise floor from heterogeneity in FedAvg--the experiment should be run for longer to observe more separation between the lines.  However, ResNet-18 on CIFAR-100 is a very computationally-intensive task, and we were only able to run the experiment for 500 rounds on the resources available to us (as opposed to 4000 in [5]).  Even on a low number of rounds, we still observe gains from our framework.  We are running the experiment for a larger number of rounds, but the experiment is computationally intensive and time consuming.  We hope to possibly present it to the reviewer before the discussion period ends.

---

> ### Author Response · Authors · 2021-12-01
> **A Gentle Request for the Final Feedback**
>
> We would like to thank the reviewer again for the encouraging feedback. We hope our response has adequately addressed your comments related to (1) the correctness of our results (2) novelty of our results (3) experimental verification of the behavior of our algorithms under large $K$ (4) other experimental concerns. Please kindly let us know if there are additional comments you have for us.

---

### Official Review · Reviewer_ZoaG · 2021-10-29

**Correctness:** 3
**Technical Novelty And Significance:** 2
**Empirical Novelty And Significance:** 3
**Recommendation:** 5
**Confidence:** 3

**Main Review:**

Please see the "Summary Of The Paper" for the brief summary.

From the theoretical point of view, in the strongly convex case, multistage algorithms indeed outperform the other methods like SCAFFOLD or minibatch algorithms.

From the empirical perspective, this paper considers the strongly convex case with logistic regression and the nonconvex case with neural networks. It also compares many algorithms, and I think the experiment part is generally enough.

However, I think this paper suffers from the following weaknesses:

1. In my point of view, neither the algorithm itself nor its analysis is interesting enough. The algorithm seems like a direct combination of different algorithms and the proof is also very straightforward: directly applying and combining the convergence analysis from the algorithm algorithms.

2. Missing important citation "Federated Accelerated Stochastic Gradient Descent" by Honglin Yuan, Tengyu Ma. This paper presents accelerated local methods. The results are not discussed in Table 1.

3. Although in the contribution part, the author claims: "This multistage has the added benefit of not requiring the knowledge of the noise bound $\sigma^2$, the heterogeneity $\xi^2$, or the initial distance to the optimum $\Delta$." However, in my understanding, the algorithm also needs to know the heterogeneity parameter $\xi^2$ in advance in order to choose the appropriate local steps $K$ in the first stage. Otherwise, I can choose all the functions $f_i$ to be the same thus $\xi^2 = 0$, and choosing $\sigma^2 = 0$. Theorem 4 and 5 imply that if $\xi^2 = \sigma^2 = 0$, it requires 0 communication rounds to achieve 0 training loss. Without knowing the heterogeneity parameters in advance, it seems impossible to do that. Also, please do not use "large enough $K$" in the theorems (e.g. Theorem 4 and Theorem 5), directly write out the constraints on $K$.

4. For the experiment hyperparameter selection, I think that the comparison is not very fair. In Appendix K "Experiment Setup Details", for single-stage algorithms, the global learning rate $\eta_g$ and the local learning rate $\eta_l$ are the same and they do not change during the training procedure. However, for two-stage algorithms, the learning rate for the second stage is $K$ times smaller than the global learning rate in the first stage. Even for one-stage algorithms, using two different learning rates may also speed up the training (i.e. we may not need to choose the local learning rate in the second stage to be 0, shrinking global and local LR in the second stage may also work).

Based on the previous concerns, I have the following suggestions and questions:
1. In terms of the algorithm, I think that it is better to analyze a single algorithm with different learning rates in different stages, e.g. only running FedAvg or SCAFFOLD with different learning rates. As minibatch algorithms are special cases for local step algorithms by choosing $\eta_l = 0$ (local learning rate), I believe that, in fact, we do not need to choose $\eta_l = 0$ in the second stage, we just need $\eta_l$ to be smaller.

2. Please have a check on the paper "Federated Accelerated Stochastic Gradient Descent" by Honglin Yuan, Tengyu Ma, and compare the results with multi-stage procedures. Or see if the results from that paper can improve the results for multi-stage algorithms.

3. This paper only analyzes the strongly convex case. I think that adding the analysis (just using different $\eta_g$ and $\eta_l$ in different stages, but not choosing $\eta_l = 0$ in the second stage) for the nonconvex case will make the results more interesting.

4. Please write the theorem more formally. Especially, should specify how $K$ is chosen.

5. I suggest adding some experiment results when keeping the same $\eta_g$ in different stages, or when decaying $\eta_g$ for a single algorithm, e.g. for FedAvg, using $\eta_{1,g} = \eta_{1,l} = 0.01, \eta_{2,g} = \eta_{2,l} = 0.005$.

Please point out if I have conceptual errors or misunderstandings.

**Summary Of The Paper:**

This paper introduces the multistage optimization technique for federated learning applications. Specifically, multistage optimization first uses federated optimization algorithms like FedAvg and SCAFFOLD and converges to some budget, and then uses minibatch algorithms like SGD or accelerated SGD in order to converge faster to a point with very small error. Because centralized methods are optimal when data are heterogeneous and local methods are optimal when data are purely homogeneous, using a multistage optimization technique can incorporate the benefits from both sides.

The theoretical part is relatively easy. The proof is to choose an appropriate error budget to which federated optimization algorithms converge, and then choose the hyperparameters for the federated optimization algorithms in the first stage and the minibatch algorithms in the second stage, e.g. learning rate, momentum, etc. The theoretical part only includes the convergence results for the strongly convex case.

The empirical part includes two experiments: logistic regression and neural network, which belongs to strongly convex case and nonconvex case respectively. For each experiment, the authors compare different minibatch algorithms like SGD, AGD, different local methods like FedAvg, SCAFFOLD, and some multistage procedures that combine local methods with minibatch methods. For the convex setting, multistage procedures perform the best, and for the nonconvex setting, multistage algorithms also perform generally the best.



**Summary Of The Review:**

In summary, I think this paper needs to be revised because of the following reasons:
1. The algorithm and the analysis seem a little incremental.
2. Missing important citation and comparison.

---

> ### Author Response · Authors · 2021-11-23
> **Response to Reviewer ZoaG [1/1]**
>
> We thank the reviewer for their comments.
>
> **On technical significance and novelty**:
>
> Given the framework, we agree that the proof is simple and straightforward. We see this as a strength of our framework, and this showcases its generalizability in combining centralized and federated algorithms to achieve better theoretical guarantees. Further, we believe the proposed framework is novel for the following reasons:
>
> 1. The framework’s exponential improvement in communication complexity over previous work in heterogeneous-data FL [1,2,3,4,5,6] was not previously known (to the best of our knowledge).  It also resolves an open problem from [1] (end of Section 5) by producing improvements over AC-SA when $\zeta_*$ is bounded but not insignificant.  Thus, we believe our framework’s contributions are significant and novel.
>
> 2. The framework does combine old algorithms in a simple and intuitive way.  But the simplicity should not count against the framework’s novelty.  We believe this simplicity is a positive attribute, as it encourages adoption in practice and provides clarity in understanding where the gain is coming from. Our proposed changes are simple but  effective in achieving performance that has remained unachievable in the literature with existing algorithms.
>
> 3. The idea of starting with a local update algorithm and finishing with a centralized algorithm for heterogeneous FL appears obvious in hindsight after our work.  However, we would like to emphasize that even in the influential recent work of [7], it was not known that one should combine these approaches.  Indeed, they posed an open question (end of section 6 in [7]) of whether one could constructively combine the strengths of both FedAvg and SGD together to achieve faster convergence rates.  We give a new (and simple) approach that answers this open question affirmatively in the heterogeneous setting, which is technically significant and novel.
>
> **On [Yuan & Ma., 2020]**:
>
> Thank you for mentioning this paper. While very interesting, it studies the homogeneous-data setting, i.e. $\zeta = 0$.  We include the citation in the revised draft (page 2), but the results are not comparable to the rates in Table 1.
>
> **On the multistage algorithm not requiring problem parameters**:
>
> The multistage algorithm we are referring to in the contributions section is the one in Theorem 3; the algorithms in theorem 4 and 5 are two-stage algorithms. We make this more clear in the revised draft (page 3).  The algorithm in Theorem 3 does not require knowledge of problem parameters, but the algorithms referenced in theorems 4 and 5 (correspondingly, 1 and 2) do.    The algorithm in theorem 3 does not require knowledge of problem parameters but in exchange gives up some communication efficiency.
>
> **On writing out constraints on K**:
>
> We have written out all constraints on K in the formal theorem statements of the appendix of the revised draft.
>
> **On investigating stepsize decay of single superstage algorithms**:
>
> Yes, we have updated the plots in the paper to include your suggestions (decaying learning rates for single-superstage algorithms) in App. N of the revised draft for the convex experiments and in Figure 3 (left) the EMNIST experiment.  In both of these settings, we confirm the gain of our framework versus these new baselines.  The CIFAR experiments are very computationally intensive and require substantial amounts of time to run.  We hope to possibly update the reviewer before the final discussion period is over.
>
> **On the nonconvex setting**:
>
> In the general nonconvex setting, technical difficulties arise.  Finding a way to decrease the error after the second superstage using the progress from the first superstage appears to be a difficult technical challenge: for example, one cannot get an upper bound on suboptimality after the first superstage in the nonconvex setting.  It is an important direction for future work.
>
> 1. Woodworth et al., "Minibatch vs local sgd for heterogeneous distributed learning." NeurIPS 2020
> 2. Gorbunov et al., "Local sgd: Unified theory and new efficient methods." AISTATS 2021
> 3. Mitra et al., "Linear Convergence in Federated Learning: Tackling Client Heterogeneity and Sparse Gradients." NeurIPS 2021
> 4. Karimireddy et al., "Scaffold: Stochastic controlled averaging for federated learning." ICML 2020
> 5. Reddi et al., "Adaptive federated optimization." ICLR 2021
> 6. Yuan et al., "Federated composite optimization." ICML 2021
> 7. Woodworth et al., "Is local SGD better than minibatch SGD?" ICML 2020

---

> > ### Comment · Reviewer_ZoaG · 2021-12-02
> > **After rebuttal**
> >
> > I have already read the rebuttal and my concerns about the experiments are alleviated. Currently, to me, this paper is a 5.5 point paper, which means that either 5 or 6 seems very reasonable to me. I will finalize my score after more thinking about the theoretical contributions and maybe discussing them with other reviewers.

---

> > > ### Author Response · Authors · 2021-12-02
> > > **Thank you!  A detailed discussion on the first suggestion of your original review**
> > >
> > > We thank the reviewer for their time and are happy to provide any information if it will help the reviewer with their decision.
> > >
> > > There is a technical detail we would like to make more concrete, regarding **theoretically analyzing only FedAvg or SCAFFOLD with different learning rates**.  Our message is that the theoretical evidence seems to suggest that setting $\eta_l > 0$ in the theory only introduces additional error terms.
> > >
> > > **On running only FedAvg with different learning rates**:
> > >
> > > Let $\sigma = 0$ and $K \geq \frac{\kappa}{R} \log(\frac{\mu^2 R^2 D^2}{\zeta^2})$ (all notation from the revised version of the paper).
> > >
> > > 1. Running FedAvg for the first stage using the stepsize choice from [2] gives the convergence rate $\frac{\beta \zeta^2}{\mu^2 R^2}$.
> > >
> > > 2. One can then run FedAvg using the stepsize choice from [4] (which sets $\eta_l \leq O(1/(\beta K \sqrt{N}))$, $\eta_g = O(1/\beta K)$, see Theorem 1 in [4]).  This setting of $\eta_l$ is quite small, shrinking with $N$ and $K$.  The convergence rate for FedAvg given this stepsize setting is $\frac{\beta \zeta^2}{\mu^2 R^2} + \Delta \exp(-R/\kappa)$ (Theorem 5 in [4]--note that if you set $B = 1$ in their heterogeneity assumptions, then $G = \zeta$, which makes their assumptions on heterogeneity consistent with ours).
> > >
> > > 3. The resulting convergence rate from using (1) in stage 1 and (2) in stage 2 is therefore $\frac{\beta \zeta^2}{\mu^2 R^2} \exp(-R/\kappa) + \frac{\beta \zeta^2}{\mu^2 R^2}$.
> > >
> > > 4. This convergence rate fails to even improve the convergence rate from (1), which had a convergence rate of $\frac{\beta \zeta^2}{\mu^2 R^2}$.  The reason, upon inspecting the proof of Theorem 5 in [4], is that setting $\eta_l > 0$ incurs significant error in $\zeta$.
> > >
> > > **On running only SCAFFOLD with different learning rates**
> > >
> > > Let $\sigma = 0$ and $K \geq \frac{\kappa}{R} \log(\frac{\mu^2 R^2 D^2}{\zeta^2})$ (all notation from the revised version of the paper).  Also assume $\beta \geq 1$.
> > >
> > > 1. Running SCAFFOLD for the first stage using the stepsize choice from our paper’s Theorem 10 gives gives the convergence rate $\frac{\beta \zeta^2}{\mu^2 R^2}$.
> > >
> > > 2. One can run SCAFFOLD using the stepsize choice from [4] (which sets $\eta_l \leq 1/(\beta K \sqrt{N})$ and $\eta_g = 1/(\beta K)$).  The convergence rate for SCAFFOLD given this learning rate, as noted in our Table 1, is $(\beta \Delta + \frac{\mu \zeta_*^2}{\beta^2}) \exp(-R/\kappa)$.
> > >
> > > 3. The resulting convergence rate from using (1) in stage 1 and (2) in stage 2 is therefore $(\frac{\beta^2 \zeta^2}{\mu^2 R^2} + \frac{\mu \zeta_*^2}{\beta^2}) \exp(-R/\kappa)$.  We incur an additional term of heterogeneity error: $(\frac{\mu \zeta_*^2}{\beta^2}) \exp(-R/\kappa)$ compared to if we used a centralized algorithm in stage 2 (such as ASG, Algorithm 3 in our paper), though it can be upper bounded by $\frac{\beta^2 \zeta^2}{\mu^2 R^2} \exp(-R/\kappa)$.
> > >
> > > 4. If we compare this to our FedAvg->ASG rate $\frac{\beta \zeta^2}{\mu^2 R^2} \exp(-R/\sqrt{\kappa})$, we see that our rate is better by a $\sqrt{\kappa}$ factor in the exponential error shrinkage rate, and better by a factor of $\beta$ in the term in front of the exponential.
> > >
> > > In summary, setting $\eta_l > 0$ in the second stage appears to only add additional heterogeneity error terms over setting $\eta_l = 0$ in the second stage.  In the case of FedAvg, it adds a significant amount of heterogeneity error.  In the case of SCAFFOLD, it adds some heterogeneity error that ultimately does not matter order-wise, but is added nevertheless.
> > >
> > > Note the settings of $\eta_l$ in both cases were less than $O(1/\beta K)$, yet still incurred additional heterogeneity errors!  This appears to rule out larger settings of $\eta_l$, which would incur larger additional heterogeneity errors.  Smaller (order-wise) settings of $\eta_l$, while possibly decreasing additional heterogeneity errors, would essentially be executing centralized algorithms ($\eta_l$ smaller than $O(1/\beta K)$ can only make one “step” of gradient progress per communication round, thus not materially improving on centralized algorithms), thus not improving on our Theorem 1 or 2.
> > >
> > > Given this evidence, we believe that setting $\eta_l = 0$ is the correct thing to do in the second stage, at least theoretically.  Our experimental results that we added in the revision (App. N and Figure 3) seem to also corroborate this.

---

> ### Author Response · Authors · 2021-12-01
> **A Gentle Request for the Final Feedback**
>
> We would like to thank the reviewer again for the encouraging feedback. We hope our response has adequately addressed your comments related to (1) the citation of Yuan & Ma (2) novelty of our results (3) evaluating single algorithms with decaying learning rates (4) writing explicit constraints on $K$ (5) correctness of multistage algorithm claims. Please kindly let us know if there are additional comments you have for us.

---

### Official Review · Reviewer_BBMC · 2021-11-01

**Correctness:** 4
**Technical Novelty And Significance:** 2
**Empirical Novelty And Significance:** 1
**Recommendation:** 6
**Confidence:** 4

**Main Review:**

- The paper is not well-written. I had difficulty following some key parts of the paper. see the comments below.
- The writing of the paper is misleading about the methods. It took me a while to figure out that all the methods and analyses that are covered in the paper require full participation from the clients. First of all, this limits the applicability of the results in any realistic FL scenarios as all FL algorithms across many devices can only work with limited participation. It is important to note that the dominating terms in convergence guarantees would actually change when one looks at the limited participation regime for each of these methods and it is not clear how much of the results translate to that regime. Moreover, this means that the methods that are called SGD, FedAvg, and .... are not what they are called. For example, FedAvg sub-samples the clients in each step and does not use full participation.
- The assumptions and many other necessary details about algorithms and theorems are unnecessarily relegated to the appendix. This made it very hard to understand the paper and results.
- I never understood what is difference between $\zeta$ and $\zeta_*$. Only one is defined in the assumptions.
- The StageWeightedAverage step of algorithm 1 has a lot of implications for the communication and memory of the method (as it requires access to the intermediate local updates of the clients). I believe this makes the algorithm impractical.
- In most of the rates, e.g. eq (9), the non-exponential terms are actually more important/dominant practically than the exponential part. But the authors are mainly focused on the exponential part (which is not dominating in practice). The only reasonable justification for that is assuming $K=\infty$ or very large, which is not necessarily the case and limits the significance of the results in practice. At the minimum, such a scenario should be elaborated in the paper and authors should refrain from over-claiming the significance of their results.
- Theorem 3 was very difficult to understand due to the complications in the definition of M-FedAvg and M-ASG.
- In Theorem 3 it seems that the regime in which the authors have proven the optimality of their methods ($\zeta^2<\kappa \mu \Delta$) is only a very small fraction of the original target regime ($\zeta^2<\sqrt{\kappa}\beta\Delta$) and gets even smaller with higher $\kappa$.
- The experimental section is very limited.
  - It is not very clear to me how the HP tuning is done for each algorithm. what HPs are tuned, and based on what metric and which values. This is of utmost importance because there are many algorithms with many different HPs. And in fact, one selling point of some multi-stage algorithms is their capability to be independent of many constants. But it is not clear to me how it plays in practice and is different from the other algorithms.
  - There is only one result on the test accuracy and it is very low (far from the state of the art) which makes the results not very convincing. The other result on accuracies in L.2 is not even clear which experimental setup it is referring to.

It is worth noting that I did not check the proofs thoroughly and in detail, but the claims of the paper seem to be reasonable and correct.

Minor comments:
- the authors mention that FedAvg's lower bound can only result in lower communication complexity than AC-SA when $\zeta_*^2\leq \mu \epsilon$. Is it possible to elaborate on how this result is driven?
- why is there a need to consider a batch size of B for Theorem 3; it seems to only affect the variance.

===================== After reading other reviews and the authors' responses ========================
- I agree with other reviewers that the novelty of the work is limited, especially given the fact that it applies to limited scenarios and does not even consider the general convex setting (which in my opinion would be more informative as the local updates can converge to different solutions of the same problem). Also, the obtained results are not as strong as advertised (for example as I mentioned Theorem 3 results are far from closing the gap). But at the same time, I commend the authors for improving the paper and including partial participation scenarios. Because of this, I will increase my score to 6.
- One point that was brought up by reviewer AN2w and I think does actually question the need and possible insight from the results is the fact that you do not need to run two stages with multiple steps, and the local update could all be done in 1 step and then run centralized method afterwards and get the same rate. This limits the importance of the achieved results (basically just changing the initialization for the centralized approach based on a local method could result in an optimal rate). I do not think that this is translatable to more complicated cases (other than strongly convex cases), but the authors have not empirically or theoretically studied the limitations of such an approach.

**Summary Of The Paper:**

The paper proposes "multi-stage FL algorithms" to bridge the gap between local step methods (that are theoretically known to be optimal for a no-heterogeneity regime where $\zeta=0$) and SGD type methods that are known to be good for high heterogeneity regimes. The goal of the paper is to find methods that are optimal across all values of $\zeta$, esp $0<\zeta^2< \beta^{3/2}\Delta/{\mu^{1/2}}$.

**Summary Of The Review:**

Based on the above comments, I believe the paper lacks in the areas of contribution, applicability, significance, practicality, experiments, and writing. As a result, I believe the current version of the paper is slightly below the acceptance criteria. I would like the authors to address the comments above before I can increase the score of the paper.

---

> ### Author Response · Authors · 2021-11-23
> **Response to Reviewer BBMC [1/3]**
>
> We thank the reviewer for their comments.
>
> **On technical significance and novelty**:
>
> Given the framework, we agree that the proof is simple and straightforward. We see this as a strength of our framework, and this showcases its generalizability in combining centralized and federated algorithms to achieve better theoretical guarantees. Further, we believe the proposed framework is novel for the following reasons:
>
> 1. The framework’s exponential improvement in communication complexity over previous work in heterogeneous-data FL [1,2,3,4,5,6] was not previously known (to the best of our knowledge).  It also resolves an open problem from [1] (end of Section 5) by producing improvements over AC-SA when $\zeta_*$ is bounded but not insignificant.  Thus, we believe our framework’s contributions are significant and novel.
>
> 2. The framework does combine old algorithms in a simple and intuitive way.  But the simplicity should not count against the framework’s novelty.  We believe this simplicity is a positive attribute, as it encourages adoption in practice and provides clarity in understanding where the gain is coming from. Our proposed changes are simple but  effective in achieving performance that has remained unachievable in the literature with existing algorithms.
>
> 3. The idea of starting with a local update algorithm and finishing with a centralized algorithm for heterogeneous FL appears obvious in hindsight after our work.  However, we would like to emphasize that even in the influential recent work of [7], it was not known that one should combine these approaches.  Indeed, they posed an open question (end of section 6 in [7]) of whether one could constructively combine the strengths of both FedAvg and SGD together to achieve faster convergence rates.  We give a new (and simple) approach that answers this open question affirmatively in the heterogeneous setting, which we believe is technically significant and novel.
>
> **On the applicability and significance of the full participation setting**:
>
> Full participation with data heterogeneity models the cross-silo setting in FL [8].  This is a setting where we are training a model using data situated at different organizations which do not want to explicitly share their data due to privacy or business needs.  Here, there are a few clients (typically 2-100 clients) that are fairly reliable, though the bottleneck can still be communication due to internet speeds and the size of models.  The canonical examples are models trained across organizations such as hospitals or financial institutions.  We include this motivation in the revised draft (page 1).
>
> Achieving low communication round complexity for FL in full participation has been a long-standing goal in the Federated Learning optimization literature, with several existing papers also studying the full participation setting as the sole focus [1,2,3,6] and more which studied full participation in one or more of their main results [4,5].
>
> We also believe solving the full participation setting is an important step towards solving the partial participation setting.  Our empirical results demonstrate that our improvements in the full participation setting also (experimentally) lead to improvements in the partial participation setting.  They show that our framework, which was designed with full participation in mind, also gets significant improvements under partial participation over previous algorithms.  For example, we demonstrate significant improvements in training loss on EMNIST ConvNet classification where 10 clients are sampled per round, out of 3400 clients.  And in CIFAR-100 where 10 clients are sampled out of 500 per round, in 500 communication rounds (4000 rounds were used in [5] for their state of the art results) we demonstrate that the framework performs the best in training loss and test accuracy.
>
> 1. Woodworth et al., "Minibatch vs local sgd for heterogeneous distributed learning." NeurIPS 2020
> 2. Gorbunov et al., "Local sgd: Unified theory and new efficient methods." AISTATS 2021
> 3. Mitra et al., "Linear Convergence in Federated Learning: Tackling Client Heterogeneity and Sparse Gradients." NeurIPS 2021
> 4. Karimireddy et al., "Scaffold: Stochastic controlled averaging for federated learning." ICML 2020
> 5. Reddi et al., "Adaptive federated optimization." ICLR 2021
> 6. Yuan et al., "Federated composite optimization." ICML 2021
> 7. Woodworth et al., "Is local SGD better than minibatch SGD?" ICML 2020
> 8. Kairouz et al., "Advances and open problems in federated learning." arXiv preprint arXiv:1912.04977 (2019)

---

> > ### Author Response · Authors · 2021-11-23
> > **Response to Reviewer BBMC [2/3]**
> >
> > **On our results applied to the partial participation setting**:
> >
> > Partial participation is an important setting to study in Federated Learning, as it encompasses the cross-device setting [8].  We include a discussion on it in the revised draft in App. L.
> >
> > In App. L, we show that the rates produced by our framework still do achieve some improvement in the partial participation setting.  We summarize the results in the following.  Let $\hat{\zeta}^2 = \text{sup}_{i,x} \|\nabla F_i(x) - \nabla F(x)\|^2$.  Concretely, FedAvg->SGD improves over SGD if $\frac{\hat{\zeta}^2}{\mu} \leq \Delta$, and similar is true for FedAvg->ASG vs ASG. FedAvg->SGD improves over FedAvg by a factor of R and S, and similar is true for FedAvg->ASG vs ASG.
> >
> > The gains are more modest compared to the full participation setting.  However, it is unclear whether the algorithm/analysis is tight or not, because there is no lower bound for the partial participation setting.  A lower bound for the partial participation setting is an important direction for future work.
> >
> > **On StageWeightedAverage**:
> >
> > Our framework does not require server-side access to intermediate local updates of the clients.
> >
> > In each round, client i can send
> > $\sum_{k=1}^K w_k^{(s,r)} x_{i,k}^{(s,r)}$
> > to the server, rather than all the individual iterates.
> >
> > We thank the reviewer for the comment and have clarified this in the revised draft (pg. 6).
> >
> > **On assuming K to be large**:
> >
> > We are at liberty to take K large because we are focusing on minimizing communication complexity.  In the important cross-silo setting [8], organizations may have fairly powerful computers but communication through the internet can be slow. We add this to the revision (Section 2).
> >
> > We see that we did not explicitly mention that local steps K are considered to be cheap under our model; we emphasize it in Section 2 in the revised draft.  Throughout the paper, we choose K large enough so that the terms that are shrinking with K match the terms that do not shrink with K.  In the revised draft, we have explicitly written out the constraints on K in the formal theorem statements in the appendix.
> >
> > **On the regime in which we prove the optimality of our methods**:
> >
> > Correct, the regimes in which we have proven the strict optimality of our methods are smaller than the target regime.  Our methods are only optimal up to exponentially shrinking condition number factors and assuming $\zeta_* \simeq \zeta$ (which, as mentioned in the paper, is what we mean by “nearly optimal”).
> >
> > **On the experimental section**:
> >
> > We should have mentioned in section 4 that the tuning process is detailed in App K.1 and App K.2.  This is noted in the revised draft (Section 4).  We have also updated the descriptions of the tuning process to be clearer.
> >
> > We believe our empirical results are significant: first, we demonstrate that our algorithms dominate other well-known algorithms in FL such as SGD, FedAvg, and SCAFFOLD in both training loss and test accuracy.  This both shows the strength of our algorithms on practical problems and also supports our theory.
> >
> > The results on test accuracy should be understood in the context of the regime we are looking at (low R).  State of the art results are achieved by running on many more rounds than we evaluate (4000 rounds on CIFAR-100 in [5] while 500 for us).  We have demonstrated that when R = 500, our methods perform the best.  We were unable to run experiments for such large values of R due to computational limitations. We explicitly mention this in the revised draft (Section 4).  We are currently running the CIFAR experiment for a larger number of rounds (though it is very computationally intensive and time consuming), and hope to possibly update the reviewer on it before the final discussion period is over.
> >
> > The Table in L.2. (now K.3 in the revised draft) is the EMNIST setting from App K.3.  We explicitly mention this in the revised draft.
> >
> > **On FedAvg's lower bound vs AC-SA**:
> >
> > From the discussion on page 4 of the revised draft (which was also in the submitted draft) we see that FedAvg can only ever outperform AC-SA in the noiseless setting if $\Delta \exp(-R/\sqrt{\kappa})$ requires more rounds to reach $\epsilon$ accuracy than $\frac{\kappa \zeta_*^2}{\mu R^2}$.
> >
> > So AC-SA requires $R \geq \sqrt{\kappa}$ rounds ignoring log factors, and FedAvg requires at least $R \geq \frac{\sqrt{\kappa}\zeta_*}{\sqrt{\mu} \sqrt{\epsilon}}$.  Setting $\sqrt{\kappa} \geq \frac{\sqrt{\kappa}\zeta_*}{\sqrt{\mu} \sqrt{\epsilon}}$ and solving for $\zeta_*^2$, we get that FedAvg's lower bound can only result in lower communication complexity when $\zeta_*^2 \leq \mu \epsilon$.

---

> > > ### Author Response · Authors · 2021-11-23
> > > **Response to Reviewer BBMC [3/3]**
> > >
> > > **Why is there a need for batch size B**:
> > >
> > > In the multistage algorithm there is a limit on how large K can be ($K \leq \kappa$).  We use B here to shrink the variance terms rather than K like in the other algorithms.  We make this more explicit in the revised draft (Theorem 3).
> > >
> > > **On writing & organization**:
> > >
> > > We regret that our section 2 in the paper was confusing. Please note that $\zeta$ and $\zeta_*$ were defined in the introduction and in the appendix; we have now included more explicit pointers to these definitions in Section 2 (modifications on pp. 1, 3).  Furthermore, we now emphasize that we consider full client participation (pp. 1, 3). We also include references to pseudocode to make Theorem 3 easier to parse.  We address these concerns in the revised draft.

---

> ### Author Response · Authors · 2021-12-01
> **Thank you!**
>
> Thank you for the useful comments and for reconsidering your evaluation!

---

### Decision · Program_Chairs · 2022-01-20

**Decision:**

Accept (Poster)

**Comment:**

The paper analyzes a 2-stage method for federated learning, first using FL with local steps, followed by a final phase of 'always-communicate' centralized SGD. For the convex case, the paper studies the influence of the data heterogeneity, a key parameter in FL, on the convergence of related schemes. Surprisingly the results of the 2-stage method seem to be basically identical to pure local training followed by the final centralized phase, and almost match the lower bound for communication.

Reviewers liked the interesting aspect of the heterogeneity-induced error floor when the phases are switched, and its impact on the convergence rates, which can be substantial. Downsides are that the analysis only works for strongly convex setting, and the combination of the two methods and proof being relatively straight-forward. Simplicity of the algorithm is a plus, while of the proof depends on novelty, about which reviewers are border-line but positive.

Deep learning experiments should be expanded, as there the very opposite order of the two phases https://arxiv.org/abs/1808.07217 is more commonly used (i.e. more communication in early phase can help), which should be discussed. Also, in the experiments the tuning of hyperparameters in the single-stage baselines needs to be improved to be more fair, which the authors have started but not fully finished for the Cifar case.

We hope the authors will incorporate the open points as mentioned by the reviewers.